# Haze in Singapore - Source Attribution of Biomass Burning PM$_{10}$ from Southeast Asia

Ayoe Buus Hansen[1], Claire Suzanne Witham[1], Wei Ming Chong[2], Emma Kendall[1], Boon Ning Chew[2], Christopher Gan[2], Matthew Craig Hort[1], and Shao-Yi Lee[2]

[1]Met Office, Exeter, UK
[2]Meteorological Service Singapore, Singapore

*Correspondence to:* Dr. A. B. Hansen (Ayoe.Hansen@metoffice.gov.uk)

**Abstract.** This paper presents a study of haze in Singapore caused by biomass burning in Southeast Asia over the six year period from 2010 to 2015, using the Lagrangian dispersion model the Numerical Atmospheric-dispersion Modelling Environment (NAME). The major contributing source regions to the haze are identified using forwards and backwards model simulations of particulate matter.

The coincidence of relatively strong Southeast monsoonal winds with increased biomass burning activities in the Maritime Continent create the main Singapore haze season from August to October (ASO), which brings particulate matter from varying source regions to Singapore. Five regions are identified as the dominating sources of pollution during recent haze seasons: Riau, Peninsular Malaysia, South Sumatra, and Central and West Kalimantan. In contrast, off-season haze episodes in Singapore are characterised by unusual weather conditions, ideal for biomass burning, and contributions dominated by a single source region

(different for each event). The two most recent off-season haze events in mid-2013 and early-2014 have different source regions, which differ from the major contributing source regions for the haze season. These results challenge the current popular assumption that haze in Singapore is dominated by emissions/burning from only Indonesia. For example, it is shown that Peninsular Malaysia is a large source for the Maritime Continent off-season biomass burning impact on Singapore.

     The results demonstrate that haze in Singapore varies across year, season, and location and is influenced by local and regional

weather, climate, and regional burning. Differences in haze concentrations and variation in the relative contributions from the various source regions are seen between monitoring stations across Singapore, on a seasonal as well as on an inter-annual timescale. This study shows that even across small scales, such as in Singapore, variation in local meteorology can impact concentrations of particulate matter significantly, and emphasises the importance of the scale of modelling both spatially and temporally.

## 1   Introduction

Haze caused by biomass burning is a significant issue throughout Southeast Asia. Biomass burning occurs naturally across the world, but is being accelerated by human activities and interests. Clearing forest for plantations by burning is a quick and easy way to open up and fertilise the soil, however, it is also a process that is difficult to control. The emissions from these fires

can have massive and detrimental impacts far from where the original fires were lit. Biomass burning is a global phenomenon. It is an ancient practice as well as a natural process which modifies the Earth's surface (Pereira et al., 2016). The haze from biomass burning impacts human health (Crippa et al., 2016; Sigsgaard et al., 2015; Youssouf et al., 2014; Reddington et al., 2015), crops, climate, bio-diversity, tourism, and agricultural production (Jones, 2006), and also aviation and marine navigation through visibility degradation (Crippa et al., 2016; Lee et al., 2016b). Over recent decades the impacts of biomass burning have been felt in increasing degree in Southeast Asia and in Singapore (Oozeer et al., 2016).

Though haze occurs in Singapore (Hertwig et al., 2015; Lee et al., 2016b; Nichol, 1997, 1998; Sulong et al., 2017), it is not caused by activities within Singapore. Rather it is a transboundary problem caused by biomass burning across the wider region (see Fig. 1 for a map of the region), which typically occurs during distinct burning seasons (Hertwig et al., 2015; Reid et al., 2013). Scientific studies such as Kim et al. (2015), as well as the popular press, often attribute peatland destruction and related haze in the region to Indonesia (Reid et al., 2013). However, the haze cannot be attributed to only one region or country alone. To mitigate this, the Association of Southeast Asian Nations (ASEAN) Haze Agreement has been formed between the Southeast Asian nations to reduce haze and mitigate the related impacts using a scientific approach (Nazeer and Furuoka, 2017; Lee et al., 2016a). Through the ASEAN, science-based mitigation has been attempted, but many lives are still lost every year due to haze caused by biomass burning (Lee et al., 2018). The Met Office (MO) and the Meteorological Service Singapore (MSS) have previously established a haze forecast system to predict haze in Singapore (Hertwig et al., 2015). This study advances the previous work to improve our understanding of haze and the underlying causes by analysing and attributing haze events of the recent past to their sources. The work focuses on Singapore due to the availability of air quality observations with high spatial and temporal resolution for recent years.

The weather and climate in Singapore and hence the transport of smoke from biomass burning is dominated by monsoon periods and influenced by the variations of the El Niño Southern Oscillation (ENSO), which modifies temperatures in the central equatorial pacific (Ashok et al., 2007; Yeh et al., 2009; Reid et al., 2012; Yuan and Yang, 2012). Meteorologically, the year in Singapore is split into four seasons, with two monsoon seasons separated by two inter-monsoon seasons. The north-east monsoon season is generally from December to early March and dominated by northeasterly winds. The first inter-monsoon period follows from late March through May, then the south-west monsoon is from June through September, with air in Singapore generally arriving from a southeastern direction. The second inter-monsoon period is October and November (Fing, 2012). Between years, there is large variability in the onset of the monsoon over Mainland Southeast Asia (Zhang et al., 2002). Generally, the inter-monsoon periods are characterised by light and variable winds, influenced by land and sea breezes with afternoon and early evening thunderstorms (Reid et al., 2012). The later inter-monsoon period is often wetter than the earlier inter-monsoon period (Chang et al., 2005; Reid et al., 2012). The weaker winds during the inter-monsoon periods lead to air arriving in Singapore originating from the countries immediately west of and surrounding Singapore (Fig A1).

Previous studies have shown the importance of ENSO in relation to reduction in convection and precipitation over the Martime Continent (MC) and corresponding increase in haze in Southeast Asia (Ashfold et al., 2017; Inness et al., 2015; Reid et al., 2012). The ENSO conditions have varied significantly during the six year period of our study (2010 - 2015). During

2010, the conditions transitioned from a moderate El Niño to a moderate La Niña lasting through 2011. From 2012 to 2014 the ENSO conditions were neutral transitioning to very strong El Niño conditions in 2015, which lasted into 2016 (NOAA, 2017).

The combination of variation in ENSO (Fing, 2012) and anthropogenic land-use changes (Field et al., 2009; Shi and Yamaguchi, 2014) leads to considerable inter-annual variation in biomass burning and related emissions of particulate matter (PM)

in Southeast Asia. Biomass burning in the region can be divided into seasons that relate to the monsoon periods: February, March, and April (FMA) are dominated by burning in Mainland Southeast Asia; during May, June, and July (MJJ) burning starts in northern Sumatra and traverses southward; August, September, and October (ASO) is characterised by burning in Kalimantan and, in general, there is little burning in November, December, and January (NDJ) (Campbell et al., 2013; Chew et al., 2013; Reid et al., 2012, 2013). From annual weather reports by MSS (NEA, 2015; NEA, 2017), unusual weather events

from 2010 to 2015 and related haze events are linked. In 2010 a prolonged Madden-Julien Oscillation (MJO) dry phase caused a dry October, creating ideal conditions for biomass burning in the region and related haze in Singapore. 2011 began as an ENSO neutral year transitioning to La Niña, with dry conditions in early September and prevailing low level winds bringing $PM_{10}$ to Singapore from biomass burning in central and southern Sumatra. During the Southwest monsoon of 2012, an MJO dry phase created dry and ideal haze conditions in September. In June 2013 a typhoon (Gaveau et al., 2014) coincided with

major atmospheric emissions from peat fires in Southeast Asia (Oozeer et al., 2016). In 2014 Singapore experienced haze during another intense MJO dry phase and drought, described by Mcbride et al. (2015). 2015 was the joint warmest year (with 1997 and 1998) and second driest year on record. ASO 2015 saw the worst haze in recent history in Singapore (Huijnen et al., 2016; Crippa et al., 2016; Koplitz et al., 2016), caused by southwesterly/southeasterly winds and fires in Southern and Central Sumatra and Southern Kalimantan. Fire carbon emissions over maritime South-East Asia in 2015 were the largest since 1997

(Huijnen et al., 2016).

Haze concentrations in Singapore vary throughout the six year period from 2010 to 2015. Even though biomass burning contributes to (low) $PM_{10}$ concentrations in Singapore throughout large parts of the year, some peaks in the $PM_{10}$ observations can be explained by haze almost exclusively. In the six year period, haze occurs almost annually during the season of ASO, known as the haze season (see Fig. 4). Haze events occurring during other periods of the year are referred to as off-season or

atypical haze. In 2013 and 2014 two unique atypical haze events occurred in June and in FMA, respectively (Hertwig et al., 2015; Gaveau et al., 2014; Duc et al., 2016). These events caused extremely high $PM_{10}$ concentrations in Singapore.

Several previous studies have looked at attributing air pollution for different regions. Source attribution can be performed both through modelling and by looking at observations of air pollution in detail. For example, Heimann et al. (2015) carried out a source attribution study of air pollution in the United Kingdom (UK) using observations to distinguish between local

and regional emissions, whereas Redington et al. (2016) estimated the sources of annual emissions of particulate matter from the UK and the European Union (EU) by using the Numerical Atmospheric-dispersion Modelling Environment (NAME) model to look at threshold exceedences and episodes. Attribution studies have been performed using Eulerian models such as the Goddard Earth Observing System atmospheric chemistry model (GEOS-chem), the Community Multiscale Air Quality Modeling System (CMAQ), and the Weather Research Forecasting System - Sulphur Transport and dEposition Model (WRF-

STEM) to study both Asia and the Arctic (Ikeda et al., 2017; Kim et al., 2015; Sobhani et al., 2018; Yang et al., 2017;

Matsui et al., 2013) sometimes in combination with flight campaigns (Wang et al., 2011) to better constrain the emissions. Lagrangian models have also been used in combination with observations by Winiger et al. (2017). Combinations of Eulerian and Lagrangian models (Kulkarni et al., 2015) and Eulerian models and observations (Lee et al., 2017b) have been used to assess whether low visibility days were caused by fossil fuel combustion, biomass burning, or a combination of the two. In Southeast Asia, Reddington et al. (2014) used an Eulerian model to study haze and estimated emissions through a bottom up approach. Source attribution for studies of biomass burning related degradation of air quality and visibility in Southeast Asia has also been applied by Lee et al. (2017a) who used the WRF model to study the sensitivity of the results to different met data and emission inventories. Engling et al. (2014) also used observations and a chemical mass balance receptor model to compare the chemical composition of total suspended particulate matter on haze and non-haze days during a haze event in 2006.

The aim of this study is to investigate the spatial variation of haze across Singapore through source attribution. This includes the variation in concentration and the contributing source regions at different sites across Singapore. This has been achieved by linking meteorology, biomass burning emissions, and forwards and backwards dispersion modelling to study how the origin of haze has varied across Singapore during 2010 - 2015. Fire radiative power and injection height from the Copernicus Atmosphere Monitoring Service (CAMS) global fire assimilation system (GFAS, (Kaiser et al., 2012)) and higher resolution land-use data from the Centre for Remote Imaging, Sensing and Processing at the National University of Singapore have been used to calculate Particulate Matter with diameter of 10 $\mu m$ or less (PM$_{10}$) emissions from biomass burning in 29 defined source regions in Southeast Asia (Fig. 1). Using the Met Office's numerical weather prediction (NWP) model to drive the Numerical Atmospheric-dispersion Modelling Environment (NAME), a Lagrangian particle trajectory model, we are able to attribute the haze arriving in Singapore to its source region and study the difference between major contributing source regions at a western and an eastern monitoring station in Singapore. The model output is evaluated against PM$_{10}$ observations from the two monitoring stations. The paper is composed as follows: Sect. 2 describes the methods used in the study; Sect. 3 presents the results and evaluation, along with a more detailed study of four recent haze events. The results and related implications are discussed in Sect. 4.

## 2 Methods

This section describes the model used, the set up and input used for the simulations, as well as the methods used to evaluate the results.

### 2.1 The Numerical Atmospheric-dispersion Modelling Environment

We use a Lagrangian model because of its ability to track emissions and provide detailed information on source regions at any given location in the modelling domain. The Numerical Atmospheric-dispersion Modelling Environment (NAME) III v6.5 (Jones et al., 2007) is a Lagrangian particle trajectory model, designed to forecast dispersion and deposition of particles and gases on all ranges. NAME uses the topography from the relevant meteorological input and does not resolve buildings or terrain on scales smaller than the NWP. Emissions in the model are released as particles that contain information on one or

more species. During the simulation these particles are exposed to various chemical and physical processes. NAME includes a comprehensive chemistry scheme, but this is not used in this study, as we are interested only in primary PM. The only aerosol processes considered here are dispersion and wet and dry deposition of primary $PM_{10}$. In NAME the dry deposition is parametrised using the resistance-based deposition velocity and wet deposition is based on the depletion equation (Webster and Thomson, 2014). The advection is based on the winds obtained from the meteorology provided and a random component is added to represent the effects of atmospheric turbulence. NAME is driven by meteorological data, in this case the Met Office's operational weather prediction model (Davies et al., 2005), described below.

## 2.2 The Unified Model

The Unified Model (UM) is the Met Office's operational numerical weather forecast model. The UM is a global model based on the non-hydrostatic fully compressible deep-atmosphere equations of motion solved using at semi-implicit semi-Lagrangian approach on a regular longitude-latitude grid (Walters et al., 2017). Archived analysis meteorology from the global version of the UM was used to drive NAME. As the UM is an operational model, the dynamical core and spatial resolution have changed throughout the period, from ~40 km over ~25 km to ~17 km resolution. However, for the majority of the study the resolution is constant at 25 km. These upgrades are described in Walters et al. (2011, 2017), and the relevant changes for dispersion modelling are summarised in Table 1. These changes are not expected to have a significant impact on the results, e.g., no significant differences in the deposition are seen across the change from instantaneous precipitation and cloud to 3-hour mean data in 2013.

Global UM model meteorological data for 2013 have been evaluated using meteorological observations available at four sites across Singapore. The UM data are interpolated in NAME to obtain wind speed and direction, temperature, and relative humidity data for each location and an hourly time resolution. The results show that modelled wind speeds are higher on average than those observed during 2013 particularly during the monsoon seasons. Wind speeds are one of the most important factors affecting pollutant levels, particularly close to strong sources. Although haze in Singapore is predominantly caused by long range transport of biomass smoke, the higher wind speeds in the model may contribute to reducing modelled pollutant levels below those observed. There are some differences in wind direction between the model and observations, but the prevailing wind directions are captured well throughout the year.

Observed ambient temperatures are slightly higher and more variable on average than the model, although there is good agreement between the model and observations. Rainfall does not appear well represented with higher hourly means and more frequent low intensity events when compared to the observations, which show less frequent high-intensity rainfall associated with the convective activity that dominates rainfall within the tropics. Modelled total monthly rainfall is higher than observed during 2013, which may decrease modelled PM levels through wet deposition and contribute to the often negative bias observed in $PM_{10}$, see Sect. 3. As discussed in Redington et al. (2016) and Hertwig et al. (2015), the uncertainties from the meteorological data feed into the dispersion simulation.

**Table 1.** Summary of the changes in the global UM data over the period of this study, relevant to dispersion modelling.

| Start date | Approx. horizontal resolution | Relevant change |
|---|---|---|
| 1/1/2010 | 40 km | |
| 20/1/2010 | 25 km | Horizontal resolution increase |
| 30/4/2013 | 25 km | Change from use of instantaneous precipitation and cloud to 3-hour mean data |
| 15/7/2014 | 17 km | Horizontal resolution increase |

## 2.3 NAME forward model simulations

For the attribution forward NAME runs were conducted using the haze forecast set-up designed by Hertwig et al. (2015) and extending it to year-long haze simulations. Individual forward simulations were performed for each of the years from 2010 to 2015 for $PM_{10}$ for a domain covering 14°S - 23°N and 90°E - 131°E using the GFAS $PM_{10}$ biomass burning emissions described in Sect 2.3.1. Each run was initialised on the 1st of January and the simulation ran until the 31st of December of the same year. A maximum of 200 million model particles were emitted during the simulation and any particles leaving the domain were lost. The simulations used no boundary conditions and so there was no inflow of particles from the domain edges. From these simulations, modelled time series for the two monitoring sites described in Sect. 2.3.2 were produced.

### 2.3.1 Emissions and Source Regions

The $PM_{10}$ emissions used in this study were calculated from the Global Fire Assimilation System (GFAS, Kaiser et al. (2012)) v1.2 daily gridded fire radiative power (FRP) and injection height (IH) products, integrated with high resolution land-use data and emission factors in an approach aimed at combining the benefits of the MSS and GFAS v1.2 source approaches described in Hertwig et al. (2015). Additionally, the land cover map used has been updated to the 2015 version by Miettinen et al. (2016b), which now covers the entire Southeast Asia region, as compared to the earlier 2010 version (Miettinen et al., 2016a). The horizontal dimensions of the emissions were dx=dy=0.1°, and the material was released at varying heights based on the GFAS injection height information. Using the Lagrangian nature of the model, all emissions are tagged with source information to allow for assessment of contributing source regions and relative contributions. The choice of the GFAS data set as the basis for the source calculation was based on the need for daily emissions, as in the operational setup of Hertwig et al. (2015), and the good agreement of this with observations and consistency with the Global Fire Emission Database (GFED) data set documented previously, e.g., Kaiser et al. (2012) and Rémy et al. (2017).

For this study, 29 source regions have been defined to distinguish where the $PM_{10}$ from biomass burning originated from (see Fig 1). The Lagrangian nature of the model enables us to attribute the $PM_{10}$ concentrations at specific locations in Singapore to the individual source regions.

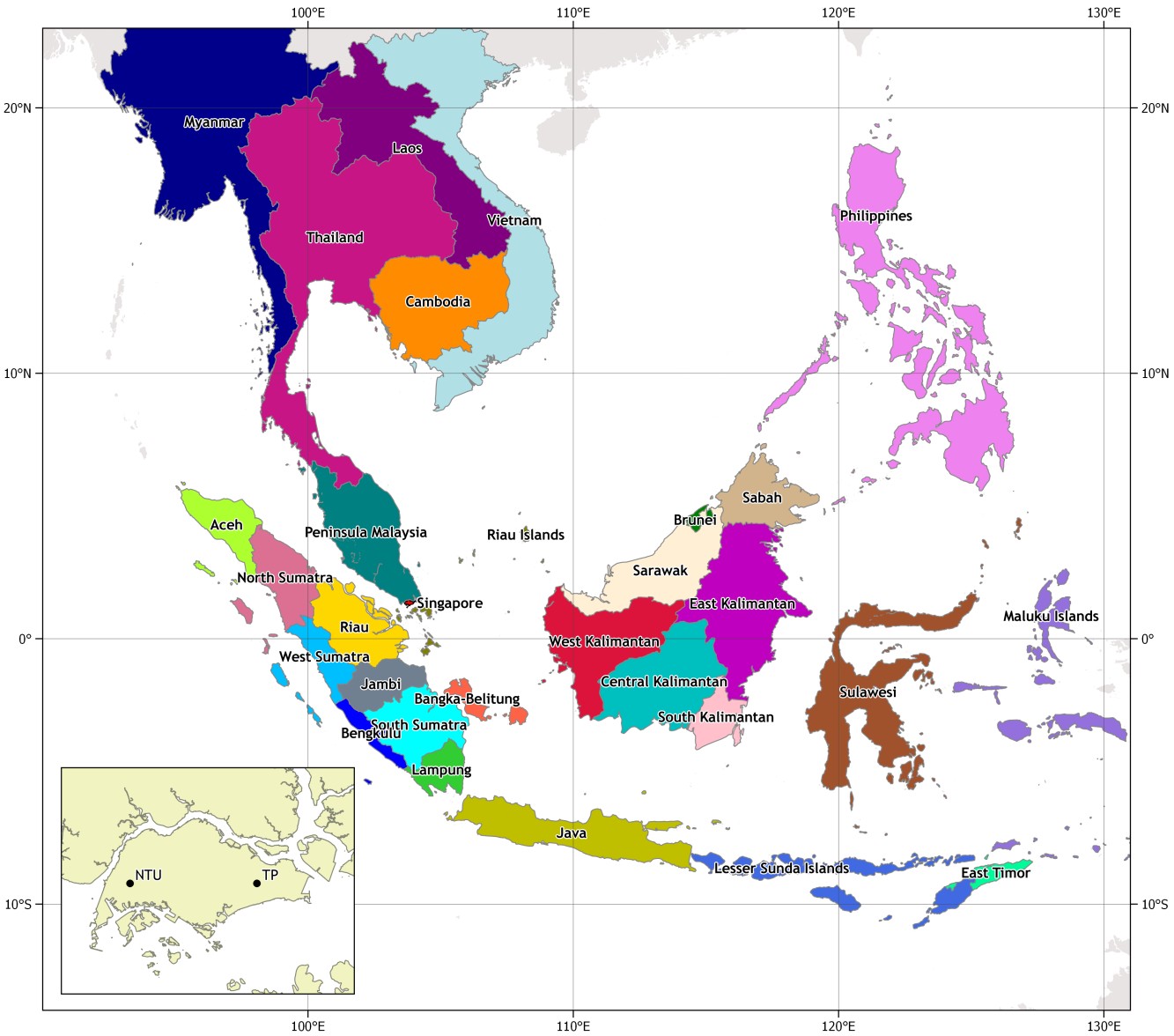

**Figure 1.** Locations and colour codes used for each of the 29 biomass burning source region within the domain from $10°$S - $20°$N and $90°$E - $130°$E considered in this study. Singapore is located south of Peninsular Malaysia and East of Riau. The insert in the bottom lefthand corner shows the relative location of the two monitoring stations in Singapore.

### 2.3.2 Observations and Performance Metrics

Some 20 air quality observation sites are located across Singapore. Of these, one eastern and one western station have been chosen to explore trans-boundary $PM_{10}$ concentrations across the main island of Singapore. In this analysis, the western station,

Nanyang Technological University (NTU; 1.34505°N, 103.6836°E), is located relatively close to the industrial western part of Singapore. The eastern station, Temasek Polytechnic (TP; 1.34506°N, 103.9304°E), is placed next to the polytechnic but is also near open fields and a water reservoir. The location of the two sites in Singapore can be seen in the insert of Fig. 1. The National Environment Agency of Singapore measures hourly $PM_{10}$ at these and other sites using beta attenuation monitoring.

5 In this technique air is drawn through a size selective inlet down a vertically mounted heated sample tube to reduce particle bound water and to decrease the relative humidity of the sample stream to prevent condensation on the filter tape. The $PM_{10}$ is drawn onto a glass fibre filter tape placed between a detector and a $^{14}C$ beta source. The beta beam passes upwards through the filter tape and the $PM_{10}$ layer. The intensity of the beta beam is attenuated with the increasing mass load on the tape resulting in a reduced beta intensity measured by the detector. From a continuously integrated count rate the mass of the $PM_{10}$ on the 10 filter tape is calculated.

The following analysis is based on hourly $PM_{10}$ observations and modelled time series at the two selected monitoring stations. Annual and seasonal pie charts showing the percentage contribution from each source region at each monitoring station have been produced to capture the spatial variation of biomass burning across the island, e.g., Figs 5c - 8c. During the period considered, several haze events occurred in Singapore. To evaluate the model results, four performance metrics have 15 been calculated. These evaluate the model performance at the two monitoring stations, for each year and selected seasons in each of the six years with available observations. The metrics considered are the Pearson correlation coefficient (R), i.e., the correlation between the model and observations used to get an indication of the match between patterns in the modelled and observed time series; the modified normalised mean bias (MNMB) which assesses the bias of the forecast and can have values between -2 and +2 (Seigneur et al., 2000); the fractional gross error (FGE) which gives the overall error of the model prediction 20 and is limited between 0 and +2 (Ordóñez et al., 2010; Savage et al., 2013); and finally, Factor of 2 (FAC2) which gives an indication of the fraction of the model results that fall within a factor 2 of the observations (Hertwig et al., 2015). Because the emissions used are at a daily resolution compared to the hourly observations of $PM_{10}$, a gap or mismatch in the timing of peak concentrations between modelled results and observation time series is possible. Biases between modelled time series and the observations are expected as some fires will be missed due to the fact that they are too small for the satellites to register and the 25 extent and/or duration of the other fires are over or under estimated due to cloud cover (Kaiser et al., 2012; Reid et al., 2013; Campbell et al., 2016).

## 2.4 Air History Maps

Air history maps provide a visual indication of where air at a given location has originated from. This helps to determine the regions that influence the composition of the air arriving at this location. To construct air history maps for Singapore, backward 30 (inverse) runs were conducted with NAME, in addition to the forward simulations with the GFAS biomass burning emissions (Sect. 2.3). Fig. 2 illustrates the air history map for Singapore for the years 2010 to 2015. For each day in the six year period from 2010 to 2015, a 10-day backrun was conducted using meteorological input from the UM global model within a domain of 90.0°E, 140.0°E, 15.0°S, 23.0°N (Fig. 2). $PM_{10}$ was emitted as a tracer from a receptor site in central Singapore and model particles were released over the first 24 hours with an emission rate of 1 g/s. The resulting concentration values in the 0-2km

**Table 2.** Overview of the four haze events studied in detail below. FMA: February, March, April; MJJ: May, June, July; ASO: August, September, October.

| Year | 2013 | 2014 | 2014 | 2015 |
|---|---|---|---|---|
| Season | MJJ | FMA | ASO | ASO |
| Section | 3.1 | 3.1 | 3.2 | 3.2 |
| Figure | 5 | 6 | 7 | 8 |

layer were output on a $0.1° \times 0.1°$ resolution grid and integrated backwards in time for 10 days with a timestep of 10 minutes. A higher integrated concentration indicates that more air has passed through a grid cell on route to the receptor site, compared to a grid cell with a lower concentration. By summing the results from multiple runs, air history data can be produced for different seasons and years, as well as the total for the whole period. For each analysis period, the multiple corresponding 10-day air concentrations were summed for each grid cell and for the total domain. A percentile value was then calculated to ascertain the proportion of air influenced by a particular grid cell vis-à-vis other areas.

Comparison between the inland site and a coastal receptor site showed insignificant variation, meaning that the central receptor site can be considered representative for the whole island when averaged over time. The results of the air history simulations helped inform the decision of domain size for the forward haze simulations.

# 3 Results

This section presents the results based on the modeling setup described in Sect. 2 above. Air history maps show where the air arriving in Singapore has travelled through and looking at the emissions provides information on when and where the largest emissions in the region occur. Using hourly $PM_{10}$ observations we evaluate our model output before using the results to address the research questions posed in Sect. 1. Four events are studied in more detail in the final subsections of this section, these are outlined in Table 2.

The air history map in Fig. 2 shows that most air arriving in Singapore has travelled from either northeastern or southeastern directions, illustrating the two monsoon seasons experienced in Singapore (see Fig. A1 for air history maps summed over the period for each of the individual seasons). The northeastern component of the bifurcation in the wind pattern is representative of the northeast monsoon in FMA (Fig. A1a), and the southeastern "fork" shows the southeast monsoon period during ASO (Fig. A1c). During the six years represented by the figure, significant variation occurs during the individual years (Fig. A2). In 2010 winds were quite weak and the air arriving in Singapore mainly came from a north-easterly direction and did not show the expected "fork" from the two monsoon seasons (Fig. A2a). This means that the air impacting Singapore that year mainly traversed through countries and regions very near to or east of Singapore, e.g., the Philippines, Peninsular Malaysia, Riau, and Riau Islands. The air history map for 2011 (Fig. A2b) shows a clear bifurcation, with air arriving from northeast and southeast, as expected from the two monsoon seasons. The air arriving in Singapore is therefore likely to have originated from

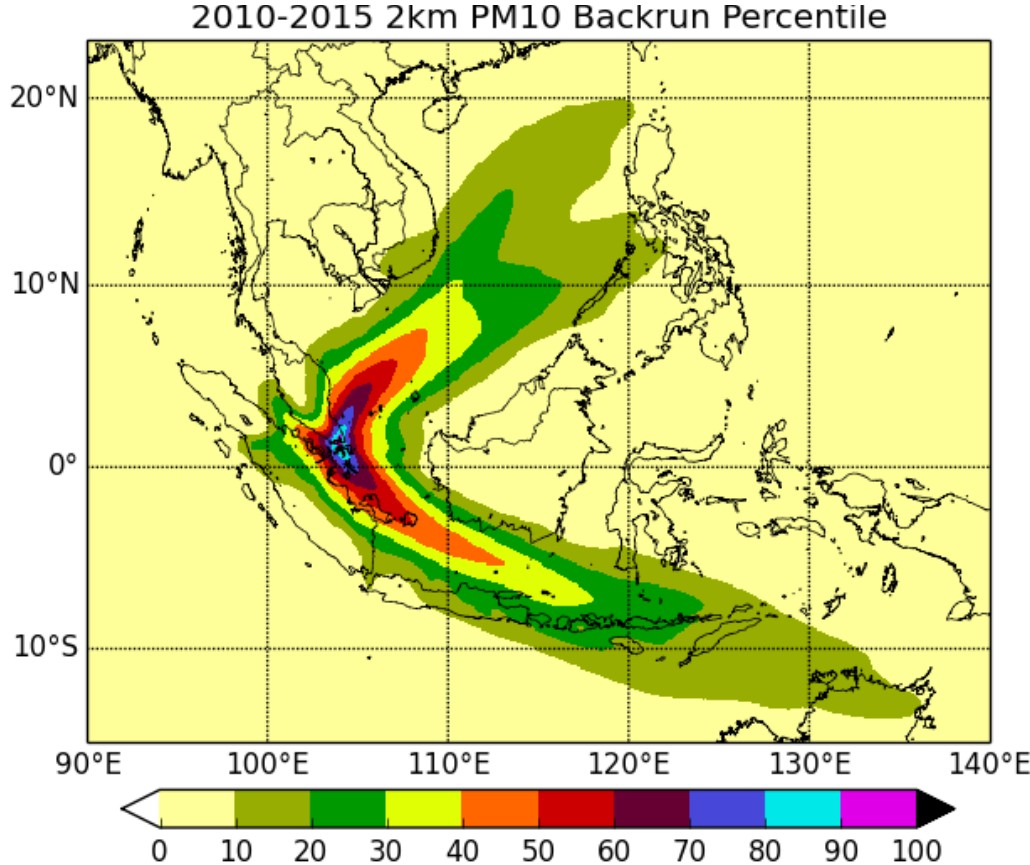

**Figure 2.** Air history map for 2010 - 2015, showing where air arriving in Singapore during this period originated from. Each shading shows the relative contribution of air to the central receptor site in Singapore in percent integrated over the atmospheric column from 0 to 2 km.

Vietnam, Cambodia, all areas of Kalimantan, Java, and the island of Sumatra including Riau. During 2012 the northeasterly wind component was significantly weaker than average. Also, a small northwesterly component is visible in the air history map (Fig. A2c). This means that air was mainly coming from the expected directions given the monsoons in the region with a small additional northwesterly component, so most air arriving in Singapore travelled through Peninsular Malaysia, or the
5   island of Sumatra including Riau. During 2013, the same general pattern as 2012 is seen but with stronger northeasterly and westerly components and somewhat weaker southeasterly component when the air history maps show a very small region of influence for the MJJ season of 2013. The majority of air arriving in Singapore had travelled only over Peninsular Malaysia or Riau. During other seasons of this year the air in Singapore arrived from as far away as Vietnam and the Philippines (Fig. A2d). 2014 was characterised by strong northeasterly and southeasterly components, both of which were stronger than those for 2013
10   and stronger southeasterly component compared to 2012 (Fig. A2e). The air history map for 2015 (Fig. A2f), shows a strong

northeasterly component and the strongest southeasterly component of all six years, these winds brought air from Peninsular Malaysia, Riau and Islands, Sumatra, Kalimantan, Sulawesi, Java, and the Lesser Sunda Islands to Singapore.

Analysis of the annual biomass burning $PM_{10}$ emissions (Fig. 3) shows that there is a bimodal pattern in the seasons/months with significant burning and also in the dominant source regions. This finding is similar to that of Reddington et al. (2014), though we see differences in the contributing source regions and temporal distribution. The most significant difference between the six years is in the magnitude of burning - note the different scales of vertical axis in Fig. 3. Overall, 2015 and 2014 were the years with the highest and second highest annual ($\sim 6.7 \times 10^6$ T and $\sim 4.2 \times 10^6$ T, respectively) and monthly ($\sim 2.7 \times 10^6$ T, October 2015 and $\sim 1.1 \times 10^6$ T, March 2014) emissions. 2010 and 2011 saw the lowest annual emissions ($\sim 2 \times 10^6$ T), though 2010 saw the third highest emissions when looking at individual months ($\sim 8.5 \times 10^5$ T, March). 2012 and 2013 saw fairly similar emissions ($\sim 2.5 \times 10^6$ T), which supports the fact that emissions are lower during La Niña and ENSO neutral conditions.

Over the six years, the highest emissions were generally seen during El Niño years and the drought of 2014. This makes sense as the majority of the fires are expected to be anthropogenic, and dry weather provides ideal conditions for initiating and maintaining burning (Reid et al., 2012, 2013; Oozeer et al., 2016). Lee et al. (2016b) looked at fire seasons and saw that there is anti-correlation between seasonal variation of fire emissions and that of rainfall, which is likely to be because underground peatland burning may not be immediately extinguished by precipitation. This also supports other papers, e.g., Reddington et al. (2014) who looked at fire/smoke seasons during the period 2004-2009 and found burning peaked from June to October and February to March, with the most burning during September - October.

Observations of $PM_{10}$ in Singapore from 2010 - 2015 show an overall background concentration during months of little or no burning of between 23 - 29 $\mu g/m^3$ at the two monitoring stations. These values fit well with those determined in other studies for Singapore. For example, Hertwig et al. (2015) estimated background concentrations for $PM_{10}$ to be around 30 $\mu g/m^3$, based on the 2013 haze episode. In general, both background and peak concentrations vary between NTU and TP. Following the approach of Kim et al. (2015) we assume a constant background of 25 $\mu g/m^3$ for the $PM_{10}$ observations at both sites and subtract this value from the observation time series.

Subtracting a constant background from the observations does not give the exact contribution of $PM_{10}$ from biomass burning alone because it does not remove all contributions from all other sources. However, it does give an indication of the periods with increased $PM_{10}$ concentrations due to biomass burning. This is not an attempt to perform an *apportionment* of the observed $PM_{10}$ concentrations in Singapore, as the observations, even with the subtracted background concentration, still includes contributions from sources other than biomass burning. However, the observations minus the constant background compared to the modelled time series provides an indication of the performance of the model, and through that the quality of the input used for the modelling. Using the modelled time series and the related source region information we are able to *attribute* the $PM_{10}$ contribution in Singapore originating from biomass burning in Southeast Asia to the respective source regions.

Because we are intentionally leaving out sources of $PM_{10}$ other than biomass burning and there is uncertainty in the biomass burning emissions, we cannot expect perfect scores from the valuation metrics presented in Tables 3 and 4. In the present study a significant haze event has been defined as any period lasting more than one week with modelled hourly $PM_{10}$ concentrations

**Table 3.** Statistics for PM$_{10}$, for both the western (NTU) and eastern (TP) monitoring stations and all years. Background concentration of 25 $\mu g/m^3$ is subtracted from the observations for all stations for all years. The metrics considered are the Pearson correlation coefficient (R), the modified normalised mean bias (MNMB), the fractional gross error (FGE), and Factor of 2 (FAC2).

|  | 2010 | | 2011 | | 2012 | | 2013 | | 2014 | | 2015 | |
|---|---|---|---|---|---|---|---|---|---|---|---|---|
|  | NTU | TP | NTU | TP | NTU | TP | NTU | TP | NTU | TP | NTU | TP |
| R | 0.12 | 0.12 | 0.08 | 0.13 | 0.17 | 0.18 | 0.79 | 0.80 | 0.27 | 0.35 | 0.44 | 0.43 |
| MNMB | 0.14 | 0.17 | 0.10 | 0.11 | 0.09 | 0.05 | 0.04 | 0.12 | -0.09 | 0.07 | -0.19 | 0.01 |
| FGE | 0.39 | 0.45 | 0.37 | 0.35 | 0.39 | 0.37 | 0.37 | 0.38 | 0.36 | 0.36 | 0.44 | 0.43 |
| FAC2 | 0.83 | 0.76 | 0.85 | 0.86 | 0.83 | 0.85 | 0.84 | 0.83 | 0.86 | 0.87 | 0.78 | 0.79 |

from biomass burning reaching 50 $\mu g/m^3$ or above at least at one of the two monitoring stations. Concentrations below 10 $\mu g/m^3$ are considered negligible in terms of haze events.

For years like 2013, which was dominated by one extreme haze event, the correlation between the modelled time series and the observations is very high (0.79 and 0.80 at NTU and TP, respectively, see Table 3). Whereas the correlations for 2010, 2011, and 2012 are very low, which is likely to be due to the low biomass burning PM$_{10}$ emissions and few haze events. In general it can be seen from the MNMB that the model under predicts, even when taking a constant background value of 25 $\mu g/m^3$ into account. This makes sense as the background in reality cannot be assumed to be constant. We know that we are not capturing all fires, which will lead to a negative bias, and there are further uncertainties in emissions, and the NWP and dispersion models. It should be expected that not all model results fall within a factor of 2 of the observations and it is not surprising that the fractional gross error is around 40 %. It is worth noticing that the FAC2 for all years is high (between 0.76 and 0.87), and in general the FAC2 values for the individual events are also very good. When comparing the scores to other studies such as Chang and Hanna (2004) (R = 0.91 - 0.95, FAC2 = 0.24 - 0.89) and Rea et al. (2016) (R = -0.33 - 0.92), it is important to keep in mind that even though the scores presented in Tables 3 and 4 are relatively lower (specifically R) our statistics are calculated for a three month period and other studies are for shorter periods focused only on air quality and haze days. Also, for the results presented here the FAC2 values are mostly better than those of Chang and Hanna (2004); Rea et al. (2016). In the results below, the estimated background value of 25 $\mu g/m^3$ has been subtracted from all observations. The timeseries and pie charts are based on results from the forward NAME simulations.

Looking at PM$_{10}$ concentrations at the two monitoring sites based on the forward simulations (Fig. 4), five years (all but 2013) have haze during ASO and three years (2011, 2013, and 2014) have some haze in FMA. 2013 is the only year with significant haze in June, although the years from 2012 to 2015 all experience some additional PM$_{10}$ from biomass burning in June. When comparing concentrations between the two stations it can be seen that the concentrations are higher at the western monitoring station (NTU) most of the time. Exceptions to this occurred during March 2011 and 2014. Of the haze events that occurred from 2010 through 2015, some were insignificant (e.g., during FMA 2010, 2012, 2013, and 2015, and MJJ 2012 and 2014), i.e., lasting less than a week and with biomass burning PM$_{10}$ concentrations below 50 $\mu g/m^3$. Some

**Table 4.** Statistics for $PM_{10}$, for both the western (NTU) and eastern (TP) monitoring stations, for selected 3 months haze seasons. Background concentration of 25 $\mu g/m^3$ is subtracted from the observations for all stations for all seasons. The metrics considered are the Pearson correlation coefficient (R), the modified normalised mean bias (MNMB), the fractional gross error (FGE), and Factor of 2 (FAC2).

| | 2010 ASO | | 2011 ASO | | 2012 ASO | | 2013 MJJ | | 2014 FMA | | 2014 ASO | | 2015 ASO | |
| --- | --- | --- | --- | --- | --- | --- | --- | --- | --- | --- | --- | --- | --- | --- |
| | NTU | TP | NTU | TP | NTU | TP | NTU | TP | NTU | TP | NTU | TP | NTU | TP |
| R | 0.15 | 0.14 | 0.08 | 0.15 | 0.14 | 0.14 | 0.81 | 0.83 | 0.30 | 0.42 | 0.29 | 0.40 | 0.35 | 0.32 |
| MNMB | 0.12 | 0.13 | -0.07 | -0.01 | -0.24 | -0.22 | -0.14 | 0.03 | -0.13 | -0.07 | -0.31 | -0.06 | -0.65 | -0.47 |
| FGE | 0.41 | 0.49 | 0.38 | 0.34 | 0.40 | 0.38 | 0.40 | 0.43 | 0.29 | 0.30 | 0.35 | 0.36 | 0.71 | 0.61 |
| FAC2 | 0.80 | 0.72 | 0.86 | 0.87 | 0.83 | 0.84 | 0.82 | 0.78 | 0.93 | 0.93 | 0.76 | 0.86 | 0.49 | 0.60 |

were significant but showed very little variation between monitoring stations (ASO 2010, MJJ 2013, FMA 2011 and 2014) (Sect. 3.1). The remaining four events (ASO 2011, 2012, 2014, and 2015) (Sect. 3.2), were significant events with variation in the main contributing source regions at the two monitoring stations. Common for all four events is that they occurred during the haze season in ASO during the southeast monsoon, when the winds are the strongest for the region and the air history maps show the largest region of influence for air arriving in Singapore. Not all peaks in the observations coincide with biomass burning due to real PM levels also containing anthropogenic and other biogenic species. However, most peaks in the modelled time series coincide with peaks in observations indicating that the highest $PM_{10}$ concentrations are due to biomass burning.

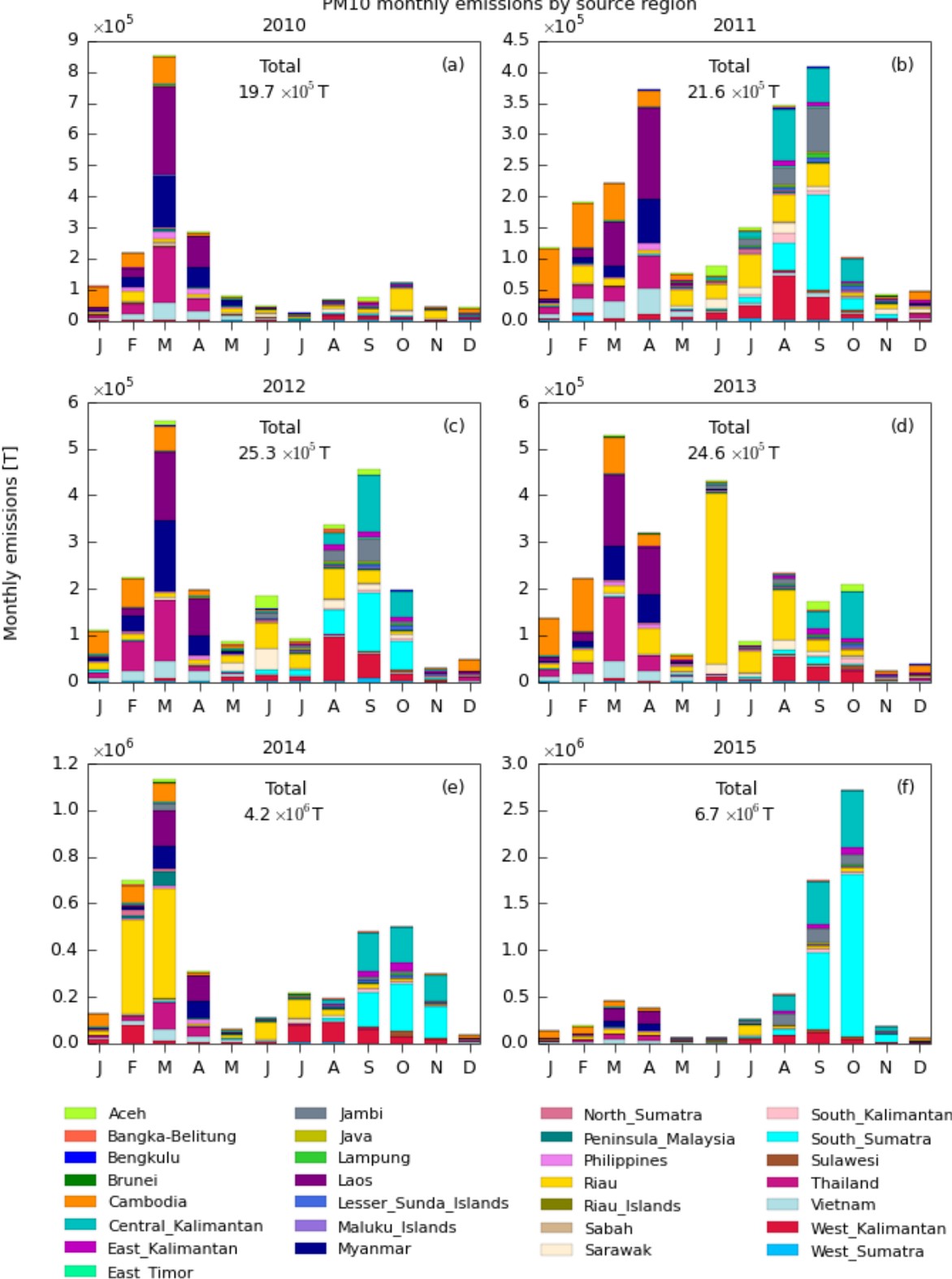

**Figure 3.** Caption on next page

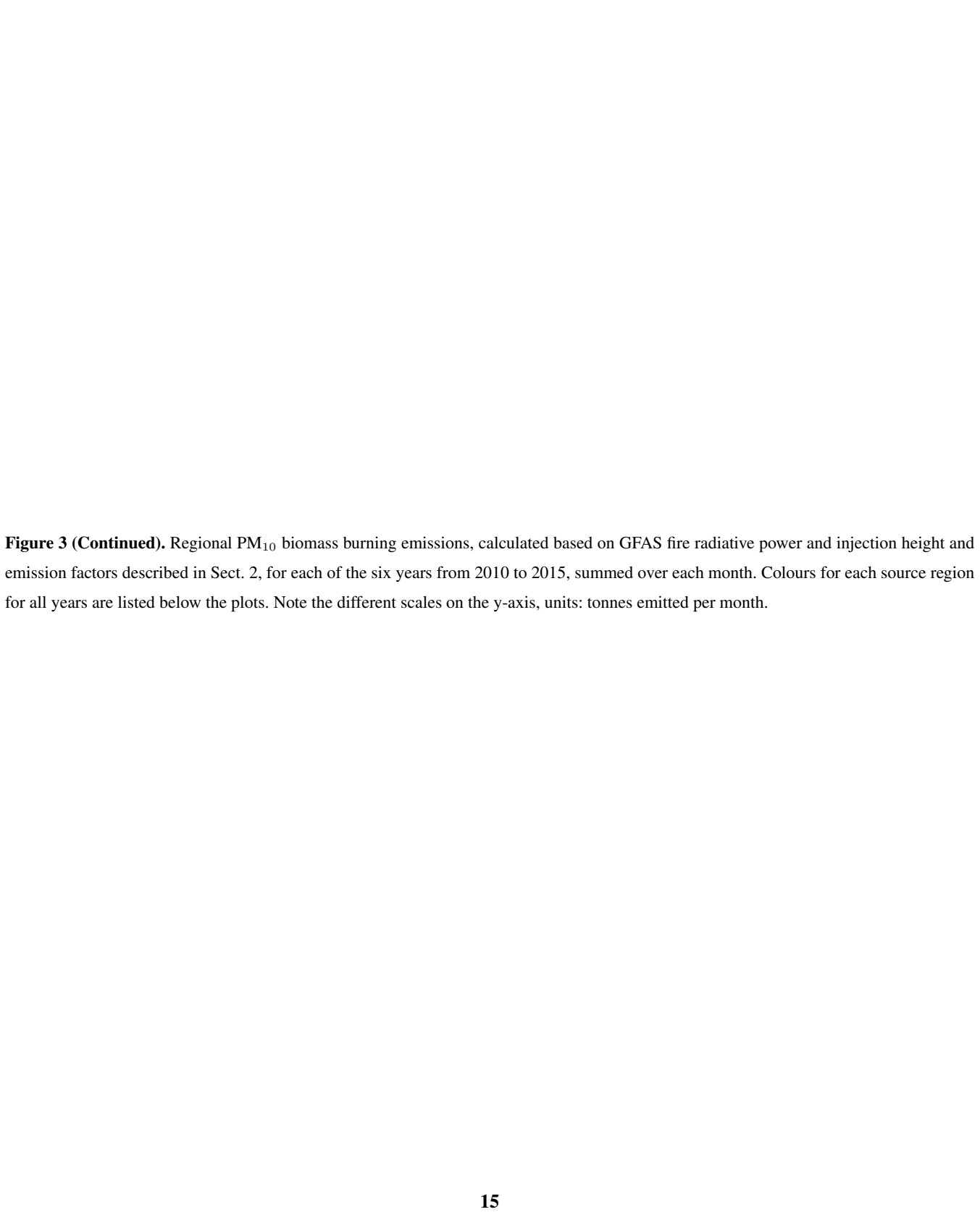

**Figure 3 (Continued).** Regional PM$_{10}$ biomass burning emissions, calculated based on GFAS fire radiative power and injection height and emission factors described in Sect. 2, for each of the six years from 2010 to 2015, summed over each month. Colours for each source region for all years are listed below the plots. Note the different scales on the y-axis, units: tonnes emitted per month.

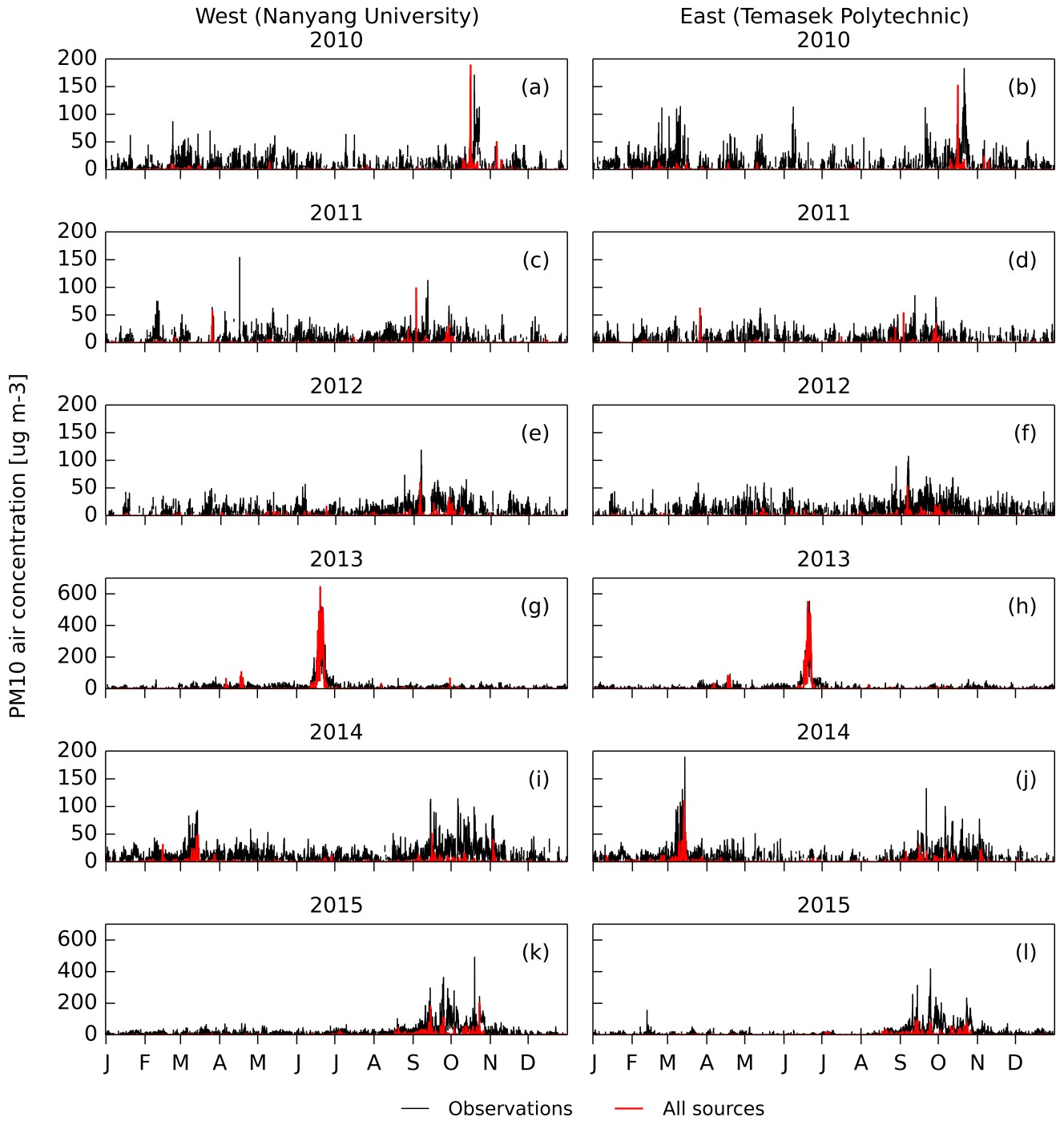

**Figure 4.** Modelled PM$_{10}$ time series (red line) with observations (black line) at each of the two monitoring stations West (NTU, left) and East (TP, right) for the six years with observations available, 2010 (top row) - 2015 (bottom row). A constant background concentration of $25\mu g/m^3$ has been subtracted from the observations and any resulting negative values have been removed.

## 3.1 Atypical haze

During the six years, the most notable atypical haze events occurred in June 2013 and February, March, April 2014. Though 2013 was generally a year with weak winds and average burning, the month of June was very unique, both in terms of meteorology and burning (Fig. 5). The June 2013 haze event was caused by a typhoon coinciding with intense burning in Riau (Fig. 3). The air history map for MJJ in Fig. 5 shows that, during this weather event, there was a small source region with air arriving in Singapore from Peninsular Malaysia, Riau Islands, and Sumatra including Riau. This is the only year of this six year period with significant burning in June, though in general the annual emissions are neither especially high nor low. In June about 98 % of the modelled $PM_{10}$ emissions reaching the two monitoring stations in Singapore were from Riau. Although the peak concentrations observed at NTU were lower than those of the modelled time series, overall the concentrations are fairly similar during the event.

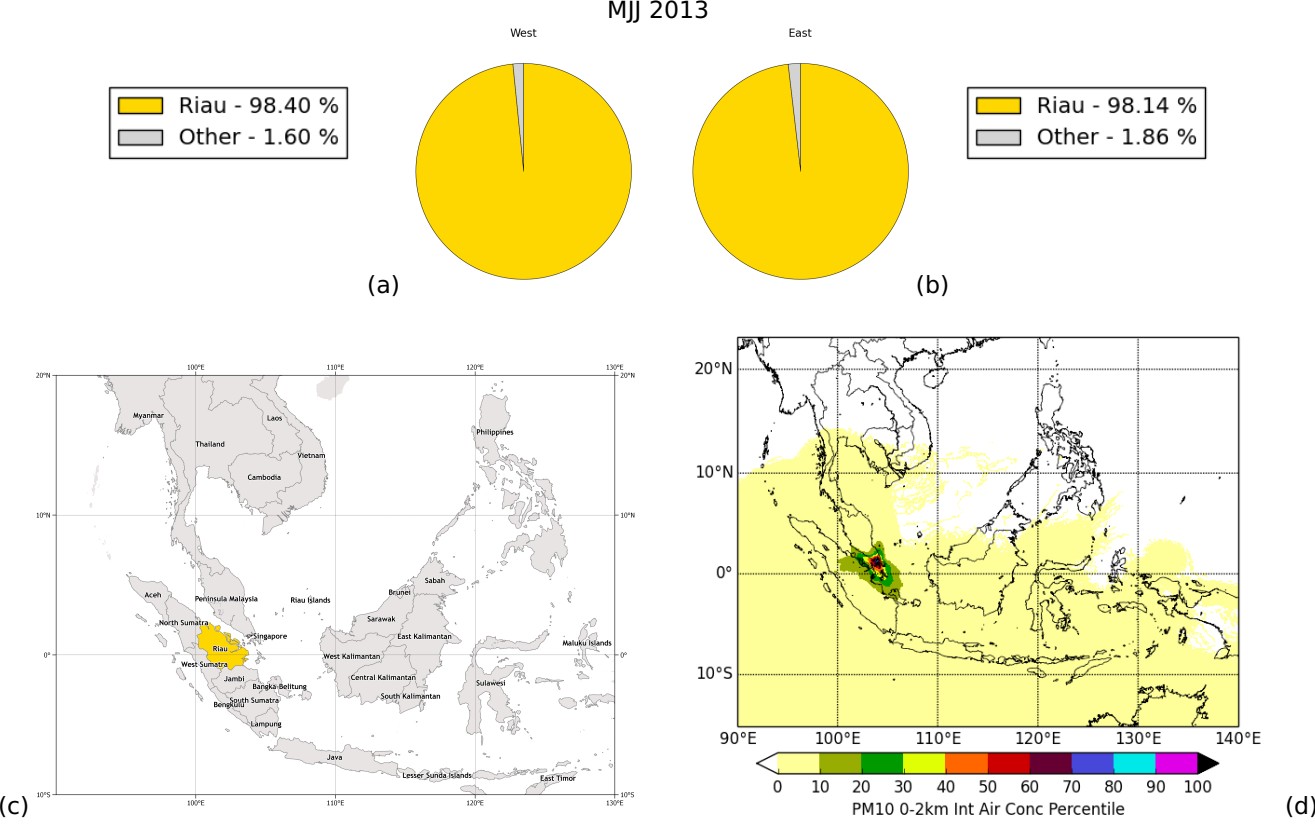

**Figure 5.** This figure shows results for $PM_{10}$ for MJJ 2013: Pie charts for the western (NTU) (a) and eastern (TP) (b) monitoring stations showing major contributing source regions, (c) shows the regional map highlighting only the major contributing source region, and the air history map (d) showing where the air arriving in Singapore originated from in MJJ 2013. The 'Other' category in the pie charts is from sources which individually contribute less than 1 %.

In early 2014, a drought coincided with air arriving in Singapore from a northeasterly direction and intense burning in the whole region giving the second highest emissions of the six year period. This resulted in unexpected haze in Singapore in FMA (Fig. 6). The months with the largest emissions were March and February which were dominated by emissions from Riau, Laos, Myanmar, Thailand, Cambodia, Peninsular Malaysia, and West Kalimantan (Fig. 3d). In general the region of influence for 2014 covered an area reaching far to the northeast and slightly south-east of Singapore and was much larger than for MJJ 2013 (Fig. 5). During FMA the winds brought air from Peninsular Malaysia, Riau, Riau Islands, and the Philippines to Singapore. In spite of the larger emissions from Riau, Laos, Myanmar, Thailand, and Cambodia, the mainly northerly wind direction resulted in the haze in Singapore being caused mainly by emissions from Peninsula Malaysia. The event lasted for about 3 months total, and was dominated by emissions from Peninsular Malaysia, which contributed over 90 % of the haze at both monitoring stations, with smaller contributions from Riau, Cambodia, Vietnam, and Riau Islands.

Common for these two atypical haze events is little variation in the source regions across the monitoring stations, most likely due to the atypical and different meteorological conditions and the clear dominance of one source region.

## 3.2 ASO - southeast monsoon season haze

As mentioned previously, the southeast monsoon season occurs during ASO and coincides with almost annual haze episodes. The two most recent episodes with highest concentrations were in 2014 and 2015. In addition to the haze event in FMA 2014 discussed above, another haze event occurred in 2014 during ASO (Fig. 7). This season saw the largest southeasterly region of influence for air arriving in Singapore during the six year period, with air and $PM_{10}$ from biomass burning pollution arriving in Singapore from Peninsular Malaysia, Riau, Riau Islands, Kalimantan, Java, and the Lesser Sunda Islands, during a period of average biomass burning emissions. During the two months of September and October the major contributing source regions to $PM_{10}$ concentrations in Singapore were Central Kalimantan, South Sumatra, and West Kalimantan (Fig. 3e). ASO is the expected haze season, however, this is also one of the seasons with the highest number of significant contributing source regions: South Sumatra, Central Kalimantan, West Kalimantan, Bangka-Belitung, Riau, Riau Islands, and the Lesser Sunda Islands (up to 2000 km from Singapore). In spite of the large annual variation (Fig. A3) in the major contributing source regions between the two monitoring stations, the difference between the relative contributions at the two stations for ASO 2014 is insignificant.

The results for ASO 2015 (Fig. 8) show a large, though seasonally "normal" region of influence, which coincided with extreme emissions. In ASO the southeasterly monsoon winds brought air from Peninsular Malaysia, Riau Islands, Sumatra including Riau, Kalimantan, Sulawesi, Java, and the Lesser Sunda Islands. During this season the largest contributing regions were Central Kalimantan, South Sumatra, and West Kalimantan. The event lasted approximately 2.5 months in ASO 2015, during which the biggest variation between the two monitoring stations was seen both for 2015 and for any season with significant burning. The most significant source regions at the western and eastern monitoring stations (NTU, TP) were South Sumatra (38.22 %, 21.82 %), Central Kalimantan (31.19 %, 41.45 %), Bangka-Belitung (11.32 %, 13.64 %), West Kalimantan (6.64 %, 9.41 %), and Jambi (6.53 %, 5.98 %). Common for both ASO 2014 and ASO 2015 are the relatively large regions

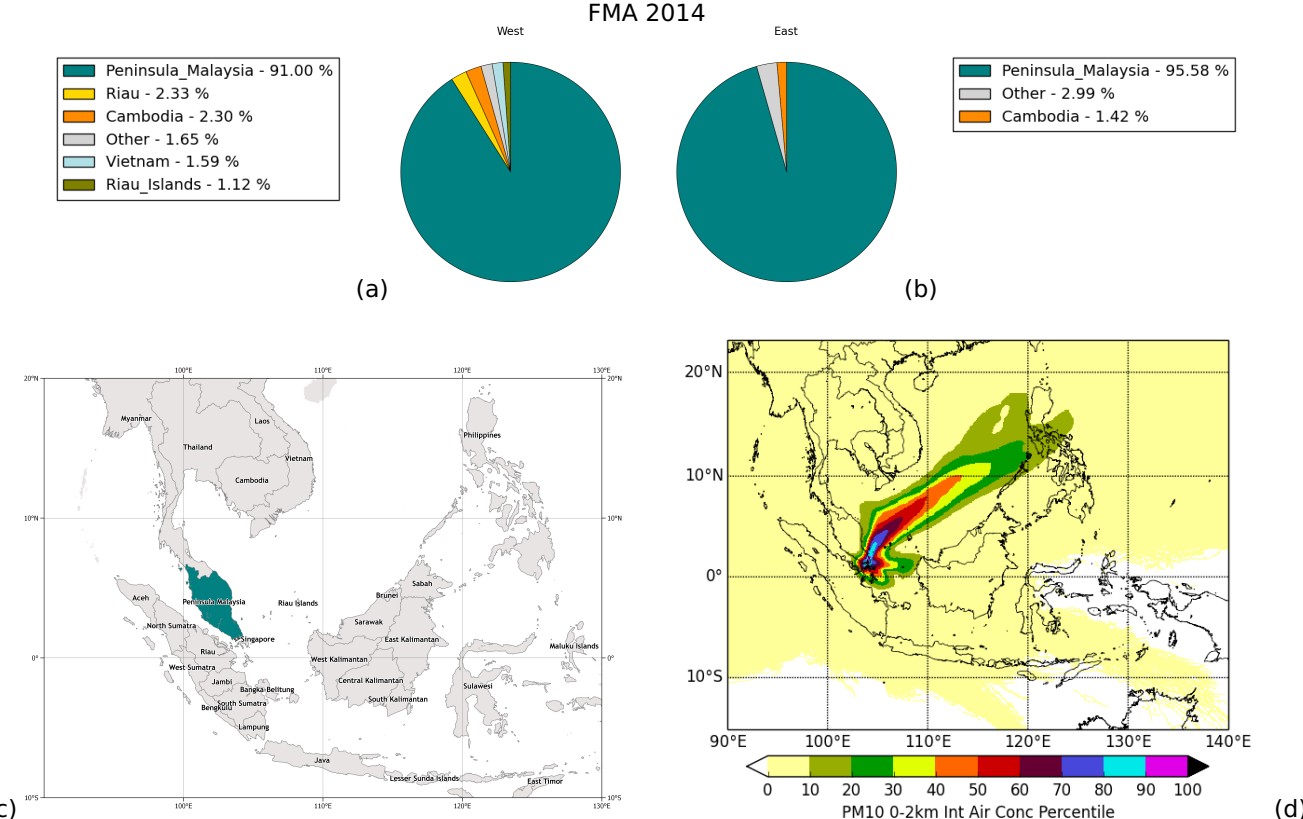

**Figure 6.** This figure shows results for $PM_{10}$ for FMA 2014: Pie charts for the western (NTU) (a) and eastern (TP) (b) monitoring stations showing major contributing source regions, (c) shows the regional map highlighting only the major contributing source region, and the air history map (d) showing where the air arriving in Singapore originated from in FMA 2014. The 'Other' category in the pie charts is from sources which individually contribute less than 1 %.

influencing $PM_{10}$ concentrations in Singapore and the variation in major contributing source regions at the two monitoring stations. This is also the case for other years with burning and related haze during this season (e.g., 2011 and 2012).

In addition to the four events discussed in detail above, events also occurred during the expected haze seasons in ASO 2010, 2011, and 2012, as well as during FMA 2011. The ASO event in 2010 was, except for significantly lower magnitude, fairly similar to the MJJ event of 2013, with an unusually small source region for the season and at least 90 % of $PM_{10}$ concentrations arriving at both monitoring stations in Singapore originating from Riau. The other two ASO events, in 2011 and 2012, were fairly similar to the events of 2014 and 2015 with contributions from the expected southeast monsoon region, a high number of contributing source regions at the two monitoring stations, and variations in major contributing source region between the two stations. The remaining event of the period was during FMA 2011, with Riau, Peninsular Malaysia, and Cambodia as major contributing source regions. Of the seasons with the most significant haze events (e.g., MJJ 2013, FMA 2014, ASO

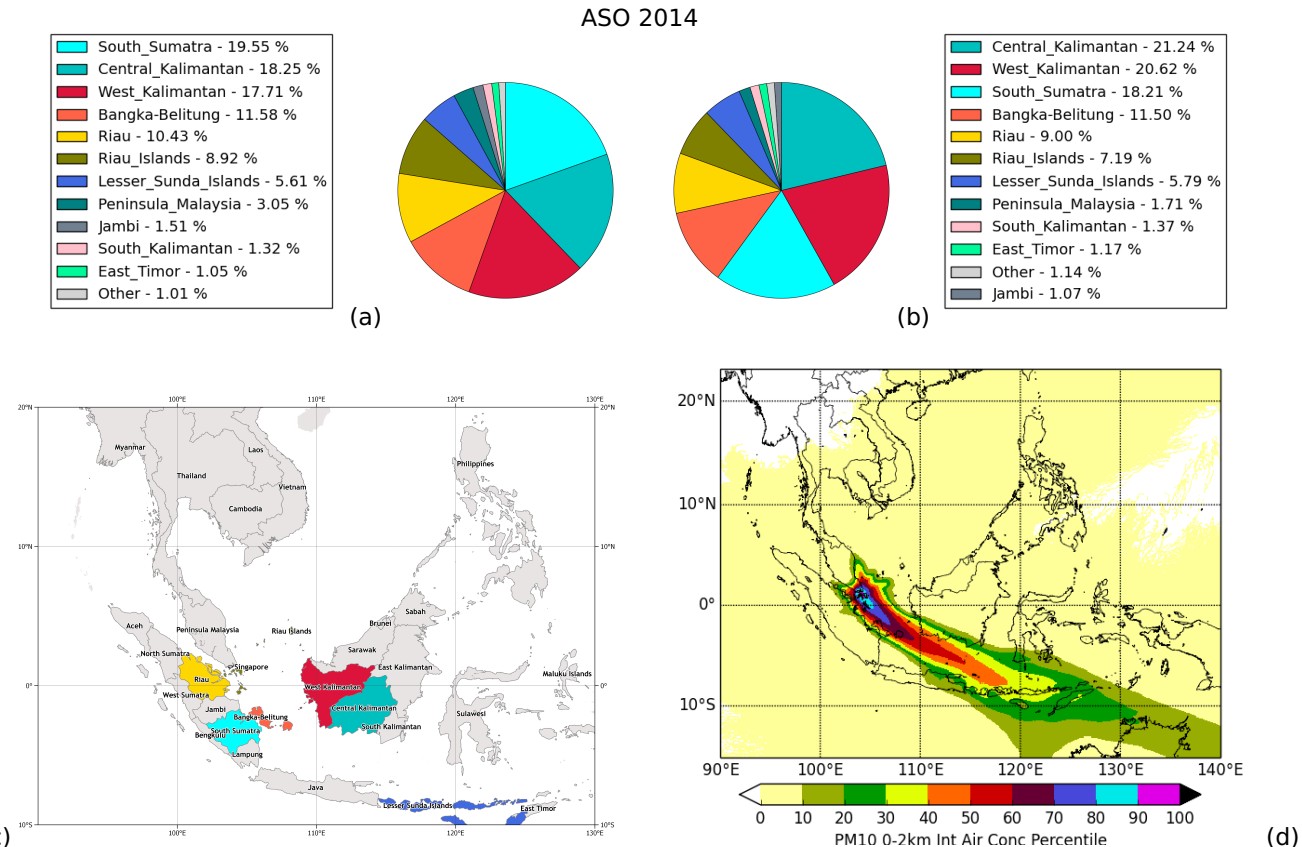

**Figure 7.** This figure shows results for $PM_{10}$ for ASO 2014: Pie charts for the western (NTU) (a) and eastern (TP) (b) monitoring stations showing major contributing source regions, (c) shows the regional map highlighting only the major contributing source regions, and the air history map (d) showing where the air arriving in Singapore originated from in ASO 2014. The 'Other' category in the pie charts is from sources which individually contribute less than 1 %.

2014, and ASO 2015) in Singapore, the air history maps show that the region of influence for Singapore generally covers the largest area during ASO when air is coming from southeasterly directions. Of the four years (2011, 2012, 2014, 2015) with haze events during ASO, 2014 saw the largest region of influence. Of the two years with events during FMA (2011 and 2014) the winds were generally from a northeasterly direction and 2014 was, again, the year influenced by the largest source

5 region. For seasons with southeasterly winds, but not during ASO, e.g., 2012 MJJ, the region of influence is relatively small compared to that of ASO. Our results, presented in Figure 3, confirm the findings of other studies such as Lee et al. (2016b) who determined the source region for Singapore to be mainly Sumatra and Borneo (i.e., Kalimantan, Sarawak, Sabah, and Brunei). Shi and Yamaguchi (2014) also saw that the biggest emitters include South Sumatra and South Kalimantan, showing that spring emissions mainly originate from Cambodia, Laos, Myanmar, Thailand, Vietnam, and on occasion Peninsular Malaysia,

10 whereas, autumn burning is seen in Central Kalimantan, Jambi, South Sumatra, West Kalimantan, and to a lesser extent Aceh

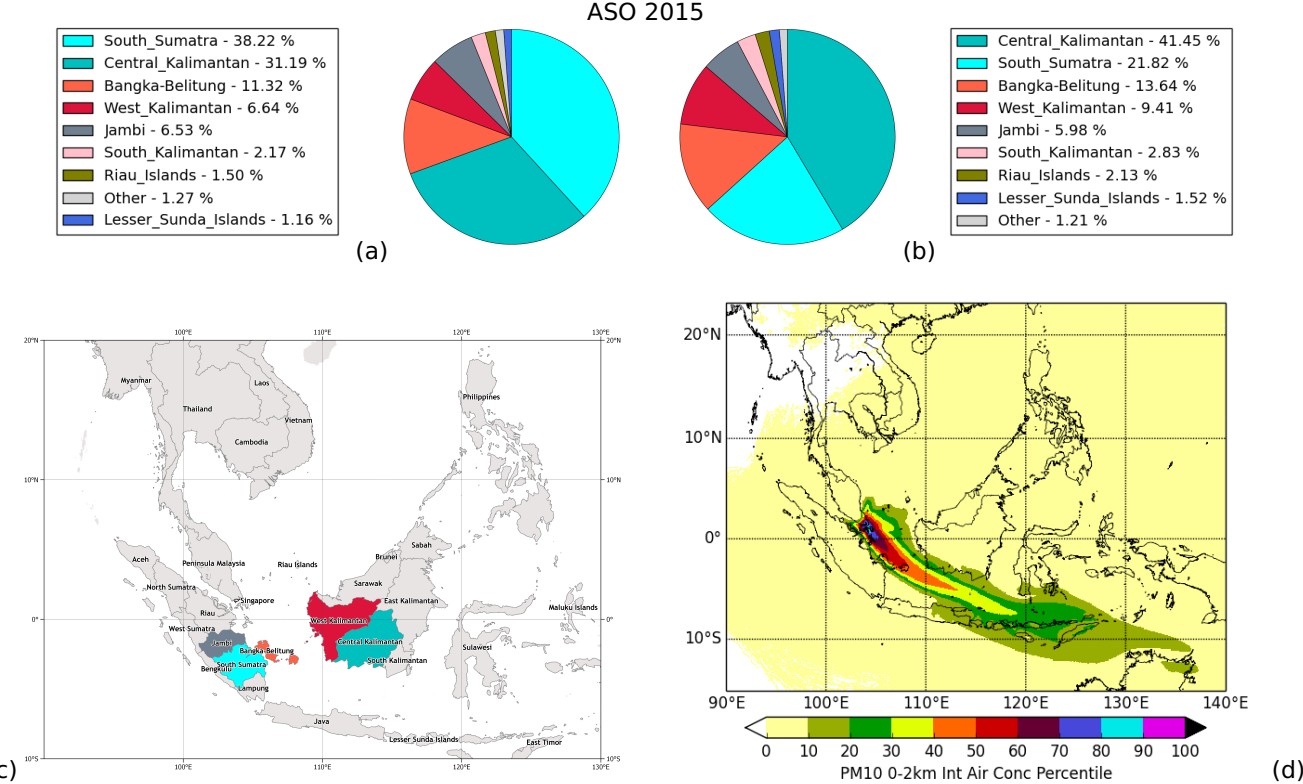

**Figure 8.** This figure shows results for $PM_{10}$ for ASO 2015: Pie charts for the western (NTU) (a) and eastern (TP) (b) monitoring stations showing major contributing source regions, (c) shows the regional map highlighting only the major contributing source regions, and the air history map (d) showing where the air arriving in Singapore originated from in ASO 2015. The 'Other' category in the pie charts is from sources which individually contribute less than 1 %.

and East Kalimantan. Emissions from Riau vary significantly throughout the years and individual months, though there are emissions from Riau in most months during most years, which is consistent with the emissions shown in Fig. 3.

## 4 Conclusions

In this study we have used the atmospheric dispersion model, NAME, to attribute $PM_{10}$ concentrations in Singapore caused
5 by biomass burning to their source region. In order to gain a deeper understanding of the causes of haze in Singapore we have compared air history maps, showing where air arriving in Singapore originates from, with modelled and observed $PM_{10}$ concentrations at two monitoring stations located at a western and an eastern location, respectively. For those two monitoring stations we have also compared the difference between relative contributions from all of the source regions.

The yearly and seasonal variations in emissions of $PM_{10}$ from biomass burning from the region are not always correlated with $PM_{10}$ concentrations in Singapore. Yet the modelled results confirm that the highest $PM_{10}$ concentrations in Singapore coincide with haze caused by biomass burning. The results show that haze in Singapore is impacted by (1) burning emissions under human influence (e.g., Fig. 3), (2) the weather through the monsoon and related winds (Fig. A2), and (3) climate, especially the variations in ENSO, which is also in line with the findings by Reid et al. (2012, 2013). In previous similar studies it has been assumed that the same emission inventory can be used for different years (Kulkarni et al., 2015; Sobhani et al., 2018), and some attribution studies even used the same meteorology when studying different years (Kim et al., 2015). Our findings demonstrate that this is not sensible for biomass burning due to the inter-annual variability of both meteorology and emissions, which can be extremely high both spatially and temporally (Kelly et al., 2018).

For the four haze events focused on here, there is variability in the correlation between the modelled and observed time series, with the best correlations seen for haze events where the emission sources are close to Singapore. As discussed by Hertwig et al. (2015), uncertainty in these results originates from the emissions and the meteorology. For the former, the uncertainties result from the fact that the emissions used here are based on one daily snapshot of FRP and IH, and though some attempts are made to resolve issues with missing fire emissions caused by the lack of transparency of clouds the data will naturally be incomplete. At the same time, hourly emissions are calculated based on this one daily snapshot adding a temporal resolution that the data does not provide, which also means that peak concentrations will not always be captured in the model simulations. The meteorology provides another significant source of uncertainty, as is usually the case in atmospheric modelling. When considering the resolution of the analysis meteorology used here and the size of Singapore it is clear that there will be unresolved features in both topography and in the meteorology and hence in the dispersion modelling. However, the differences we see between the two sites show that we are starting to capture this scale. Uncertainties in the NWP data such as elevated wind speeds and too frequent and too low intensity precipitation will disperse the pollutants further and wash out more than should be, resulting in lower modelled concentrations. These uncertainties naturally have a larger impact over longer travel distances, which is reflected in our statistics. It should also be kept in mind that the observations are measuring all $PM_{10}$ and we are only modelling primary $PM_{10}$ emissions from biomass burning. Other sources of $PM_{10}$ include sea salt, dust, secondary organic aerosol, emissions from industry, local and transboundary road traffic, as well as domestic heating, not all of which are constant throughout the year. Some of the varying difference between observed and modelled time series is also likely to be due to these many other sources of $PM_{10}$ in Singapore. However, in spite of these uncertainties our results show that we are able to model dispersion of particulate matter from biomass burning in Southeast Asia and the resulting haze in Singapore with reasonable confidence.

Emissions from many regions contribute to the concentrations of $PM_{10}$ in Singapore. The biggest contributors for the period 2010 - 2015 are Riau, Peninsular Malaysia, and South Sumatra, with smaller yet significant contributions from Jambi, Cambodia, Bangka-Belitung, Riau Islands, Central Kalimantan, and the Philippines. As Riau and Peninsular Malaysia are the nearest neighbours to Singapore and given the local wind pattern this could be expected. Looking at emissions during ASO for the four years with the most variation across the island (2011, 2012, 2014, and 2015), the largest emissions were seen from

Central Kalimantan, South Sumatra, Jambi, and also West Kalimantan. For events during FMA Cambodia, East Kalimantan, Myanmar, Thailand, and Vietnam showed larger emissions.

We investigated the spatial variation of haze across Singapore and found that variation in major contributing source regions across Singapore is dependent on distance to source regions: generally a shorter distance to the source region will mean less variation in the major contributing source region(s). We have also studied the seasonal variation by looking at four recent events occurring during different seasons and saw that air arriving from a larger geographical area often brings more variation in major contributing source regions. $PM_{10}$ concentrations at the two monitoring stations vary significantly in time, both in the observed and modelled time series; from the modelled data it is possible to attribute the major contributing source regions. These show that for the two haze events not occurring during the ASO haze season, the sources are dominated by the same source region at both sites, though a different site for the two events. For the two ASO haze events the major contributing source regions at the two monitoring sites are mainly the same but their relative contribution differ significantly. These variations are also correlated with the distance to the source regions and the season of the haze events.

The NAME model is able to provide insight into variations in major contributing source regions at a relatively smaller scale than has been done in previous studies due to its tracking capabilities and the Lagrangian nature of the model. Although the results struggle to capture the magnitude of the haze from burning further from Singapore, due to errors and uncertainties in the GFAS data and the meteorological input, they show the potential for gaining a better understanding by using higher spatial resolution. This work is a first step towards high resolution air quality forecasting for Singapore. Whilst a chemical transport model would be expected to fully capture anthropogenic and secondary particulate contributions, the inability of this study to capture the magnitude of the biomass burning concentrations shows that there is a bigger issue with emissions and potentially also modelled meteorology. Prior to investing in a full chemical transport model it is important to understand these individual components in the simulation. This work contributes towards a better understanding of the biomass burning and air quality in the region and shows that biomass burning emissions from many different source regions across Southeast Asia can reach Singapore. Accurately capturing these is essential for future air quality modelling.

In conclusion, we saw that haze events occur during seasons with both small and large regions of influence, however, most often during ASO, coinciding with a larger region of influence and often when higher emissions/increased burning occurs, resulting in variation in relative contributions from major contributing source regions across Singapore. The results emphasise the inter-annual variation between haze events and major contributing source regions, and show that Peninsular Malaysia is a dominant source of particulate matter from biomass burning for the maritime continent off-season burning impact on Singapore (Figure A4). For haze to occur in Singapore, burning is required, but so is dry weather and wind in the "right" direction. Haze comes from burning across Southeast Asia, making it a transboundary issue for the whole region. Considering that the distance from, e.g., Kalimantan to Singapore is over 500 km, this study emphasises the long-range nature of the problem.

As an extension of the current study it would be interesting to gain insight into the seasonality and the relative magnitude of $PM_{10}$ from other contributors such as industry, traffic, and domestic heating in Singapore. Further, as it is known that biomass burning varies on sub-daily timescales (Reid et al., 2013), and this study has used daily GFAS FRP and IH (Kaiser et al., 2012) for source calculation, in the future it would be interesting to study the impact of sources based on higher than

daily resolution. One could also use post fire inventories based on burnt area or conduct an inversion study, running NAME backwards from detection sites to estimate the emissions in certain areas corresponding to concentrations observed in Singapore and other locations in Southeast Asia. These results could also be compared to inventories based on satellite observations to help quantify how much burning is missing in such inventories.

5 *Code and data availability.* The NAME model and data are available by request to the Met Office, GFAS data available through the Copernicus Atmospheric Monitoring Service (CAMS).

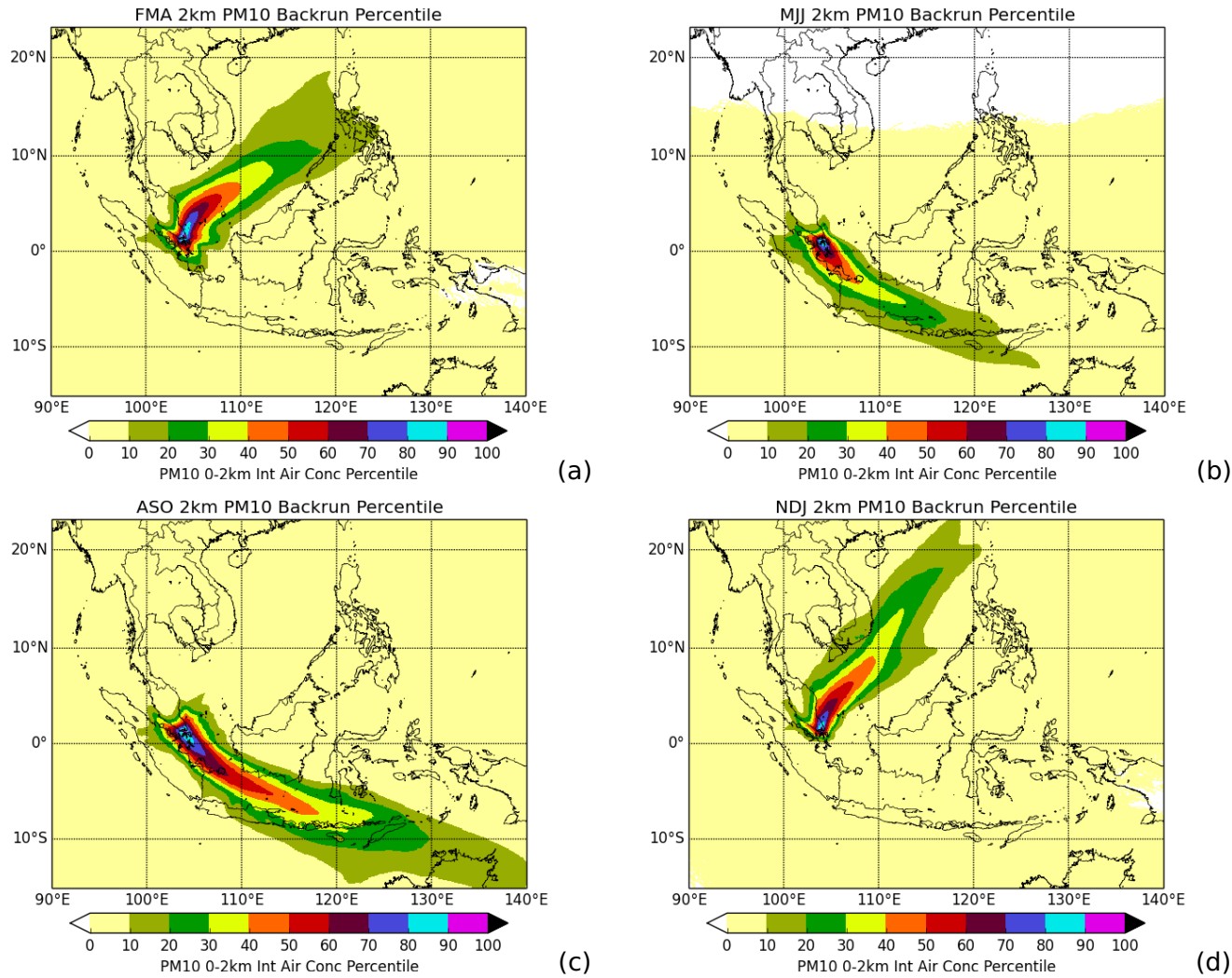

**Figure A1.** Air history maps for each of the four seasons (a) FMA, (b) MJJ, (c) ASO, and (d) NDJ, averaged over the years 2010 to 2015, showing where air arriving in Singapore during each season originated from. The backruns shown were conducted from a receptor site in central Singapore.

*Author contributions.* ABH performed most of the attribution model simulations, the data analysis and wrote the paper in collaboration with CW. WMC performed the simulations for and the visualisation of the air history maps, EK performed additional attribution model simulations, and assisted with visualisation and calculation of error metrics, BNC, CG, MCH, and SYL helped design the model setup and provided feedback on the manuscript.

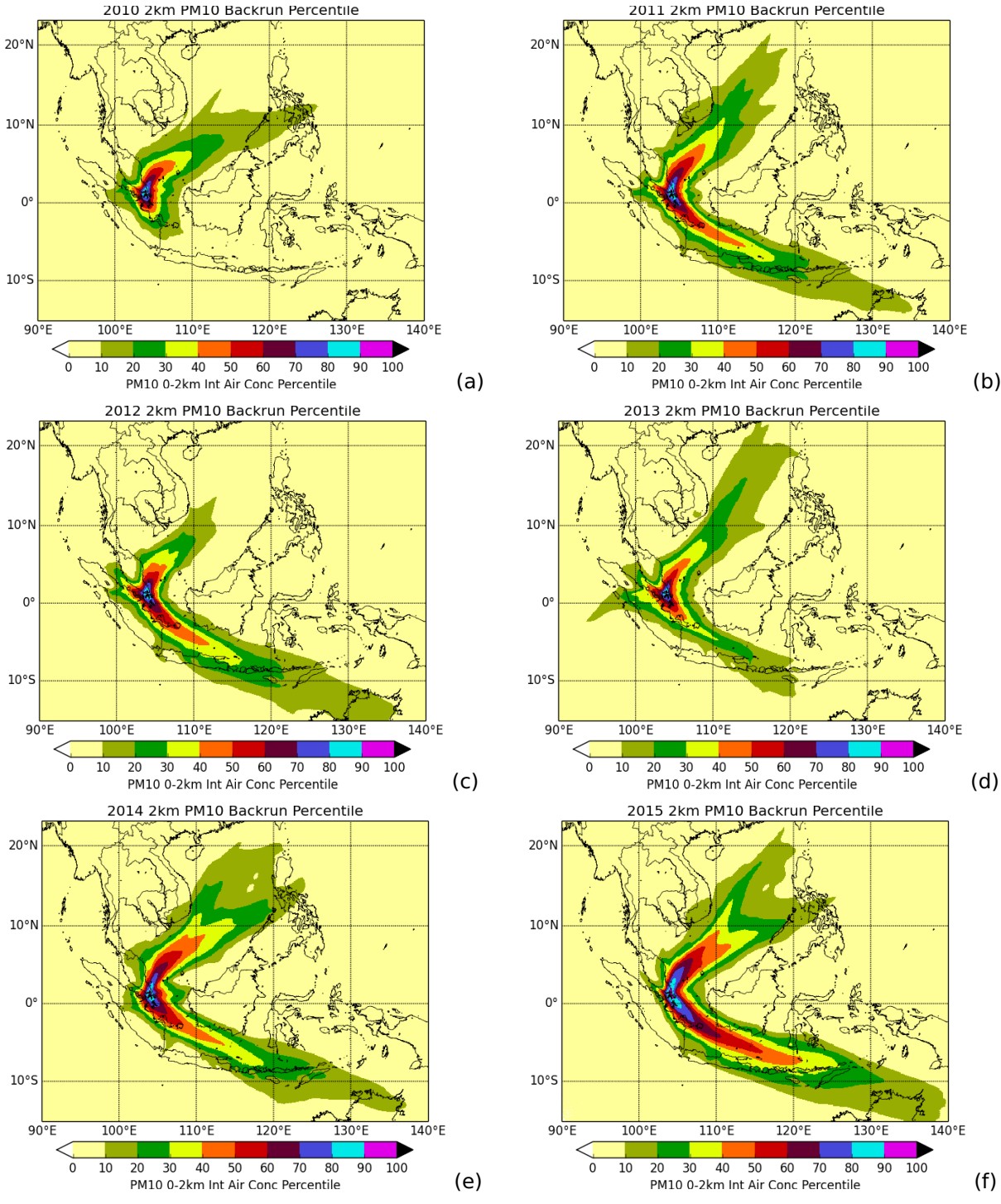

**Figure A2.** Air history maps for the years 2010 to 2015, showing where air arriving in Singapore during each year originated from. The backruns shown were conducted from a receptor site in central Singapore.

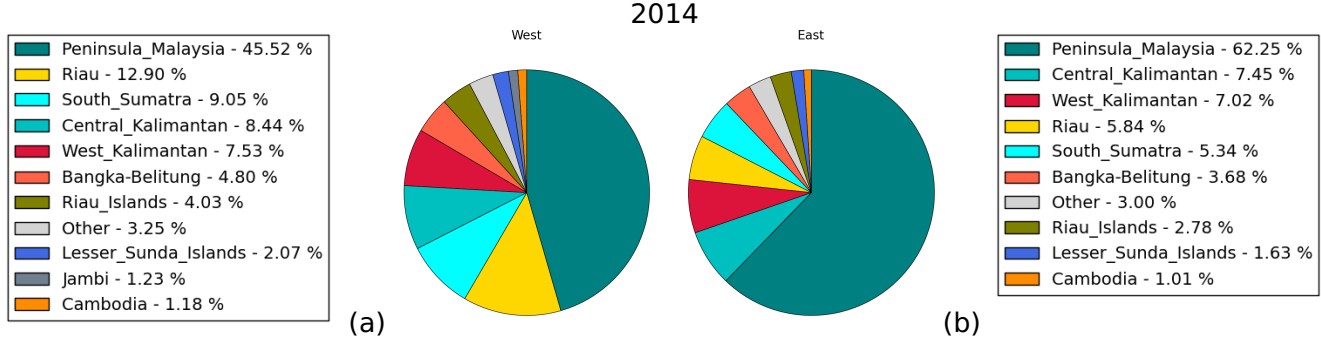

**Figure A3.** Attribution results for PM$_{10}$ for 2014: major contributing source regions for the western (NTU) (left) and eastern (TP) (right) monitoring stations.

*Competing interests.* No competing interests are present

*Acknowledgements.* We would like to acknowledge the National Environment Agency, Singapore for supplying us with PM$_{10}$ observations in this study. We are thankful for the support from the CAMS GFAS developers in using the GFAS v1.2 emissions data. We would like to thank the Centre for Remote Imaging, Sensing and Processing (CRISP) at the National University of Singapore for providing the 250 m resolution 2015 land cover map for Southeast Asia. We are grateful to the reviewers and co-editor for their valuable and challenging comments, which have significantly improved this paper.

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

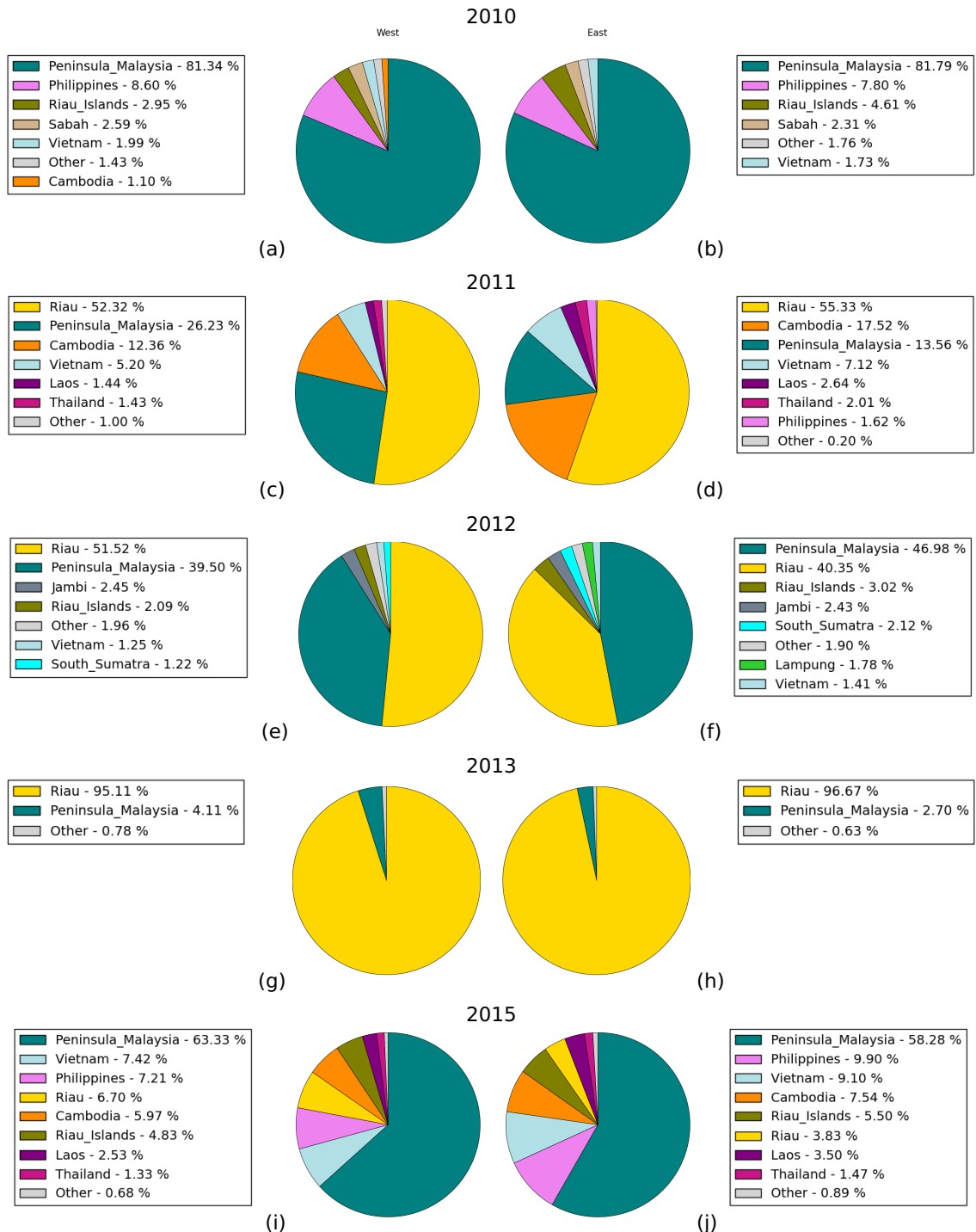

**Figure A4.** Attribution results for $PM_{10}$ for FMA for years 2010 - 2013 and 2015: major contributing source regions for the western (NTU) (left) and eastern (TP) (right) monitoring stations. (For 2014 FMA, see Fig. 6.)