# Peer review of "Haze in Singapore - Source Attribution of Biomass Burning $PM_{10}$ from Southeast Asia"

_Atmospheric Chemistry and Physics, 2018_

## Referee Comment (RC1) · Anonymous Referee #1 · 22 Jun 2018

**Review of Hansen et al. for Atmospheric Chemistry and Physics**

*General Comments*

In "Haze in Singapore - Source Attribution of Biomass Burning from Southeast Asia", Hansen et al. use the NAME Lagrangian model to apportion modeled concentrations of a tracer from biomass burning emissions in Singapore to source locations. The modeled concentrations depended on meteorology, dry and wet deposition, and daily average emissions rates for biomass burning from the Global Fire Assimilation System (GFAS) v. 1.2. The tracer concentrations were compared with observed concentrations of particulate matter less than 10 μm in diameter ($PM_{10}$) at two sites on either side of Singapore. Other contributions to $PM_{10}$ and any chemical transformation of the emissions from the fires were neglected. The literature review is helpful. NAME and the underlying meteorology have been used to study similar questions for specific episodes in the region.

The major difficulty in this work seems to be representing $PM_{10}$ concentrations in Singapore except when biomass burning occurs very close by the sites (e.g., Riau or the Malaysian Peninsula) (Fig. 4). The two times this occurred during the modeled period are labeled as "Atypical haze" (Sect. 3.1). As the authors note, the $PM_{10}$ concentrations in Singapore, especially in the other "haze seasons", are likely from diverse sources; however, the time series (Fig. 4) and statistical analysis of the agreement between the model and observations (Tables 1, 2) do not support the notion that "Southeast monsoon season haze" (Sect. 3.2) is represented accurately enough by the model to claim attribution of these sources to biomass burning regions. The difficulty may result from the lines of source regions not being stacked in the time series plot (Fig. 4), but the correlation statistics are fairly poor for periods other than 2013 MJJ, which supports the interpretation of the time series that the model is not representing concentrations well. One reason may be that the $PM_{10}$ from biomass burning from regions not adjacent to Singapore is not well represented by the modeled processes. Another reason may be, as noted by the authors, that much of the haze comes from biomass burning sources not detected by the model. Finally, it may be that the $PM_{10}$ is not from biomass burning. The first two causes would indicate that this modeling framework is not appropriate for attribution of biomass burning contributions to $PM_{10}$ concentrations in Singapore. The final potential explanation could be shown by more sophisticated analysis of the background $PM_{10}$ concentrations rather than using a fixed value of 25 μg/m$^3$; if this explanation does not hold, it seems unreasonable to claim that these attributions are appropriate for episodes other than those in which the Riau and Malaysian Peninsula contributions dominate.

Given the difficulty of interpreting the results, the weakness of the Discussion and Conclusions sections seem reasonable. Specifically, the Conclusions seem to be very repetitive of the Discussion. Accordingly, I recommend this manuscript for publication in Atmospheric Chemistry and Physics once this major issue has been addressed. Additionally, specific comments have been included below that should also be considered in the revisions of the manuscript.

*Specific Comments*

| Line | Comment |
|---|---|
| p. 1, l. 8-13 | The meaning of this interpretation of the results is unclear especially in the phrases "several and varying source regions", "atypical haze episodes … characterized by atypical weather conditions", and "different set of five regions dominate". Please refine. |

| p. 1, l. 14 | "climate" seems inappropriate when only 5 years have been considered. |
|---|---|
| p. 2, l. 3 | "Though the popular press often attribute" - grammatical error. Also, scientific literature has supported similar conclusions (Kim et al., 2015). |
| p. 2, l. 17-9 | Please insert a comma as "approach, and source" or divide these two thoughts into separate sentences. What type of source apportionment was applied by Lee et al. (2017) and Engling et al. (2014)? Please be more specific. |
| p. 2, l. 20-2 | Run-on sentence. Please correct here and throughout the manuscript (e.g., p.3, l.8-12, etc.) |
| p. 2, l. 26-7 | Poor sentence construction leads to a lack of clarity. |
| p. 3, l. 13-4 | "related haze events are linked" is redundant. |
| p. 3, l. 23-4 | A strong case for using dispersion modeling has not been made in the Introduction. |
| p. 4, l. 31 | Please provide a reference or equation for the MNMB. |
| p. 4, l. 35 | Please clarify here, as is stated later, that the emissions are emitted over a 24-hour period at the rate of 1 g/s. Please state how the same emissions rate results in different total emissions from a single fire (if it does). |
| p. 5, l. 7 | Please note the limitations of comparing a tracer with $PM_{10}$. |
| p. 6, l. 4-9 | Please include an equation for the calculation described here. It is not clear how double counting in time is avoided given this description. |
| p. 16, l. 1-2 | Was it expected that the "atypical and different meteorological conditions" would cause variation of the source regions when those were dominated by nearby fires? It seems unlikely, so the sentence is unexpected. |
| p. 19, l. 6 | "Similarly to" should be "Similar to". |
| p. 19, l. 15-7 | Two statements contradict one another. The atypical haze events are said to have little variation between monitors, but then FMA 2014, one of the atypical haze events, is noted as having the largest difference in the next sentence. Please restate. |
| p. 20, l. 3 | Please eliminate the use of contractions here ("won't") and elsewhere. |
| p. 20, l. 5-6 | No effort was made to show data that support this conclusion. Please show that it is true as suggested in the General Comments or remove the sentence. |
| p. 20, l. 9 | The grammar and sentence construction in the Conclusions section of the document require careful revision. |

Table 1     Please include the meaning of the abbreviations for the statistical correlations in the caption.

Figure 2    The caption states "air history" but the colorbar label indicates "Air Conc Percentile", which seems to include information about emissions or concentrations rather than simply where the air has been. Please clarify.

Figure 4    Please specify the meaning of the colored values. Are the lines for the source regions stacked? If not, they should be in order to show how they contribute to the total observed concentration.

Figures 6-8 The design of these figures is nice. It is not clear why only some of the contributing regions are colored. Please be consistent between the "other", which are grey in the legend, and the grey countries in the map. Also, please order the pie chart wedges in the same order as the legend names. The colors are too similar to be able to distinguish when the pie chart is not in the same order as the regions in the legend.

Kim, P. S., Jacob, D. J., Mickley, L. J., Koplitz, S. N., Marlier, M. E., DeFries, R. S., Myers, S. S., Chew, B. and Mao, Y. H.: Sensitivity of population smoke exposure to fire locations in Equatorial Asia, Atmos Environ, 102, 11–17, doi:10.1016/j.atmosenv.2014.09.045 , 2015.

---

## Referee Comment (RC2) · Anonymous Referee #2 · 6 Jul 2018

**Review of "Haze in Singapore - Source Attribution of Biomass Burning from Southeast Asia" by Hansen et al.**

This study investigates the sources of biomass burning from Southeast Asia for 6 years (2010-2015) using the Lagrangian dispersion model NAME. The tracer concentrations were evaluated using observations at two sites in Singapore. The authors also discussed the seasonal variations of emissions sources to Singapore. The topic of this paper is very interesting and important and I really appreciate the seasonal focus in this study. However, the conclusion and discussion are very confusing and repetitive. The paper lacks a coherent flow and the method section lacks significant information and clarity. I would only recommend this manuscript for publications in ACP only after substantial modifications to the manuscripts and figures. I also suggest the authors make higher quality figures for publication in ACP (Figs1, 4, ...). In general, it is difficult to interpret the results and the discussion is weak.

1- **Meteorology:**

Dispersion models are highly sensitive to their meteorological inputs. However, there is no analysis of the metrological values fed into the model. First, there is not clear statistical analysis or comparisons between modeled meteorology from UM and observations in that region. Without first evaluating the meteorological input, we cannot draw any conclusion from the Lagrangian models. For example, slight errors in modeled wind speed (and direction) and observations, makes the originating source region of the tracer very different.

Second, P4:L1-2 mentions that the metrology runs were different (resolution and settings?) for different years. Based on previous studies, the modeled meteorological values (especially wind speed) are sensitive to the model resolution. Considering that in this manuscript, the authors compared different years with each other, I strongly recommend use of consistent settings and resolutions for the NWP runs. Else the inter-annual difference between the sources of biomass burning can easily be attributed to the difference between meteorological model differences.

2- In general, there are large discrepancies between modeled and observed PM10 for all years and both stations. The authors assumed a constant 25 ug/m3 background concentration for all year. However, the emissions from various sectors (especially residential) have high seasonal variability (see Sobhani et al. 2018). Considering the same background concentration (meaning the same contribution of other sectors to your PM10) for all seasons may introduce large errors to your analysis.

**Specific Comments:**

**Introduction:**

1- This part lacking significant discussions and references. For example, the reference this part of P2, L2 is missing, "it is not caused by activities within Singapore, rather it is a transboundary problem caused by biomass burning across the wider region." Maybe adding sample studies.

2- P2, L13: I would recommend adding more discussions here. I suggest citing and/or describing some source attribution studies with the focus on other regions or bigger domains using different methods (Eulerian, Lagrangian, Observation analysis). I am not sure why the very few studies in the next lines are cited. Few examples for source appointment in different region of the world (with Eulerian methods) are: (Ikeda et al., 2017; Sobhani et al., 2018; Wang et al., 2011; Yang et al., 2017), With both Eulerian and Lagrangian methods:(Kulkarni et al., 2015) Observation Analysis + Lagrangian: (Winiger et al., 2017).

3- P2, L 22: Any reference for this sentence or is it the result of the study? If it is this result of the study please mention so.

4- P2, L34 and P3, L3: Please add a reference for each sentence.

5- In general, I suggest restructuring this section a bit for cohesiveness by moving few first sentences of the 5th paragraph (P2, L28) in introduction before the 4th paragraph (P2, L20). It is not clear if some the sentences in the 4th paragraph are result of this study or previous studies.

6- It is not obvious why the focus of study is Singapore. Can you please add why the focus of this study Singapore?

**Methods:**

1- This section lacks a lot of details. Can you please add some information and a paragraph describing the NAME model? How are they numerically represented in the model? What kind of aerosol processes are accounted for? Are there any know biases? Why have you used this model for this study?

2- Can you please add some information and more description on the modeling setup for this study instead of just citing Hertwig et al. 2015. How are the wet and dry deposition processes calculated in the model?

3- Also, can you please describe your meteorological model (UM) and why this model is used to drive NAME?

4- Significant lack of clarity and explanations regarding observations: It is not clear at all where the locations of the observation sites are (maybe add them to all maps and include lat lon of the measurement sites?)? The authors should add more information on the method of measurements in those locations. Also, tit is not clear where does these measurements come from (paper?, organization?)? Also, is this data available for public if so please include the link to the data either here or in the code and data availability section of the paper (or both).

5- Air history map?? Do you mean PM10 or all aerosols (air?) lumped together? What chemical species and aerosols are considered in air history maps? Or is it only PM10 or tracer? This term is very confusing.

**Results:**

1- The assumption of 25 ug/m3 for both stations is problematic. Could it be because the background value from another source is higher in the western station??Also emissions from other important sectors are not accounted for which might cause the difference between the stations.

2- P14, L13: Can you add some figure (maybe to SM) to show the meteorology difference for 2013 and other years. In general, this sentence is vague. What do you mean by 2013 is a unique year in terms of meteorology and burning?

3- Are peaks concurrent with biomass burning incidents? Several other factors influence the peaks. For example, high residential emissions in winter in South East Asia can be attributed to the peaks.

4- There is a large redundancy between results (section 3) and discussions (section 4). I suggest merging section 4 into section 3 and conclusions.

5- P 15, L30: It seems like the model did not capture the observation contrary to the claim.

6- P16, L10: The model significantly underestimates the peaks (30/125) Please explain why?

**Figures:**

1- Does Figure 1 show the entire domain? It seems smaller that the domain mentioned in the method section. Please correct the figure include all the domain in this figure.

2- Please add the location of Singapore to Figure1 and all other spatial figures. It is hard for someone who does not know the regions geography to find Singapore in each figure. Based on the captions description the reader might think Singapore is located in the Riau Islands.

3- Figure 2: Wrong caption. The second line of caption of this figure is not related to manuscript. Central receptor sites???? Inland and coastal sites? Are these sites discussed in this manuscript??

4- Figure 2: Please correct the label title.

5- Figure 2: Can you please add more discussions about this figure to the paper? It is confusing what these figures show.

6- Figure 2: It is very nice that you included figure A2 (Figure 2 for all years to the discussion). I suggest also adding similar plots for each season. (each season averaged over the years). The season specific "Air history maps" would make it easier to understand the transport pathway in different seasons as discussed in P6, L10.

7- Figure 3: This is a good plot; however, it is difficult to compare different years because of the different scales. Also, the y axix label denote T as the unit for monthly emission which is different from the caption.

8- Figure 4: This figure is very hard to read and should be modified before publications. First, it seems like hourly observations are plotted against (daily averages of model??). It is very

hard to distinguish any modeled data points. Please make different plots for this figure. One way is showing monthly averages for both model and observation similar to Figure 3. Or time series of the daily observations as points overlaid on top of the modeled output. I recommend area chart for modeled value. Please include sum multiple region as "the other regions" multiple of the regions in this plot. Please only include important regions with visible high impacts. Very few of the 28 regions are visible in this plot. Maybe another scale (e.g. a log scale) is better for the purpose of this plot. Please use the same scale for all years and denote the events discuss on these plots.

The scale for all the years vary significantly. I would suggest having all PM 10 for all years on the same scale 0-700. Quick look at the plots, one might think there are higher pm 10 concentrations in 2010 compared to 2013 or 2015.

9- Figures 5-8: Please add a title with the name of event to the plot. Please add the stations (locations of NTU and TP) and denote Singapore on the plots.

10- Figure A2: What are the colored squares overlaid on the plots? Please put the figure in order that is mentioned in the paper.

**Minor Comments and Technical Corrections:**

7- P2, L3: Fig 1 is technically not related to this sentence.

8- P2, L3: I would recommend adding reference for the second part of this sentence. (the reference for transboundary problem…)

9- P2, L25: This sentence is very vague. Two events in each of June 2014-2015 and FMA 2014-2015. Or one event in June (2014 or 2015?) and one in FMA (2015?). Also, why using FMA vs June. I would recommend either using month or season. Can you use the season instead of June for consistency?

10- P2, L 33: Can you please elaborate what you mean by north-east monsoon and south-west monsoon seasons?

11- P3, L 9: FMA acronym were explained last page and redundant here.

12- P3, L30: What does NAME stands for?

13- P3, L32: I recommend adding a figure to SM with the modeling domain. It seems like figure 1 does not show the complete modeling domain. (not extended 14 S or 23 N)

14- P4, L1-2: What do you mean? Is it different meteorological setup for each year??? Is the resolution of NWP runs different from 17 km to 40 km??

15- P4, L 14: Please add what GFED stands for.

16- P4, L 16: Redundant, very similar sentence in the above paragraph L10….

17- P4, L24-25: Can you please point to the pie charts?

18- P4- L30-35: Can you please explain why did you use these metrics instead of other metrics like R2 or RMSE and many others? I suggest adding few more metrics to the tables 1 and 2 (R2 and RMSE). Please provide references or descriptions of the metrics used.

19- P4, L35: I strongly suggest using daily averages instead of hourly averages.

20- P5, L7: Please add what NWP stands for.

21- P5-6: Air history vs air conc. percentile? Please clarify?

22- P7, L20: I would highly suggest including emission maps for each year.

23- P7, L29: What is the reference for this sentence?

24- P7, L32: Why did you assume constant 25 ug/m3 for background concentration? Is there any reference for that?

25- P8, L3-5: I suggest including the values for clarity and readability.

26- P9, L4: Please add the name of the western monitoring station here and throughout the manuscript

27- P14, L15-16: A sentence without a paragraph. remove the unnecessary line break.

28- P14, L 16-18: This sentence is very confusing. Please rephrase it.

29- P14: In general, adding the locations of the sites to the maps would make reading the paper much easier.

30- P14, L24: It is not obvious if the maximum observed and modeled are concurrent or the values indicate maximum observation and maximum modeled value occurring at different times?

31- P15, L25: Would you discuss FMA 2014 or February 2014 only. Earlier in the text you mentioned June 2013 and February 2014 as the haze events but discuss FMA 2014 as the haze event here. In general, there is a lack of consistency between using months and seasons.

32- P15, L30: Can you explain the reason why concentrations at TP is double of NTU?

33- P16, L1-2: Very unclear and vague sentence. Different meteorology for events or between the monitoring stations? The sentence implies that in spite of the clear dominance of one source region, there is a little variation in the source regions across the monitoring stations???

34- P16, L3: I suggest adding ASO to the title of this section. The inconsistency between using southwest monsoon haze and ASO makes it confusing.

35- P16, L4: Please remove the extra line break.

36- In general, not clear when the events are. I would highly suggest making a table including all the events discussed (and their corresponding figure) and also denoting each event on the time series plot.

37- P 19, L1-3: This sentence is extremely confusing. For ASO or for the seasons with the most significant haze events including MJJ, FMA, and ASO?

38- Please check the sentence constructions of discussions and conclusions sections carefully.

39- P19- 20: The discussions seems like an extended conclusion section and there is a lot of redundancy between results (section3), discussions (section 4) and conclusions (section 5), which decrease the readability of paper.

40- P20, L30: Please restate this sentence. It is very confusing.

41- Code and data availability: Please include the link to the observations used for this study.

**References**

Ikeda, K., Tanimoto, H., Sugita, T., Akiyoshi, H., Kanaya, Y., Zhu, C. and Taketani, F.: Tagged tracer simulations of black carbon in the Arctic: transport, source contributions, and budget, Atmos. Chem. Phys., 17(17), 10515–10533, doi:10.5194/acp-17-10515-2017, 2017.

Kulkarni, S., Sobhani, N., Miller-Schulze, J. P., Shafer, M. M., Schauer, J. J., Solomon, P. A., Saide, P. E., Spak, S. N., Cheng, Y. F., Denier van der Gon, H. A. C., Lu, Z., Streets, D. G., Janssens-Maenhout, G., Wiedinmyer, C., Lantz, J., Artamonova, M., Chen, B., Imashev, S., Sverdlik, L., Deminter, J. T., Adhikary, B., D'Allura, A., Wei, C. and Carmichael, G. R.: Source sector and region contributions to BC and $PM_{2.5}$ in Central Asia, Atmos. Chem. Phys., 15(4), 1683–1705, doi:10.5194/acp-15-1683-2015, 2015.

Sobhani, N., Kulkarni, S. and Carmichael, G. R.: Source Sector and Region Contributions to Black Carbon and PM2.5 in the Arctic, Atmos. Chem. Phys. Discuss., 2018, 1–43, doi:10.5194/acp-2018-65, 2018.

Wang, Q., Jacob, D. J., Fisher, J. a., Mao, J., Leibensperger, E. M., Carouge, C. C., Le Sager, P., Kondo, Y., Jimenez, J. L., Cubison, M. J. and Doherty, S. J.: Sources of carbonaceous aerosols and deposited black carbon in the Arctic in winter-spring: implications for

radiative forcing, Atmos. Chem. Phys., 11(23), 12453–12473, doi:10.5194/acp-11-12453-2011, 2011.

Winiger, P., Andersson, A., Eckhardt, S., Stohl, A., Semiletov, I. P., Dudarev, O. V, Charkin, A., Shakhova, N., Klimont, Z., Heyes, C. and Gustafsson, Ö.: Siberian Arctic black carbon sources constrained by model and observation, Proc. Natl. Acad. Sci., 114(7), E1054 LP-E1061 [online] Available from: http://www.pnas.org/content/114/7/E1054.abstract, 2017.

Yang, Y., Wang, H., Smith, S. J., Ma, P.-L. and Rasch, P. J.: Source attribution of black carbon and its direct radiative forcing in China, Atmos. Chem. Phys., 17(6), 4319–4336, doi:10.5194/acp-17-4319-2017, 2017.

---

## Author Comment (AC1) · 21 Sep 2018

Review 1: General Comments

The major difficulty in this work seems to be representing PM10 concentrations in Singapore except when biomass burning occurs very close by the sites (e.g., Riau or the Malaysian Peninsula) (Fig. 4). The two times this occurred during the modeled period are labeled as "Atypical haze" (Sect. 3.1). As the authors note, the PM10 concentrations in Singapore, especially in the other "haze seasons", are likely from diverse sources; however, the time series (Fig. 4) and statistical analysis of the agreement between the model and observations (Tables 1, 2) do not support the notion that "Southeast monsoon season haze" (Sect. 3.2) is represented accurately enough by the

model to claim attribution of these sources to biomass burning regions. The difficulty may result from the lines of source regions not being stacked in the time series plot (Fig. 4), but the correlation statistics are fairly poor for periods other than 2013 MJJ, which supports the interpretation of the time series that the model is not representing concentrations well. One reason may be that the PM10 from biomass burning from regions not adjacent to Singapore is not well represented by the modeled processes. Another reason may be, as noted by the authors, that much of the haze comes from biomass burning sources not detected by the model. Finally, it may be that the PM10 is not from biomass burning. The first two causes would indicate that this modeling framework is not appropriate for attribution of biomass burning contributions to PM10 concentrations in Singapore. The final potential explanation could be shown by more sophisticated analysis of the background PM10 concentrations rather than using a fixed value of 25 $\mu$g/m3; if this explanation does not hold, it seems unreasonable to claim that these attributions are appropriate for episodes other than those in which the Riau and Malaysian Peninsula contributions dominate. Given the difficulty of interpreting the results, the weakness of the Discussion and Conclusions sections seem reasonable. Specifically, the Conclusions seem to be very repetitive of the Discussion. Accordingly, I recommend this manuscript for publication in Atmospheric Chemistry and Physics once this major issue has been addressed. Additionally, specific comments have been included below that should also be considered in the revisions of the manuscript.

We thank the reviewers for their constructive feedback and hope that our subsequent changes to the manuscript has improved the content and readability of the paper. We acknowledge that our results are not as good as would be desired, but this work now acts as a stepping stone to improving the understanding of haze in the region and how to model haze at high temporal and spatial resolution. And we feel it is worth sharing our findings with the community. We have also highlighted uncertainties in the work, including the limitations to emissions and focussed on the four events with the best correlation between model and observations. We have rewritten and restructured the Introduction and removed the Discussions section to make the Results and

Conclusions clearer and more coherent. The figures have been updated following the reviewers' suggestions.

Specific Comments Line Comment p. 1, l. 8-13 The meaning of this interpretation of the results is unclear especially in the phrases "several and varying source regions", "atypical haze episodes . . . characterized by atypical weather conditions", and "different set of five regions dominate". Please refine. - The paragraph has been changed to: As should be expected, the relatively stronger Southeast monsoonal winds that coincide with increased biomass burning activities in the Maritime Continent create the main haze season from August to October (ASO), which brings particulate matter from varying source regions to Singapore. Five regions dominate as the source of pollution during recent haze seasons. In contrast, off-season haze episodes in Singapore are characterised by unusual weather conditions, ideal for biomass burning, and emissions dominated by a single source region (for each event). The two most recent off-season haze events in mid-2013 and early-2014 have different source regions, which differ to the major contributing source regions for the haze season.

p. 1, l. 14 "climate" seems inappropriate when only 5 years have been considered. – The study covers 6 years, but more pertinently also considers the impact of El Nino and other non-weather timescale phenomena. The use of the word "climate" in this context seems appropriate is supported by other literature.

p. 2, l. 3 "Though the popular press often attribute" - grammatical error. Also, scientific literature has supported similar conclusions (Kim et al., 2015). – Reference added as suggested: "Scientific studies such as Kim et al 2015 as well as the popular press often attribute peatland destruction and related haze in the region to Indonesia (Reid et al 2013), however, the haze cannot be attributed to only one region or country alone."

p. 2, l. 17-9 Please insert a comma as "approach, and source" or divide these two thoughts into separate sentences. What type of source apportionment was applied by Lee et al. (2017) and Engling et al. (2014)? Please be more specific. – more details

have been added to the text: "Several previous studies have looked at attributing air pollution for different regions. Source attribution can be performed both through modelling and by looking at observations of air pollution in detail. For example, Heimann et al. (2015) carried out a source attribution study of UK air pollution using observations to distinguish between local and regional emissions, whereas Redington et al. (2016) estimated the sources of annual emissions of particulate matter from the UK and the EU by using the NAME model to look at threshold exceedences and episodes. Attribution studies have been performed using Eulerian models such as GEOS-chem, CMAQ, and WRF-STEM to study both Asia and the Arctic (Ikeda et al., 2017; Kim et al., 2015; Sobhani et al., 2018; Yang et al., 2017; Matsui et al., 2013) sometimes in combination with flight campaigns (Wang et al., 2011) to better constrain the emissions. Lagrangian models have also been used in combination with observations by Winiger et al. (2017). Combinations of Eulerian and Lagrangian models (Kulkarni et al., 2015) and Eulerian models and observations (Lee et al., 2017b) have been used to assess whether low visibility days were caused by fossil fuel combustion, biomass burning or a combination of the two. In Southeast Asia, Reddington et al. (2014) used an Eulerian model to study haze and estimated emissions through a bottom up approach. Source apportionment for studies of biomass burning related degradation of air quality and visibility in Southeast Asia has also been applied by Lee et al. (2017a) who used the WRF model to study the sensitivity of the results to different met data and emission inventories and Engling et al. (2014), who used observations and a chemical mass balance receptor model to compare the chemical composition of total suspended particulate matter on haze and non-haze days during a haze event in 2006."

p. 2, l. 20-2 Run-on sentence. Please correct here and throughout the manuscript (e.g., p.3, l.8-12, etc.) – corrected: "Haze concentrations in Singapore vary throughout the six year period from 2010 to 2015. Even though biomass burning contributes to (low) PM10 concentrations in Singapore throughout large parts of the year, some peaks in the PM10 observations can be explained by haze almost exclusively."

"In terms of biomass burning, the year in this region can be divided into seasons that relate to the monsoon seasons: FMA dominated by burning in Mainland Southeast Asia; during May, June, and July (MJJ) burning starts in northern Sumatra and traverses southward; ASO is characterised by burning in Southern Kalimantan and, in general, there is little or no burning influencing Singapore in November, December, and January (NDJ) (Campbell et al., 2013; Chew et al., 2013; Reid et al., 2012, 2013)." p. 2, l. 26-7 Poor sentence construction leads to a lack of clarity. - corrected: "These events caused extremely high PM10 concentrations in Singapore.

p. 3, l. 13-4 "related haze events are linked" is redundant. – As this section is used to link biomass burning and meteorology, we want to emphasise that the weather reports link weather and haze events.

p. 3, l. 23-4 A strong case for using dispersion modeling has not been made in the Introduction. –more details have been added to the Introduction and significant text to make the case stronger, see comments above and the section below: "The aim of this study is to investigate spatial variation of haze across Singapore through source attribution, including the variation in concentration and the contributing source regions across Singapore depending on the distance to source regions and the seasonal variation by looking at four recent haze events occurring during different seasons between January 2010 and December 2015. This is done by linking meteorology, biomass burning, and dispersion modelling to study how the origin of haze has varied across Singapore during this whole period. Fire radiative power and injection height from the CAMS global fire assimilation system (Kaiser et al., 2012) and higher resolution land-use data from the Centre for Remote Imaging, Sensing and Processing at the National University of Singapore have been used to calculate PM10 emissions from biomass burning in 29 defined source regions in Southeast Asia. Using the Met Office's numerical weather prediction (NWP) model to drive the Numerical Atmospheric-dispersion Modelling Environment (NAME), a Lagrangian particle trajectory model, we are able to attribute the haze arriving in Singapore to its source region and study the difference between major

contributing source regions at a western and an eastern monitoring station in Singapore."

p. 4, l. 31 Please provide a reference or equation for the MNMB. - the following reference has been included in the text: Seigneur, C., Pun, B., Pai, P., Louis, J-F., Solomom, P., Emery, C., Morris, R., Zahniser, M., Worsnop, D., Koutrakis, P., White, W., and Tombach, I.: Guidance for the performance evaluation of three dimensional air quality modelling systems for particulate matter and visibility, J. Air Waste Manage. Assoc., 50, 588–599, 2000.

p. 4, l. 35 Please clarify here, as is stated later, that the emissions are emitted over a 24-hour period at the rate of 1 g/s. Please state how the same emissions rate results in different total emissions from a single fire (if it does). – this emission rate only applies to the calculation of the air history maps, the text here is consistent with the biomass burning simulations. The section regarding the calculations of air history maps has been modified for clarity, see below.

p. 5, l. 7 Please note the limitations of comparing a tracer with PM10. – please refer to reply to previous reviewer comment

p. 6, l. 4-9 Please include an equation for the calculation described here. It is not clear how double counting in time is avoided given this description. - It is unclear which aspect the reviewer is referring to, we agree that the explanation was confusing. To clarify, each 10-day back run is based on an emission over 24 hours and one run is conducted for each emission-period so there is no double counting, the text has been updated to reflect this: "The resulting 10-day back air concentrations for each day's run were summed over the entire analysis period and a percentile value calculated to ascertain the likelihood of air originating from a particular grid cell (0.1 x 0.1) vis-à-vis other areas. The backruns shown were conducted from a receptor site in central Singapore, after comparison between a coastal receptor site and this inland site showed insignificant variation, meaning that the central receptor site can be considered representative for the whole island. This also helped inform the decision of domain size for the actual haze simulations."

p. 16, l. 1-2 Was it expected that the "atypical and different meteorological conditions" would cause variation of the source regions when those were dominated by nearby fires? It seems unlikely, so the sentence is unexpected. – Sentence has been re-worded: "Common for these two atypical haze events is little variation in the source regions across the monitoring stations, most likely due to the atypical and different meteorological conditions and the clear dominance of one source region."

p. 19, l. 6 "Similarly to" should be "Similar to". - corrected: "Similar to the results presented in Figure 3,"

p. 19, l. 15-7 Two statements contradict one another. The atypical haze events are said to have little variation between monitors, but then FMA 2014, one of the atypical haze events, is noted as having the largest difference in the next sentence. Please restate. – We disagree with the reviewer here, as one sentence concerned the variation between major contributing source regions the other relates to absolute concentrations at the two monitoring stations. The section has been removed however to avoid confusion.

p. 20, l. 3 Please eliminate the use of contractions here ("won't") and elsewhere. - Done: "peak concentrations will not always be captured in the model simulations"

p. 20, l. 5-6 No effort was made to show data that support this conclusion. Please show that it is true as suggested in the General Comments or remove the sentence. – This sentence is now part of a broader paragraph in the conclusion that addresses sources of uncertainty in the results. The specific sentence has been modified to: "Some of the varying difference between observed and modelled time series is also likely to be due to these many other sources of PM10 in Singapore."

p. 20, l. 9 The grammar and sentence construction in the Conclusions section of the document require careful revision. A significant part of the Conclusion has been

reworded to improve the text, please see the revised version.

Table 1 Please include the meaning of the abbreviations for the statistical correlations in the caption. – The caption now reads: "Table 2. Statistics for PM10, for both the western (NTU) and eastern (TP) monitoring stations and all years. Background concentration of 25 ug/m3 is subtracted from the observations for all stations for all years. The metrics considered are the Pearson correlation coefficient (R), the modified normalised mean bias (MNMB), the fractional gross error (FGE), and Factor of 2 (FAC2)."

Figure 2 The caption states "air history" but the colorbar label indicates "Air Conc Percentile", which seems to include information about emissions or concentrations rather than simply where the air has been. Please clarify. – The meaning is the same, the air history is given in percent, as mentioned above, the text has been modified to clearly describe the air history map. If the reviewer still finds the figures unclear we will be happy to change the labelling.

Figure 4 Please specify the meaning of the colored values. Are the lines for the source regions stacked? If not, they should be in order to show how they contribute to the total observed concentration. – The figure has been updated to show only observations and the total modelled concentration, as a stacked plot did not add clarity to the data.

Figures 6-8 The design of these figures is nice. It is not clear why only some of the contributing regions are colored. Please be consistent between the "other", which are grey in the legend, and the grey countries in the map. Also, please order the pie chart wedges in the same order as the legend names. The colors are too similar to be able to distinguish when the pie chart is not in the same order as the regions in the legend. – We had not spotted the issue with the grey regions, so thank you for raising it. We have modified the figures so that only major contributing source regions are coloured to highlight only relevant regions. The wedges have also been reordered in pie charts to match the legend.

Suggest reference Kim, P. S., Jacob, D. J., Mickley, L. J., Koplitz, S. N., Marlier, M.

[Figure]

E., DeFries, R. S., Myers, S. S., Chew, B. and Mao, Y. H.: Sensitivity of population smoke exposure to fire locations in Equatorial Asia, Atmos Environ, 102, 11–17, doi:10.1016/j.atmosenv.2014.09.045 , 2015 - We thank the referee for pointing out this paper and have included it in the introduction.   Review 2: Review of "Haze in Singapore - Source Attribution of Biomass Burning from Southeast Asia" by Hansen et al. This study investigates the sources of biomass burning from Southeast Asia for 6 years (2010-2015) using the Lagrangian dispersion model NAME. The tracer concentrations were evaluated using observations at two sites in Singapore. The authors also discussed the seasonal variations of emissions sources to Singapore. The topic of this paper is very interesting and important and I really appreciate the seasonal focus in this study. However, the conclusion and discussion are very confusing and repetitive. The paper lacks a coherent flow and the method section lacks significant information and clarity. I would only recommend this manuscript for publications in ACP only after substantial modifications to the manuscripts and figures. I also suggest the authors make higher quality figures for publication in ACP (Figs1, 4, . . .). In general, it is difficult to interpret the results and the discussion is weak.

We thank the reviewers for their constructive feedback and hope that our subsequent changes to the manuscript has improved the content and readability of the paper. We have rewritten and restructured the Introduction and removed the Discussions section to make the Results and Conclusions clearer and more coherent. The figures have been updated following the reviewers' suggestions.

1- Meteorology: Dispersion models are highly sensitive to their meteorological inputs. However, there is no analysis of the metrological values fed into the model. First, there is not clear statistical analysis or comparisons between modeled meteorology from UM and observations in that region. Without first evaluating the meteorological input, we cannot draw any conclusion from the Lagrangian models. For example, slight errors in modelled wind speed (and direction) and observations, makes the originating source region of the tracer very different. - The UM is a world leading NWP model (see

references in added text), and these are the data that were available to us at the time of the study. It was not feasible to conduct a thorough meteorological assessment of the UM for the whole region, but we have conducted an assessment of the UM data against observations that were available for Singapore. This is part of an internal report, that we summarise here for the reviewers, however we did not think it appropriate to add this to the final manuscript, but welcome further feedback:

"This report evaluates global UM model meteorological data, interpolated in NAME to obtain wind speed and direction, temperature and relative humidity data for a given location and time. These data are evaluated using meteorological observations available at 4 sites across Singapore. The results show that modelled wind speeds are higher on average than those observed during 2013 particularly during the monsoon seasons. Wind speeds are one of the most important factors affecting pollutant levels, particularly close to strong sources. As such, when applying higher wind speeds in the model than observed may reduce modelled pollutant levels below those observed. There are some differences in wind direction between the model and observations but the prevailing winds appear to be captured well throughout the year.

Observed ambient temperatures are slightly higher and more variable on average than the model although there is good agreement between the model and observations. Relative humidity is higher in the model than the observations on average with the greatest variability inherent in the observations. Rainfall does not appear well represented in NAME with higher means and more frequent low intensity events when compared to the observations which show less frequent high intensity rainfall typically associated with convective activity which dominates rainfall within the tropics. When considering the difference in total monthly rainfall between the model and observations, the former is predominantly higher during 2013 which may decrease modelled PM levels through wet deposition and contribute to the negative bias observed in both PM2.5 and PM10.

To augment the representation of the meteorology input in NAME, increasing both the

temporal and spatial resolution of data for example using hourly averages is likely to improve both the modelled meteorology and pollutant levels."

Second, P4:L1-2 mentions that the metrology runs were different (resolution and settings?) for different years. Based on previous studies, the modeled meteorological values (especially wind speed) are sensitive to the model resolution. Considering that in this manuscript, the authors compared different years with each other, I strongly recommend use of consistent settings and resolutions for the NWP runs. Else the inter-annual difference between the sources of biomass burning can easily be attributed to the difference between meteorological model differences. - Kim et al 2015 use the same meteorology in an attribution study of biomass burning, we believe that it is better to use the highest resolution met data available. In spite of the changes in the resolution of the met data, differences between major contributing source regions (pie charts) for earlier years – 2011 and 2012 – shows results similar to 2014 and 2015 in the sense that there is significant difference between major contributing source regions at the two monitoring stations for 2011 and less so for 2012, hence, the differences in major contributing source regions at the two monitoring stations in 2014 and 2015 is not due to the changes in the NWP data resolution.

In general, there are large discrepancies between modeled and observed PM10 for all years and both stations. The authors assumed a constant 25 ug/m3 background concentration for all year. However, the emissions from various sectors (especially residential) have high seasonal variability (see Sobhani et al. 2018). Considering the same background concentration (meaning the same contribution of other sectors to your PM10) for all seasons may introduce large errors to your analysis. – The paper the reviewer mentioned studies regions further north with stronger seasonal variation in domestic emissions than is the case for Singapore, so whilst we agree that variation in background PM occurs, there is no strong seasonal signal in Singapore.

Specific Comments: Introduction: 1- This part lacking significant discussions and references. For example, the reference this part of P2, L2 is missing, "it is not caused

by activities within Singapore, rather it is a transboundary problem caused by biomass burning across the wider region." Maybe adding sample studies. – References to Hertwig et al 2015, Reid et al 2013 added: "Though haze occurs in Singapore (Hertwig et al., 2015; Lee et al., 2016b; Nichol, 1997, 1998; Sulong et al., 2017), it is not caused by activities within Singapore, rather it is a transboundary problem caused by biomass burning across the wider region (see Fig. 1 for a map of the region), which occurs during distinct 'burning seasons' (Hertwig et al., 2015; Reid et al., 2013)."

2- P2, L13: I would recommend adding more discussions here. I suggest citing and/or describing some source attribution studies with the focus on other regions or bigger domains using different methods (Eulerian, Lagrangian, Observation analysis). I am not sure why the very few studies in the next lines are cited. Few examples for source appointment in different region of the world (with Eulerian methods) are: (Ikeda et al.,2017; Sobhani et al., 2018; Wang et al., 2011; Yang et al., 2017), With both Eulerian and Lagrangian methods:(Kulkarni et al., 2015) Observation Analysis + Lagrangian: (Winiger et al., 2017). – Additional details and references have been added as suggested: "Several previous studies have looked at attributing air pollution for different regions. Source attribution can be performed both through modelling and by looking at observations of air pollution in detail. For example, Heimann et al. (2015) carried out a source attribution study of UK air pollution using observations to distinguish between local and regional emissions, whereas Redington et al. (2016) estimated the sources of annual emissions of particulate matter from the UK and the EU by using the NAME model to look at threshold exceedences and episodes. Attribution studies have been performed using Eulerian models such as GEOS-chem, CMAQ, and WRF-STEM to study both Asia and the Arctic (Ikeda et al., 2017; Kim et al., 2015; Sobhani et al., 2018; Yang et al., 2017; Matsui et al., 2013) sometimes in combination with flight campaigns (Wang et al., 2011) to better constrain the emissions. Lagrangian models have also been used in combination with observations by Winiger et al. (2017). Combinations of Eulerian and Lagrangian models (Kulkarni et al., 2015) and Eulerian models and observations (Lee et al., 2017b) have been used to assess whether low visibility

days were caused by fossil fuel combustion, biomass burning or a combination of the two. In Southeast Asia, Reddington et al. (2014) used an Eulerian model to study haze and estimated emissions through a bottom up approach. Source apportionment for studies of biomass burning related degradation of air quality and visibility in Southeast Asia has also been applied by Lee et al. (2017a) who used the WRF model to study the sensitivity of the results to different met data and emission inventories and Engling et al. (2014), who used observations and a chemical mass balance receptor model to compare the chemical composition of total suspended particulate matter on haze and non-haze days during a haze event in 2006."

3- P2, L 22: Any reference for this sentence or is it the result of the study? If it is this result of the study please mention so. – This can be seen from Figure 4: In the six-year period, haze occurs almost annually during the season of August, September, and October (ASO), known as the haze season (see Fig. 4).

4- P2, L34 and P3, L3: Please add a reference for each sentence. – Citations and a supporting figure with reference added: "Generally, the inter-monsoon periods are characterised by light and variable winds, influenced by land and sea breezes with afternoon and early evening thunderstorms (Reid et al., 2012). The later inter-monsoon period is often wetter than the earlier inter-monsoon period (Chang et al., 2005; Reid et al., 2012). Furthermore, the inter-monsoon periods with weaker winds lead to air arriving in Singapore originating from the countries immediately west of and surrounding Singapore, see Fig A1. Previous studies have shown the importance of the ENSO in relation to reduction in convection and precipitation over the Martime Continent (MC) and corresponding increase in haze in Southeast Asia (Ashfold et al., 2017; Inness et al., 2015; Reid et al., 2012)."

5- In general, I suggest restructuring this section a bit for cohesiveness by moving few first sentences of the 5th paragraph (P2, L28) in introduction before the 4th paragraph (P2, L20). It is not clear if some the sentences in the 4th paragraph are result of this study or previous studies. – the Introduction has been modified and sentences have

been restructured for content and readability. Please refer to the updated manuscript.

6- It is not obvious why the focus of study is Singapore. Can you please add why the focus of this study Singapore? – Singapore is the focus of this study due to the availability of observations with high spatial and temporal resolution and the interest in understanding more about the regions impacting the air quality here. We have amended the text to make this clearer. "The Met Office (MO) and the Meteorological Service Singapore (MSS) have previously established a haze forecast system to predict haze in Singapore (Hertwig et al., 2015). This study advances the previous work to improve our understanding of haze and the underlying causes by analysing and attributing haze events of the recent past to their sources. The work focuses on Singapore due to the availability of air quality observations with high spatial and temporal resolution for recent years." and "The aim of this study is to investigate spatial variation of haze across Singapore through source attribution, including the variation in concentration and the contributing source regions across Singapore depending on the distance to source regions and the seasonal variation by looking at four recent haze events occurring during different seasons between January 2010 and December 2015. This is done by linking meteorology, biomass burning, and dispersion modelling to study how the origin of haze has varied across Singapore during this whole period"

Methods: 1- This section lacks a lot of details. Can you please add some information and a paragraph describing the NAME model? How are they numerically represented in the model? What kind of aerosol processes are accounted for? Are there any know biases? Why have you used this model for this study? – Description of the model has been added to text: "The Numerical Atmospheric-dispersion Modelling Environment (NAME) III v6.5 (Jones et al., 2007) is a Lagrangian particle trajectory model, designed to forecast dispersion and deposition of particles and gasses on all ranges. Using the topography from the relevant met input, as NAME does not resolve buildings or terrain on scales smaller than the NWP. Emissions in the model are released as particles that contain information of one or more species, during the simulation these particles are

exposed to various chemical and physical processes. NAME includes a comprehensive chemistry scheme which is not used in this study. Plume rise can also be considered, if applicable, in the model, here injection height is inferred from plume height information from the GFAS emissions. The only aerosol processes considered here are dispersion and wet and dry deposition of primary PM10. In NAME the dry deposition is parametrised using the resistance-based deposition velocity and wet deposition is based on the depletion equation. The advection is based on the winds obtained from the meteorology provided and a random component is added to represent the effects of atmospheric turbulence NAME is driven by meteorological data, which can be of various forms, in this case the Met Office's operational weather prediction model."

2- Can you please add some information and more description on the modeling setup for this study instead of just citing Hertwig et al. 2015. How are the wet and dry deposition processes calculated in the model? – See added text above.

3- Also, can you please describe your meteorological model (UM) and why this model is used to drive NAME? – Descriptions have been added to text: "The Unified Model (UM) is the Met Office's operational numerical weather forecast model. The UM is a global model based on the non-hydrostatic fully compressible deep-atmosphere equations of motion solved using at semi-implicit semi-Lagrangian approach on a regular longitude-latitude grid (Walters et al., 2017). Archived meteorology from the global version of the Met Office Unified Model (UM) (Davies et al., 2005) was used to drive the NAME model"

4- Significant lack of clarity and explanations regarding observations: It is not clear at all where the locations of the observation sites are (maybe add them to all maps and include lat lon of the measurement sites?)? The authors should add more information on the method of measurements in those locations. Also, tit is not clear where does these measurements come from (paper?, organization?)? Also, is this data available for public if so please include the link to the data either here or in the code and data availability section of the paper (or both). – The location of the stations has been added

as an insert to Fig 1, we thank the reviewer for pointing out that this information was missing. The observations data are from the Singapore NEA and are not publically available. The manuscript text has been extended to include details: "Some 20 observation sites are located across Singapore, of these, one eastern and one western station have been chosen for best representation of trans-boundary PM10 concentrations across the main island of Singapore. In this analysis, the western station, Nanyang Technological University (NTU; 1.34505N, 103.6836E), is located relatively close to the industrial western part of Singapore and the eastern station, Temasek Polytechnic (TP; 1.34506N, 103.9304E), is placed next to the polytechnic but is also near open fields and a water reservoir, the location of the two sites in Singapore can be seen from Fig. 1. In Singapore the National Environment Agency measure hourly PM10 at several sites using the beta attenuation monitoring, where air is drawn through a size selective inlet down a vertically mounted heated sample tube to reduce particle bound water and to decrease the relative humidity of the sample stream to prevent condensation on the filter tape. The PM is drawn onto a glass fibre filter tape placed between a detector and a 14C beta source. The beta beam passes upwards through the filter tape and the PM layer. The intensity of the beta beam is attenuated with the increasing mass load on the tape resulting in a reduced beta intensity measured by the detector. From a continuously integrated count rate the mass of the PM on the filter tape is calculated."

5- Air history map?? Do you mean PM10 or all aerosols (air?) lumped together? What chemical species and aerosols are considered in air history maps? Or is it only PM10 or tracer? This term is very confusing. - Following our reply to Reviewer 1, we agree that the explanation was confusing. To clarify, each 10-day back run is based on 24 hours emissions of PM10 so there is no double counting, the text has been updated to reflect this: "Air history maps provide an indication of where air at a given location has originated from. Fig. 2, illustrates an air history map for Singapore for the years 2010 to 2015. This helps determine the regions that influence the composition of the air arriving in Singapore. NAME backruns were conducted using the UM global Numerical Weather Prediction (NWP) model with PM10 as a tracer within a domain of 90.0_E, 140.0_E, 15.0_S, 23.0_N (Fig. 2). Wet and dry deposition are both turned on to simulate actual scenarios during the modelled time periods. Concentration values in the 0-2km layer were integrated at 10 minute time steps up till 10 days previous. The emission rate was set at a unit 1 g/s and emitted over 24 hours. A 10-day backrun was conducted for every single day in the six year time period from 2010 to 2015. The resulting 10-day back air concentrations for each day's run were summed over the entire analysis period and a percentile value calculated to ascertain the likelihood of air originating from a particular grid cell (0.1 x 0.1) vis-à-vis other areas. The backruns shown were conducted from a receptor site in central Singapore, after comparison between a coastal receptor site and this inland site showed insignificant variation, meaning that the central receptor site can be considered representative for the whole island. This also helped inform the decision of domain size for the actual haze simulations."

Results: 1- The assumption of 25 ug/m3 for both stations is problematic. Could it be because the background value from another source is higher in the western station??Also emissions from other important sectors are not accounted for which might cause the difference between the stations. – Looking at observations for periods without haze in 2013 and 2015 we found average concentrations between 23 and 29 ug/m3 at both sites (see numbers below), therefore a background concentration of 25 seems entirely reasonable when looking at haze contributions that are of a similar order of magnitude. We do acknowledge that we are not capturing variations in local contributions to these sites and that there does appear to be a slight difference between the two sites. 2013 : except June P09 25.99653 P28 23.66771 2013 ASO: P09 26.92174 P28 22.79807 2015 : except ASO P09 28.76008 P28 23.44001

2- P14, L13: Can you add some figure (maybe to SM) to show the meteorology difference for 2013 and other years. In general, this sentence is vague. What do you mean by 2013 is a unique year in terms of meteorology and burning? – Figure A1 has been added to show air history for each season for all years (also see point 6 under Figures

below), a reference to Oozeer et al 2016 has been added to P4 L10 to explain the meteorology of June 2013: "In June 2013 a typhoon (Gaveau et al., 2014) coincided with major atmospheric emissions from peat fires in Southeast Asia (Oozeer et al., 2016)."

3- Are peaks concurrent with biomass burning incidents? Several other factors influence the peaks. For example, high residential emissions in winter in South East Asia can be attributed to the peaks. – We are not entirely clear what the reviewer means by "winter" in this context. We are not convinced that high residential emissions in northern hemisphere winter have an impact as far south as Singapore. By "winter", we assume that the reviewer is saying it is cold in winter, therefore people burn more for heating. Then that "winter" will be during JJASON when it is wet in Northern SEA. Besides rain being great for wet deposition, the monsoonal flow is mainly southwesterly which is unfavourable for transport from Northern SEA to Singapore (Reid et al., 2013). People in maritime SEA also don't burn more for heating as we are in the tropics. We experience a "wet" and "wetter" season for JJASON and DJFMAM respectively (Reid et al., 2013).

Reid et al., 2013. Observing and understanding the Southeast Asian aerosol system by remote sensing. Atm Res.

We see no peaks/increased background concentrations in the observations during any particular season for any year, nor ASO 2013. See previous general introductory comments on meteorology, comment 1 and 2 in this section, and corresponding replies in this response to reviewers. The increased concentrations and peaks during ASO coincides with the biomass burning season in the region, which supports the idea that the increases are due to haze caused by biomass burning.

4- There is a large redundancy between results (section 3) and discussions (section 4). I suggest merging section 4 into section 3 and conclusions. - this has been done, please refer to the last section of the Results section and the Conclusions in the revised manuscript.

5- P 15, L30: It seems like the model did not capture the observation contrary to the claim. - See reply to comment below and the updated Conclusions: "For the four haze events focused on here, there is variability in the correlation between the modelled and observed time series, with the best correlations seen for haze events where the emission sources are close to Singapore. As discussed by Hertwig et al. (2015), uncertainty in these results originates from the emissions and the meteorology. For the former, the uncertainties result from the fact that the emissions used here are based on one daily snapshot of FRP and IH, and though some attempts are made to resolve issues with missing fire emissions caused by the lack of transparency of clouds the data will naturally be incomplete. At the same time, hourly emissions are calculated based on this one daily snapshot adding a temporal resolution that the data does not provide, which also means that peak concentrations will not always be captured in the model simulations. The meteorology provides another significant source of uncertainty, as is usually the case in atmospheric modelling. When considering the resolution of the analysis meteorology used here and the size of Singapore it is clear that there will be unresolved features in both topography and in the meteorology and hence in the dispersion modelling. However, the differences we see between the two sites show that we are starting to capture this scale. Uncertainties in the NWP data such as elevated wind speeds and too frequent and too low intensity precipitation will disperse the pollutants further and wash out more than should be, resulting in lower modelled concentrations These uncertainties naturally have a larger impact over longer travel distances, which is reflected in our statistics. It should also be kept in mind that the observations are measuring all PM10 and we are only modelling primary PM10 emissions from biomass burning. Other sources of PM10 include sea salt, dust, secondary organic aerosol, emissions from industry, local and transboundary road traffic, as well as domestic heating, not all of which are constant throughout the year. Some of the varying difference between observed and modelled time series is also likely to be due to these many other sources of PM10 in Singapore. However, in spite of these uncertainties our results show that we are able to model dispersion of particulate matter from

biomass burning in Southeast Asia and the resulting haze in Singapore with reasonable confidence."

6- P16, L10: The model significantly underestimates the peaks (30/125) Please explain why? - We have taken out the peak values from the paper due to the uncertainties we are discussing in the new conclusions section. Upon reflection on the reviewers feedback, we feel that using only an hourly peak value in the text misrepresents the ability of the model to represent the broader haze event.

Figures: 1- Does Figure 1 show the entire domain? It seems smaller that the domain mentioned in the method section. Please correct the figure include all the domain in this figure. – The figure has been updated to include the full domain

2- Please add the location of Singapore to Figure1 and all other spatial figures. It is hard for someone who does not know the regions geography to find Singapore in each figure. Based on the captions description the reader might think Singapore is located in the Riau Islands. – An insert has been added to Figure 1 showing Singapore and the relative locations of the monitoring stations. Singapore has also been added to the subplots of Fig 5 – 8; we had not spotted that it was missing, so thank you for pointing this out.

3- Figure 2: Wrong caption. The second line of caption of this figure is not related to manuscript. Central receptor sites???? Inland and coastal sites? Are these sites discussed in this manuscript?? – The figure caption has been reduced to the relevant information and the description in the text has been extended: "Figure 2. Air history map for 2010 - 2015, showing where air arriving in Singapore during this period originated from. Each shading shows the relative contribution of air/PM10 to the central receptor site in Singapore in percent integrated over the atmospheric column from 0 to 2 km."

"The backruns shown were conducted from a receptor site in central Singapore. After comparison between a coastal receptor site and this inland site showed insignificant

variation, meaning that the central receptor site can be considered representative for the whole island. This also helped inform the decision of domain size for the actual haze simulations."

4- Figure 2: Please correct the label title. - See reply above

5- Figure 2: Can you please add more discussions about this figure to the paper? It is confusing what these figures show. – The text has been modified to provide more information. Please refer to previous section of this reply to reviewers, e.g., the final paragraph of Methods/section 2 and reply to Comment 5 in Methods above.

6- Figure 2: It is very nice that you included figure A2 (Figure 2 for all years to the discussion). I suggest also adding similar plots for each season. (each season averaged over the years). The season specific "Air history maps" would make it easier to understand the transport pathway in different seasons as discussed in P6, L10. – Thank you. We have included these maps in the supplementary material. Also see response to point 2 in Methods above

7- Figure 3: This is a good plot; however, it is difficult to compare different years because of the different scales. Also, the y axix label denote T as the unit for monthly emission which is different from the caption. – The caption is consistent with the plot – the figure shows the monthly emissions in Tonnes which is also the tonnes emitted per month. We have decided not to change the y-axes, as too much information would be lost from using the same scale, but we have removed the units from the figure caption to avoid confusion.

8- Figure 4: This figure is very hard to read and should be modified before publications. First, it seems like hourly observations are plotted against (daily averages of model??). It is very hard to distinguish any modeled data points. Please make different plots for this figure. One way is showing monthly averages for both model and observation similar to Figure 3. Or time series of the daily observations as points overlaid on top of the modeled output. I recommend area chart for modeled value. Please include

sum multiple region as "the other regions" multiple of the regions in this plot. Please only include important regions with visible high impacts. Very few of the 28 regions are visible in this plot. Maybe another scale (e.g. a log scale) is better for the purpose of this plot. Please use the same scale for all years and denote the events discuss on these plots. The scale for all the years vary significantly. I would suggest having all PM 10 for all years on the same scale 0-700. Quick look at the plots, one might think there are higher pm 10 concentrations in 2010 compared to 2013 or 2015. – The figure has been modified so that all years are on the same scale and only one line is plotted for the modelled time series, which is the sum of all sources (see also response to Reviewer 1 on this matter). We investigated reducing the data to daily averages as suggested, but this removes too much detail from the results so we have chosen to stick with the hourly data.

9- Figures 5-8: Please add a title with the name of event to the plot. Please add the stations (locations of NTU and TP) and denote Singapore on the plots. – Title added and Singapore added in c) subfigures. The stations have been added to a subset image of Figure 1 as they would not be visible in these smaller figures.

10- Figure A2: What are the colored squares overlaid on the plots? Please put the figure in order that is mentioned in the paper. – The coloured squares have been removed, and the figures in the supplementary material have been reordered.

Minor Comments and Technical Corrections: 7- P2, L3: Fig 1 is technically not related to this sentence. – Agreed, however, a map of the region is beneficial to readers unfamiliar with Southeast Asia, an explanation has been added to this sentence: "(see Fig. 1 for a map of the region)"

8- P2, L3: I would recommend adding reference for the second part of this sentence. (the reference for transboundary problem. . .) – References have been added: "Though haze occurs in Singapore (Hertwig et al., 2015; Lee et al., 2016b; Nichol, 1997, 1998; Sulong et al., 2017), it is not caused by activities within Singapore, rather it is a trans-

boundary problem caused by biomass burning across the wider region (see Fig. 1 for a map of the region), which occurs during distinct 'burning seasons' (Hertwig et al., 2015; Reid et al., 2013)."

9- P2, L25: This sentence is very vague. Two events in each of June 2014-2015 and FMA 2014-2015. Or one event in June (2014 or 2015?) and one in FMA (2015?). Also, why using FMA vs June. I would recommend either using month or season. Can you use the season instead of June for consistency? – The sentence has been removed following the rewriting of the Introduction, we hope the updated manuscript reads more easily. For clarity: the 2013 event occurred during the month of June only, whereas the 2014 event lasted throughout FMA

10- P2, L 33: Can you please elaborate what you mean by north-east monsoon and south-west monsoon seasons? – it is outside of the scope of this paper to explain the monsoon in detail, figures have been added to the supplementary material to illustrate the seasons as defined in this study, and the manuscript has been amended to include: "Meteorologically, the year in Singapore is split into four seasons, two monsoon seasons separated by two inter-monsoon seasons: the north-east monsoon season is generally from December to early March and dominated by northeasterly winds; the first inter-monsoon period from late March through May; the south-west monsoon from June through September, with air in Singapore generally arriving from a southeasterly direction, and the second inter-monsoon period in October and November (Fing, 2012)."

11- P3, L 9: FMA acronym were explained last page and redundant here. – text revised to just FMA

12- P3, L30: What does NAME stands for? – Numerical Atmospheric-dispersion Modelling Environment, this and a model description has been added to text as described in comments above.

13- P3, L32: I recommend adding a figure to SM with the modeling domain. It seems

like figure 1 does not show the complete modeling domain. (not extended 14 S or 23 N) –Thank you for pointing this out, Figure 1 has been expanded to cover full domain.

14- P4, L1-2: What do you mean? Is it different meteorological setup for each year??? Is the resolution of NWP runs different from 17 km to 40 km?? – As the UM is an operational model the resolution has changed over the study period, this does not seem to impact the results significantly – see reply in section on Meteorology above.

15- P4, L 14: Please add what GFED stands for. – Global Fire Emissions Database, added to text: "with the Global Fire Emissions Database (GFED) data set"

16- P4, L 16: Redundant, very similar sentence in the above paragraph L10…. – Sentence removed

17- P4, L24-25: Can you please point to the pie charts? – This is a general comment referring to all figures throughout the paper. In general references are included to figures where appropriate, however reference has been added to the text to clarify: "Annual and seasonal pie charts showing the percentage contribution from each source region at each monitoring station have been produced, to capture the spatial variation of biomass burning across the island, e.g., Figs 5c-8c." 18- P4- L30-35: Can you please explain why did you use these metrics instead of other metrics like R2 or RMSE and many others? I suggest adding few more metrics to the tables 1 and 2 (R2 and RMSE). Please provide references or descriptions of the metrics used. – The metrics are explained in Methods, P6 L22 – 27, and additional references have been added. These metrics have been chosen as they have been used in other related studies. "The metrics considered are the Pearson correlation coefficient (R), i.e., the correlation between the model and observations used to get an indication of the match between patterns in the modelled and observed time series; the modified normalised mean bias (MNMB) which assesses the bias of the forecast and can have values between -2 and +2 (Seigneur et al., 2000); the fractional gross error (FGE) which gives the overall error of the model prediction and is limited between 0 and +2 (Ordóñez et al., 2010; Savage

et al., 2013); and finally, Factor of 2 (FAC2) which gives an indication of the fraction of the model results that fall within a factor 2 of the observations (Hertwig et al., 2015)."

19- P4, L35: I strongly suggest using daily averages instead of hourly averages. – This has been investigated, but not implemented as it does not add any clarity to the visualisation or data analysis.

20- P5, L7: Please add what NWP stands for. – "Numerical Weather Prediction", added to text

21- P5-6: Air history vs air conc. percentile? Please clarify? – This has been clarified in the figure caption and the text. Figure caption now reads: "Air history map for 2010 - 2015, showing where air arriving in Singapore during this period originated from. Each shading shows the relative contribution of air/PM10 to the central receptor site in Singapore in percent integrated over the atmospheric column from 0 to 2 km."

22- P7, L20: I would highly suggest including emission maps for each year. – The source regions and relative emissions are clear from Fig 3 and subfigures c) of Figs 5-8 highlight the major contributing source region(s) for each of the haze events. Figure limitations also prevent extra maps being added.

23- P7, L29: What is the reference for this sentence? –the figures in the text are based on analysis of the observations, see Results comment 1 above.

24- P7, L32: Why did you assume constant 25 ug/m3 for background concentration? Is there any reference for that? – This value was also used by Hertwig et al 2015, but we have also calculated that this is appropriate - see comments to Results comment 1 and comment above.

25- P8, L3-5: I suggest including the values for clarity and readability. - the values have been included: "For years like 2013, which was dominated by one extreme haze event, the correlation between the modelled time series and the observations is very high (0.79 and 0.80 at NTU and TP, respectively, see Table 3). To some extent, this is

also the case for the 2014 and 2015 events (0.27, 0.35 and 0.44, 0.43 for 2014 and 2015, respectively)."

26- P9, L4: Please add the name of the western monitoring station here and throughout the manuscript – this has been included: "When comparing concentrations between the two stations it can be seen that the concentrations are higher at the western monitoring station (NTU) most of the time. The opposite, concentrations at the eastern monitoring (TP) stations being higher than those at the western station (NTU),"

27- P14, L15-16: A sentence without a paragraph. remove the unnecessary line break. - the linebreak has been removed

28- P14, L 16-18: This sentence is very confusing. Please rephrase it. - the sentence has been rewritten to: "Though 2013 was generally a year with weak winds and average burning, the month of June was very unique, both in terms of meteorology and burning (Fig. 5). The June 2013 haze event was caused by a typhoon coinciding with intense burning in Riau (Fig. 3)."

29- P14: In general, adding the locations of the sites to the maps would make reading the paper much easier. – The locations of the monitoring sites have been added to Figure 1

30- P14, L24: It is not obvious if the maximum observed and modeled are concurrent or the values indicate maximum observation and maximum modeled value occurring at different times? - The values are not concurrent so have been removed to avoid confusion

31- P15, L25: Would you discuss FMA 2014 or February 2014 only. Earlier in the text you mentioned June 2013 and February 2014 as the haze events but discuss FMA 2014 as the haze event here. In general, there is a lack of consistency between using months and seasons. – this has been corrected to FMA, June is used on occasion as the 2013 event lasted less than a month which is not the case for the other events

considered here

32- P15, L30: Can you explain the reason why concentrations at TP is double of NTU? We have not looked into this in great detail, but the difference highlights the importance of local scale meteorology on results (both observations and model) and the importance of using higher-resolution data and Langrangian (and/or very high resolution Eulerian) models for interpretation at this spatial scale. As noted above the peak values this refers to have been removed from the text (see previous comment).

33- P16, L1-2: Very unclear and vague sentence. Different meteorology for events or between the monitoring stations? The sentence implies that in spite of the clear dominance of one source region, there is a little variation in the source regions across the monitoring stations??? – text updated for clarity: "Common for these two atypical haze events is little variation in the source regions across the monitoring stations, most likely due to the atypical and different meteorological conditions and the clear dominance of one source region."

34- P16, L3: I suggest adding ASO to the title of this section. The inconsistency between using southwest monsoon haze and ASO makes it confusing. – title now reads: "3.2 ASO - southeast monsoon season haze"

35- P16, L4: Please remove the extra line break. - linebreak removed 36- In general, not clear when the events are. I would highly suggest making a table including all the events discussed (and their corresponding figure) and also denoting each event on the time series plot. – Thank you for this suggestion. A table has been added to beginning of results section, see Table 1.

37- P 19, L1-3: This sentence is extremely confusing. For ASO or for the seasons with the most significant haze events including MJJ, FMA, and ASO? – Sentence moved to results section and reworded to read: "Of the seasons with the most significant haze events (e.g., MJJ 2013, FMA 2014, ASO 2014, and ASO 2015) in Singapore, the air history maps show that the region of influence for Singapore generally covers the

largest area during ASO when air is coming from southeasterly directions"

38- Please check the sentence constructions of discussions and conclusions sections carefully. – as mentioned above (Results comment 4) these sections have been restructured and reworded. Please see revised manuscript.

39- P19- 20: The discussions seems like an extended conclusion section and there is a lot of redundancy between results (section3), discussions (section 4) and conclusions (section 5), which decrease the readability of paper. – as above (Results comment 4) these sections have been restructured and reworded, please refer to the manuscript for updated text.

40- P20, L30: Please restate this sentence. It is very confusing. - the sentences now read: "Looking at emissions during ASO for the four years with the most variation across the island (2011, 2012, 2014, and 2015), the largest emissions were seen from Central Kalimantan, South Sumatra, Jambi, and also West Kalimantan. For events during FMA Cambodia, East Kalimantan, Myanmar, Thailand, and Vietnam showed larger emissions during FMA."

41- Code and data availability: Please include the link to the observations used for this study. – This is not possible as the observations are not publicly available.

Please also note the supplement to this comment:
https://www.atmos-chem-phys-discuss.net/acp-2018-311/acp-2018-311-AC1-supplement.pdf

---

## Referee Report (RR1)

**Review of revision of Hansen et al. for Atmospheric Chemistry and Physics**

*General Comments*

      In the revised submission of "Haze in Singapore - Source Attribution of Biomass Burning from Southeast Asia", Hansen et al. have responded to some of the concerns expressed in the first round of reviews. The manuscript is strengthened somewhat, but conflicting and unclear statements remain. Additionally, the grammar still needs to be refined though the duplication of the Discussion and Conclusions has been somewhat mitigated.

      To the concerns expressed that the degree of agreement between the observations and model do not constitute a case for "source apportionment", the authors only replied in agreement that the model performance was worse than they desired. If the authors think that a comparison of NAME results with observations is worth sharing with the community, they need to consider another title for the paper that does not involve "source apportionment". They have not modeled any other sources of PM10 nor have they made an effort to determine a statistically reasonable background concentration across the years. If I understand p. 7, l. 25-7 correctly, they are simply comparing a single back trajectory model convolved with satellite-based fire emissions estimate to observations at two distinct sites, which it would be a misrepresentation to call source apportionment.

      Since the authors did not address the major issue raised in the first round of reviews nor all of the specific comments that were made, it is not clear that they are willing or able to do so. Accordingly, I will not recommend publication in Atmospheric Chemistry and Physics unless they are able to revisit the original comments, some of which are repeated here, and address them thoroughly.

*Specific Comments*

The third sentence of the introduction is a run-on sentence. Additionally, the second sentence of the second paragraph in the introduction is a run-on sentence. I will not note other grammatical errors, but someone must correct these and others before this article is suitable for publication.

Acronyms including but not limited to NAME, GFAS, and CAMS are introduced after being used previously. Please ensure that every acronym is introduced upon first usage.

"Sec" is not an appropriate abbreviation for "Section". Please replace all occurrences.

"Air history" is an inaccurate term to refer to the convolution of emissions and back trajectory information. Please revise throughout.

| *Line* | *Comment* |
|---|---|
| p. 4, l. 16 | "validated" should be "evaluated" here and elsewhere (e.g., p.7, l. 4). Please change all occurrences when speaking of a comparison of measurements and models. Both have errors, which indicates that neither is sufficient for validating the other. |
| p. 7, l. 25-7 | This statement conflicts with the last sentence of the abstract, which states that "variation in local meteorology can impact concentrations of particulate matter significantly". If that were true, it would not be sufficient to use a central |

meteorological site. Please resolve by removing one of the statements. If the abstract statement is not changed, then the entire study needs to be presented for only one Singapore site. If a single receptor site was used for the back trajectories, it is not clear how the modeled concentrations in Figure 4 would be distinct as they appear to be or how these distinctions were investigated as indicated on p. 21, l. 1-3.

---

## Referee Report (RR2)

**Review of second revision of Hansen et al. for Atmospheric Chemistry and Physics**

*General Comments*

In the second revised submission of "Haze in Singapore - Source Attribution of Biomass Burning from Southeast Asia", Hansen et al. responded to the comments made in the first two rounds of reviews. Importantly, they clarified the application of the forward and backward runs of NAME. The backward run of NAME was done to identify the extent of the domain necessary for forward Lagrangian modeling with tagging of particles. The forward run was then conducted with particles that were tagged with the source region. This method is appropriately deemed source attribution. Since some other authors have used the information in the backwards run with observations to ascertain emissions rates, it should be clear that the only utility of the backward run was to show the extent of the domain. The "air history maps" do not contribute at all to the estimate of the influence of biomass burning on the sites; all of this information comes from the forward execution of NAME. Therefore, I would recommend changing the order of the methods section by moving 2.4 to 2.3 so that the backwards run can be described as a forerunner to the forwards runs.

The poor agreement between the model and observations is still concerning, but the authors have characterized this error. Since the observations are not used in ascertaining emissions rates, the reader is left to understand that the "source attribution" is only of the modeled concentrations rather than those observed. To this point, the authors seem to indicate in the responses to Reviewer 2 that "attribution" differs from "apportionment", which seems to be the justification for not treating the background concentration or other sources of $PM_{10}$ more carefully. However, any literature review will demonstrate that "attribution" and "apportionment" are used interchangeably in atmospheric modeling literature. Therefore, the paragraph beginning on page 11, line 25 of the revised manuscript should not depend on the distinction of these two words. Rather, using "attribute" instead of "perform an apportionment of" would appropriately indicate that there is no distinction between the meaning of these two words. Also, it would keep intact the clarifying argument the authors make that they are not trying to attribute observed $PM_{10}$ concentrations but modeled concentrations. It is left to the editor to decide whether apportioning modeled concentrations is a valuable endeavor when they differ as much as these do from observations.

Accordingly, an extensive effort to clarify the nature of the work has helped tremendously in this revision of the manuscript.

---

## Author Response (AR2)

**Paper review #2:**

Again, we wish to thank Reviewer 1 and the Co-editor for their time and comments on our manuscript. This second review is focused on clarifying the difference between the forward runs used for the attribution of biomass burning and the backrun used to generate the air history maps used to illustrate the general wind patterns of the air influencing Singapore. We also add explanations regarding the met data used and how the background concentrations were estimated.

For clarity we are attaching a pdf document highlighting the changes between the original submission and the manuscript following the two revisions. We hope that these additions, in addition to the improvements from the previous review, will accommodate the expectations of the reviewers and Co-editor.

**From Co-editor:**

Please take careful account of the comments of Referee 1 when preparing your revised manuscript. In addition, I have some more requests based on the review of and your responses to Referee 2.

First, I suggest that you include your assessment of UM model against observation data available in the manuscript. It was included in your response but not in the modified manuscript. Overall dispersion models are highly sensitive to their meteorological inputs and it is necessary to first evaluate these errors and discuss the uncertainties.

Second, I suggest adding a sentence or two to the manuscript on why you used inconsistent settings and resolutions for each year and if these inconsistencies have any impact on the results.

- The key thing to note is that the UM is an operational NWP model. The paragraph has been edited to the following, including the assessment of the UM:

"The Unified Model (UM) is the Met Office's operational numerical weather forecast model. The UM is a global model based on the non-hydrostatic fully compressible deep-atmosphere equations of motion solved using at semi-implicit semi-Lagrangian approach on a regular longitude-latitude grid (Walters et al., 2017). Archived analysis meteorology from the global version of the UM was used to drive NAME. As the UM is an operational model, the dynamical core and spatial resolution have changed throughout the period, from ~40 km over ~25 km to ~17 km resolution. However, for the majority of the study the resolution is constant at 25 km. These upgrades are described in Walters et al. (2011, 2017), and the relevant changes for dispersion modelling are summarised in Table 1. These changes are not expected to have a significant impact on the results, e.g., no significant differences in the deposition are seen across the change from instantaneous precipitation and cloud to 3-hour mean data in 2013.

"Global UM model meteorological data for 2013 have been evaluated using meteorological observations available at four sites across Singapore. The UM data are interpolated in NAME to obtain wind speed and direction, temperature, and relative 20 humidity data for each location and an hourly time resolution. The results show that modelled wind speeds are higher on average than those observed during 2013 particularly during the monsoon seasons. Wind speeds are one of the most important factors affecting pollutant levels, particularly close to strong sources. Although haze in Singapore is predominantly caused by long range transport of biomass smoke, the higher wind speeds in the model may contribute to reducing modelled pollutant levels below those observed. There are some differences in wind direction between the model and observations, but the prevailing wind directions are captured well throughout the year.

"Observed ambient temperatures are slightly higher and more variable on average than the model, although there is good agreement between the model and observations. Rainfall does not appear well represented with higher hourly means and more frequent low intensity events when compared to the observations, which show less frequent high-intensity rainfall associated with the convective activity that dominates rainfall within the tropics. Modelled total monthly rainfall is higher than observed during 2013, which may decrease modelled PM levels through wet deposition and contribute to the often negative bias observed in PM10, see Sect. 3. As discussed in Redington et al. (2016) and Hertwig et al. (2015), the uncertainties from the meteorological data feed into the dispersion simulation."

| Start date | Approx. horizontal resolution | Relevant change                                                                    |
|------------|-------------------------------|------------------------------------------------------------------------------------|
| 01/1/2010  | 40 km                         |                                                                                    |
| 20/1/2010  | 25 km                         | Horizontal resolution increase                                                     |
| 30/4/2013  | 25 km                         | Change from use of instantaneous
precipitation and cloud to 3-hour mean
data |
| 15/7/2014  | 17 km                         | Horizontal resolution increase                                                     |

Minor: please add the acronyms definitions to the caption for Table 1. (ASO: August-October, MJJ). Also, please reduce the number of labels on the y-axis of Fig. 4 (e.g. labels at every 200 instead of every 100). Perhaps it would be useful to improve the quality of this figure to improve readability"

- The acronym definitions have been added to the Table caption (now Table 2). The comments on the figure axes have been accommodated. Finally, the quality of the figures is significantly reduced during the upload, we would be more than happy to provide higher quality files.

**From Reviewer 1:**

**Review of revision of Hansen et al. for Atmospheric Chemistry and Physics**

**General Comments**

In the revised submission of "Haze in Singapore - Source Attribution of Biomass Burning from Southeast Asia", Hansen et al. have responded to some of the concerns expressed in the first round of reviews. The manuscript is strengthened somewhat, but conflicting and unclear statements remain. Additionally, the grammar still needs to be refined though the duplication of the Discussion and Conclusions has been somewhat mitigated.

To the concerns expressed that the degree of agreement between the observations and model do not constitute a case for "source apportionment", the authors only replied in agreement that the model performance was worse than they desired. If the authors think that a comparison of NAME results with observations is worth sharing with the community, they need to consider another title for the paper that does not involve "source apportionment". They have not modeled any other sources of PM10 nor have they made an effort to determine a statistically reasonable background concentration across the years.

- In the previous response to reviewers, background concentrations at the two stations were estimated based on averages of the  $PM_{10}$  concentrations during non-haze periods, this follows the method used by Kim et al, 2015:

"We estimate the smoke concentration at each site in the observations by subtracting as baseline the mean concentration for the bracketing non-burning months (June and December)."

**https://doi.org/10.1016/j.atmosenv.2014.09.045**

The text now reads:

"Observations of PM10 in Singapore from 2010 - 2015 show an overall background concentration during months of little or no burning of between 23 - 29  $\mu$ g/m3 at the two monitoring stations. These values fit well with those determined in other studies for Singapore. For example, Hertwig et al. (2015) estimated background concentrations for PM10 to be around 30  $\mu$ g/m3, based on the 2013 haze episode. In general, both background and peak concentrations vary between NTU and TP. Following the approach of Kim et al. (2015) we assume a constant background of 25  $\mu$ g/m3 for the PM10 observations at both sites and subtract this value from the observation time series. "

If I understand p. 7, I. 25-7 correctly, they are simply comparing a single back trajectory model convolved with satellite-based fire emissions estimate to observations at two distinct sites, which it would be a misrepresentation to call source apportionment.

- This is not correct. For clarity: we have performed two kinds of simulations:

- Forward year-long simulations using emissions from all of the areas depicted in Figure 1, which the main results (time series and pie charts) are based on.
- Back-runs with numerous particles (also known inverse simulations), which are only used for the air history maps. These have been included as a means to provide the reader with an indication of the region of influence on the air reaching Singapore.

We had thought that this was clear in the original text, but acknowledge that this was obviously not the case. The differences between the two simulations have been highlighted in the text where relevant, e.g.:

- P4, line 9: "...biomass burning, and forwards and backwards dispersion modelling to study how..."
- P8, line 29: "In addition to the forward simulations with the GFAS biomass emissions, backward (inverse) runs were conducted with NAME using the UM...
- P12, line 14: "The timeseries and pie charts are based on results from the forward NAME simulations."

And the subsection on air history maps was revised to:

"Air history maps provide a visual indication of where air at a given location has originated from. This helps to determine the regions that influence the composition of the air arriving at this location. To construct air history maps for Singapore, backward (inverse) runs were conducted with NAME, in addition to the forward simulations with the GFAS biomass burning emissions (Sect. 2.3). Fig. 2 illustrates the air history map for Singapore for the years 2010 to 2015. For each day in the six year period from 2010 to 2015, a 10-day backrun was conducted using meteorological input from the UM global model within a domain of 90.0°E, 140.0°E, 15.0°S, 23.0°N (Fig. 2). PM10 was emitted as a tracer from a receptor site in central Singapore and model particles were released over the first 24 hours with an emission rate of 1 g/s. The resulting concentration values in the 0-2km layer were output on a 0.1° × 0.1° resolution grid and integrated backwards in time for 10 days with a timestep of 10 minutes. By summing the results from several runs, air history data can be produced for different seasons and years, as well as the total for the whole period. A higher integrated concentration indicates that more air has passed through a grid cell on route to the receptor site, compared to a grid cell with a lower concentration. For each analysis period, the multiple corresponding 10-day air concentrations were summed for each grid cell and for the total domain. A percentile value was then calculated to ascertain the proportion of air influenced by a particular grid cell vis-àvis other areas.

Comparison between the inland site and a coastal receptor site showed insignificant variation, meaning that the central receptor site can be considered representative for the whole island when averaged over time. The results of the air history simulations helped inform the decision of domain size for the forward haze simulations. "

Since the authors did not address the major issue raised in the first round of reviews nor all of the specific comments that were made, it is not clear that they are willing or able to do so. Accordingly, I will not recommend publication in Atmospheric Chemistry and Physics unless they are able to revisit the original comments, some of which are repeated here, and address them thoroughly.

- At the risk of repeating ourselves, we have gone through the previous review comments below, to clarify and elaborate on our responses where relevant.

**Review 1:**

**General Comments**

The major difficulty in this work seems to be representing PM10 concentrations in Singapore except when biomass burning occurs very close by the sites (e.g., Riau or the Malaysian Peninsula) (Fig. 4). The two times this occurred during the modeled period are labeled as "Atypical haze" (Sect. 3.1). As the authors note, the PM10 concentrations in Singapore, especially in the other "haze seasons", are likely from diverse sources; however, the time series (Fig. 4) and statistical analysis of the agreement between the model and observations (Tables 1, 2) do not support the notion that "Southeast monsoon season haze" (Sect. 3.2) is represented accurately enough by the model to claim attribution of these sources to biomass burning regions. The difficulty may result from the lines of source regions not being stacked in the time series plot (Fig. 4), but the correlation statistics are fairly poor for periods other than 2013 MJJ, which supports the interpretation of the time series that the model is not representing concentrations well. One reason may be that the PM10 from biomass burning from regions not adjacent to Singapore is not well represented by the modeled processes. Another reason may be, as noted by the authors, that much of the haze comes from biomass burning sources not detected by the model. Finally, it may be that the PM10 is not from biomass burning. The first two causes would indicate that this modeling framework is not appropriate for attribution of biomass burning contributions to PM10 concentrations in Singapore. The final potential explanation could be shown by more sophisticated analysis of the background PM10 concentrations rather than using a fixed value of 25 µg/m3; if this explanation does not hold, it seems unreasonable to claim that these attributions are appropriate for episodes other than those in which the Riau and Malaysian Peninsula contributions dominate.

Given the difficulty of interpreting the results, the weakness of the Discussion and Conclusions sections seem reasonable. Specifically, the Conclusions seem to be very repetitive of the Discussion. Accordingly, I recommend this manuscript for publication in Atmospheric Chemistry and Physics once this major issue has been

**addressed. Additionally, specific comments have been included below that should also be considered in the revisions of the manuscript.**

We thank the reviewers for their constructive feedback and hope that our subsequent changes to the manuscript has improved the content and readability of the paper. We acknowledge that our results are not as good as would be desired, but this work now acts as a stepping stone to improving the understanding of haze in the region and how to model haze at high temporal and spatial resolution. And we feel it is worth sharing our findings with the community. We have also highlighted uncertainties in the work, including the limitations to emissions and focussed on the four events with the best correlation between model and observations. We have rewritten and restructured the Introduction and removed the Discussions section to make the Results and Conclusions clearer and more coherent. The figures have been updated following the reviewers' suggestions.

**Specific Comments**

**Line Comment**

*p.* 1, *l.* 8-13 The meaning of this interpretation of the results is unclear especially in the phrases "several and varying source regions", "atypical haze episodes … characterized by atypical weather conditions", and "different set of five regions dominate". Please refine.

**- The paragraph has been changed to:**

As should be expected, the relatively stronger Southeast monsoonal winds that coincide with increased biomass burning activities in the Maritime Continent create the main haze season from August to October (ASO), which brings particulate matter from varying source regions to Singapore. Five regions dominate as the source of pollution during recent haze seasons. In contrast, off-season haze episodes in Singapore are characterised by unusual weather conditions, ideal for biomass burning, and emissions dominated by a single source region (for each event). The two most recent off-season haze events in mid-2013 and early-2014 have different source regions, which differ to the major contributing source regions for the haze season.

*p. 1, I. 14 "climate" seems inappropriate when only 5 years have been considered.*The study covers 6 years, but more pertinently also considers the impact of El Nino and other non-weather timescale phenomena. The use of the word "climate" in this context seems appropriate and is supported by other literature.

**p. 2, I. 3 "Though the popular press often attribute" - grammatical error. Also, scientific literature has supported similar conclusions (Kim et al., 2015).**

Reference added as suggested:

"Scientific studies such as Kim et al 2015 as well as the popular press often attribute peatland destruction and related haze in the region to Indonesia (Reid et al 2013), however, the haze cannot be attributed to only one region or country alone."

*p.* 2, *l.* 17-9 Please insert a comma as "approach, and source" or divide these two thoughts into separate sentences. What type of source apportionment was applied by Lee et al. (2017) and Engling et al. (2014)? Please be more specific. – more details have been added to the text:

"Several previous studies have looked at attributing air pollution for different regions. Source attribution can be performed both through modelling and by looking at observations of air pollution in detail. For example, Heimann et al. (2015) carried out a source attribution study of UK air pollution using observations to distinguish between local and regional emissions, whereas Redington et al. (2016) estimated the sources of annual emissions of particulate matter from the UK and the EU by using the NAME model to look at threshold exceedences and episodes. Attribution studies have been performed using Eulerian models such as GEOS-chem, CMAQ, and WRF-STEM to study both Asia and the Arctic (Ikeda et al., 2017; Kim et al., 2015; Sobhani et al., 2018; Yang et al., 2017; Matsui et al., 2013) sometimes in combination with flight campaigns (Wang et al., 2011) to better constrain the emissions. Lagrangian models have also been used in combination with observations by Winiger et al. (2017). Combinations of Eulerian and Lagrangian models (Kulkarni et al., 2015) and Eulerian models and observations (Lee et al., 2017b) have been used to assess whether low visibility days were caused by fossil fuel combustion, biomass burning or a combination of the two. In Southeast Asia, Reddington et al. (2014) used an Eulerian model to study haze and estimated emissions through a bottom up approach. Source apportionment for studies of biomass burning related degradation of air quality and visibility in Southeast Asia has also been applied by Lee et al. (2017a) who used the WRF model to study the sensitivity of the results to different met data and emission inventories and Engling et al. (2014), who used observations and a chemical mass balance receptor model to compare the chemical composition of total suspended particulate matter on haze and non-haze days during a haze event in 2006."

p. 2, I. 20-2 Run-on sentence. Please correct here and throughout the manuscript (e.g., p.3, I.8-12, etc.)

- corrected:

"Haze concentrations in Singapore vary throughout the six year period from 2010 to 2015. Even though biomass burning contributes to (low) PM10 concentrations in Singapore throughout large parts of the year, some peaks in the PM10 observations can be explained by haze almost exclusively."

"In terms of biomass burning, the year in this region can be divided into seasons that relate to the monsoon seasons: FMA dominated by burning in Mainland Southeast Asia; during May, June, and July (MJJ) burning starts in northern Sumatra and traverses southward; ASO is characterised by burning in Southern Kalimantan and, in general, there is little or no burning influencing Singapore in November, December, and January (NDJ) (Campbell et al., 2013; Chew et al., 2013; Reid et al., 2012, 2013)."

*p.* 2, *l.* 26-7 Poor sentence construction leads to a lack of clarity. - corrected:

"These events caused extremely high PM10 concentrations in Singapore.

**p. 3, l. 13-4 "related haze events are linked" is redundant.**

– As this section is used to link biomass burning and meteorology, we want to emphasise that the weather reports link weather and haze events.

**p. 3, I. 23-4 A strong case for using dispersion modeling has not been made in the Introduction.**

-more details have been added to the Introduction and significant text to make the case stronger, see comments above and the section below:

"The aim of this study is to investigate spatial variation of haze across Singapore through source attribution, including the variation in concentration and the contributing source regions across Singapore depending on the distance to source regions and the seasonal variation by looking at four recent haze events occurring during different seasons between January 2010 and December 2015. This is done by linking meteorology, biomass burning, and dispersion modelling to study how the origin of haze has varied across Singapore during this whole period. Fire radiative power and injection height from the CAMS global fire assimilation system (Kaiser et al., 2012) and higher resolution land-use data from the Centre for Remote Imaging, Sensing and Processing at the National University of Singapore have been used to calculate PM10 emissions from biomass burning in 29 defined source regions in Southeast Asia. Using the Met Office's numerical weather prediction (NWP) model to drive the Numerical Atmospheric-dispersion Modelling Environment (NAME), a Lagrangian particle trajectory model, we are able to attribute the haze arriving in Singapore to its source region and study the difference between major contributing source regions at a western and an eastern monitoring station in Singapore."

**p. 4, I. 31 Please provide a reference or equation for the MNMB.**

- the following reference has been included in the text:

Seigneur, C., Pun, B., Pai, P., Louis, J-F., Solomom, P., Emery, C., Morris, R., Zahniser, M., Worsnop, D., Koutrakis, P., White, W., and Tombach, I.: Guidance for the performance evaluation of three dimensional air quality modelling systems for particulate matter and visibility, J. Air Waste Manage. Assoc., 50, 588–599, 2000.

**p. 4, I. 35 Please clarify here, as is stated later, that the emissions are emitted over a 24-hour period at the rate of 1 g/s. Please state how the same emissions rate results in different total emissions from a single fire (if it does).**

– this emission rate only applies to the calculation of the air history maps, the text here is consistent with the biomass burning simulations. The section regarding the calculations of air history maps has been modified for clarity, see below.

Review #2: The Methods section has been divided in to subsections to add clarity to the difference between the simulations. The description of the air history maps has been revised for clarity again, please refer to the updated text above.

p. 5, I. 7 Please note the limitations of comparing a tracer with PM10.

- please refer to reply to previous reviewer comment

**p.* 6, *l.* 4-9 Please include an equation for the calculation described here. It is not clear how double counting in time is avoided given this description.**

- It is unclear which aspect the reviewer is referring to, we agree that the explanation was confusing. To clarify, each 10-day back run is based on an emission over 24 hours and one run is conducted for each emission-period so there is no double counting, the text has been updated to reflect this:

"The resulting 10-day back air concentrations for each day's run were summed over the entire analysis period and a percentile value calculated to ascertain the likelihood of air originating from a particular grid cell  $(0.1 \times 0.1)$  vis-à-vis other areas. The backruns shown were conducted from a receptor site in central Singapore, after comparison between a coastal receptor site and this inland site showed insignificant variation, meaning that the central receptor site can be considered representative for the whole island. This also helped inform the decision of domain size for the actual haze simulations."

Following the second review the text has been revised to:

"Air history maps provide a visual indication of where air at a given location has originated from. This helps to determine the regions that influence the composition of the air arriving at this location. To construct air history maps for Singapore, backward (inverse) runs were conducted with NAME, in addition to the forward simulations with the GFAS biomass burning emissions (Sect. 2.3). Fig. 2 illustrates the air history map for Singapore for the years 2010 to 2015. For each day in the six year period from 2010 to 2015, a 10-day backrun was conducted using meteorological input from the UM global model within a domain of 90.0°E, 140.0°E, 15.0°S, 23.0°N (Fig. 2). PM10 was emitted as a tracer from a receptor site in central Singapore and model particles were released over the first 24 hours with an emission rate of 1 g/s. The resulting concentration values in the 0-2km layer were output on a 0.1° × 0.1° resolution grid and integrated backwards in time for 10 days with a timestep of 10 minutes. By summing the results from several runs, air history data can be produced for different seasons and years, as well as the total for the whole period. A higher integrated concentration indicates that more air has passed through a grid cell on route to the receptor site, compared to a grid cell with a lower concentration. For each analysis period, the multiple corresponding 10-day air concentrations were summed for each grid cell and for the total domain. A percentile value was then calculated to ascertain the proportion of air influenced by a particular grid cell vis-àvis other areas."

p. 16, I. 1-2 Was it expected that the "atypical and different meteorological conditions" would cause variation of the source regions when those were dominated by nearby fires? It seems unlikely, so the sentence is unexpected.
– Sentence has been reworded:

"Common for these two atypical haze events is little variation in the source regions across the monitoring stations, most likely due to the atypical and different meteorological conditions and the clear dominance of one source region."

p. 19, I. 6 "Similarly to" should be "Similar to".

- corrected:

"Similar to the results presented in Figure 3,"

p. 19, I. 15-7 Two statements contradict one another. The atypical haze events are said to have little variation between monitors, but then FMA 2014, one of the atypical haze events, is noted as having the largest difference in the next sentence. Please restate.

- We disagree with the reviewer here, as one sentence concerned the variation between major contributing source regions the other relates to absolute concentrations at the two monitoring stations. The section has been removed to avoid confusion.

*p.* 20, *I.* 3 Please eliminate the use of contractions here ("won't") and elsewhere. - Done:

"peak concentrations will not always be captured in the model simulations"

p. 20, I. 5-6 No effort was made to show data that support this conclusion. Please show that it is true as suggested in the General Comments or remove the sentence.
This sentence is now part of a broader paragraph in the conclusion that addresses sources of uncertainty in the results. The specific sentence has been modified to:
"Some of the varying difference between observed and modelled time series is also likely to be due to these many other sources of PM10 in Singapore."

**p.* 20, *I.* 9 The grammar and sentence construction in the Conclusions section of the document require careful revision.**

A significant part of the Conclusion has been reworded to improve the text, please see the revised version of the manuscript and the document highlighting the changes between the two versions of the manuscript.

**Table 1 Please include the meaning of the abbreviations for the statistical correlations in the caption. – The caption now reads:**

"Table 2. Statistics for PM10, for both the western (NTU) and eastern (TP) monitoring stations and all years. Background concentration of 25 ug/m3 is subtracted from the observations for all stations for all years. The metrics considered are the Pearson correlation coefficient (R), the modified normalised mean bias (MNMB), the fractional gross error (FGE), and Factor of 2 (FAC2)."

Figure 2 The caption states "air history" but the colorbar label indicates "Air Conc Percentile", which seems to include information about emissions or concentrations rather than simply where the air has been. Please clarify.

– The meaning is the same, the air history is given in percent, as mentioned above, the text has been modified to clearly describe the air history map. If the reviewer still finds the figures unclear we will be happy to change the labelling.

Figure 4 Please specify the meaning of the colored values. Are the lines for the source regions stacked? If not, they should be in order to show how they contribute to the total observed concentration.

- The figure has been updated to show only observations and the total modelled concentration, as a stacked plot did not add clarity to the data.

Figures 6-8 The design of these figures is nice. It is not clear why only some of the contributing regions are colored. Please be consistent between the "other", which are grey in the legend, and the grey countries in the map. Also, please order the pie chart wedges in the same order as the legend names. The colors are too similar to be able to distinguish when the pie chart is not in the same order as the regions in the legend.

- We had not spotted the issue with the grey regions, so thank you for raising it. We have modified the figures so that only major contributing source regions are coloured to highlight only relevant regions. The wedges have also been reordered in pie charts to match the legend.

Suggest reference Kim, P. S., Jacob, D. J., Mickley, L. J., Koplitz, S. N., Marlier, M. E., DeFries, R. S., Myers, S. S., Chew, B. and Mao, Y. H.: Sensitivity of population smoke exposure to fire locations in Equatorial Asia, Atmos Environ, 102, 11–17, doi:10.1016/j.atmosenv.2014.09.045, 2015

- We thank the referee for pointing out this paper and have included it in the introduction.

**Review 2:**

**Review of "Haze in Singapore - Source Attribution of Biomass Burning from Southeast Asia"**

**by Hansen et al.**

This study investigates the sources of biomass burning from Southeast Asia for 6 years (2010-2015) using the Lagrangian dispersion model NAME. The tracer concentrations were evaluated using observations at two sites in Singapore. The authors also discussed the seasonal variations of emissions sources to Singapore. The topic of this paper is very interesting and important and I really appreciate the seasonal focus in this study. However, the conclusion and discussion are very confusing and repetitive. The paper lacks a coherent flow and the method section lacks significant information and clarity. I would only recommend this manuscript for publications in ACP only after substantial modifications to the manuscripts and figures. I also suggest the authors make higher quality figures for publication in ACP (Figs1, 4, ...). In general, it is difficult to interpret the results and the discussion is weak.

We thank the reviewers for their constructive feedback and hope that our subsequent changes to the manuscript has improved the content and readability of the paper.

We have rewritten and restructured the Introduction and removed the Discussions section to make the Results and Conclusions clearer and more coherent. The figures have been updated following the reviewers' suggestions.

**1- Meteorology:**

Dispersion models are highly sensitive to their meteorological inputs. However, there is no analysis of the metrological values fed into the model. First, there is not clear statistical analysis or comparisons between modeled meteorology from UM and observations in that region. Without first evaluating the meteorological input, we cannot draw any conclusion from the Lagrangian models. For example, slight errors in modelled wind speed (and direction) and observations, makes the originating source region of the tracer very different.

The UM is a world leading NWP model (see references in added text), and these are the data that were available to us at the time of the study. It was not feasible to conduct a thorough meteorological assessment of the UM for the whole region, but we have conducted an assessment of the UM data against observations that were available for Singapore. This is part of an internal report, that we summarise here for the reviewers, however we did not think it appropriate to add this to the final manuscript, but welcome further feedback:

"This report evaluates global UM model meteorological data, interpolated in NAME to obtain wind speed and direction, temperature and relative humidity data for a given location and time. These data are evaluated using meteorological observations available at 4 sites across Singapore.

The results show that modelled wind speeds are higher on average than those observed during 2013 particularly during the monsoon seasons. Wind speeds are one of the most important factors affecting pollutant levels, particularly close to strong sources. As such, when applying higher wind speeds in the model than observed may reduce modelled pollutant levels below those observed. There are some differences in wind direction between the model and observations but the prevailing winds appear to be captured well throughout the year.

Observed ambient temperatures are slightly higher and more variable on average than the model although there is good agreement between the model and observations. Relative humidity is higher in the model than the observations on average with the greatest variability inherent in the observations. Rainfall does not appear well represented in NAME with higher means and more frequent low intensity events when compared to the observations which show less frequent high intensity rainfall typically associated with convective activity which dominates rainfall within the tropics. When considering the difference in total monthly rainfall between the model and observations, the former is predominantly higher during 2013 which may decrease modelled PM levels through wet deposition and contribute to the negative bias observed in both PM2.5 and PM10.

To augment the representation of the meteorology input in NAME, increasing both the temporal and spatial resolution of data for example using hourly averages is likely to improve both the modelled meteorology and pollutant levels."

Review #2: This has now been included in the revised manuscript with a table highlighting relevant changes:

"The Unified Model (UM) is the Met Office's operational numerical weather forecast model. The UM is a global model based on the non-hydrostatic fully compressible deep-atmosphere equations of motion solved using at semi-implicit semi-Lagrangian approach on a regular longitude-latitude grid (Walters et al., 2017). Archived analysis meteorology from the global version of the UM was used to drive NAME. As the UM is an operational model, the dynamical core and spatial resolution have changed throughout the period, from ~40 km over ~25 km to ~17 km resolution. However, for the majority of the study the resolution is constant at 25 km. These upgrades are described in Walters et al. (2011, 2017), and the relevant changes for dispersion modelling are summarised in Table 1. These changes are not expected to have a significant impact on the results, e.g., no significant differences in the deposition are seen across the change from instantaneous precipitation and cloud to 3-hour mean data in 2013.

"Global UM model meteorological data for 2013 have been evaluated using meteorological observations available at four sites across Singapore. The UM data are interpolated in NAME to obtain wind speed and direction, temperature, and relative 20 humidity data for each location and an hourly time resolution. The results show that modelled wind speeds are higher on average than those observed during 2013 particularly during the monsoon seasons. Wind speeds are one of the most important factors affecting pollutant levels, particularly close to strong sources. Although haze in Singapore is predominantly caused by long range transport of biomass smoke, the higher wind speeds in the model may contribute to reducing modelled pollutant levels below those observed. There are some differences in wind direction between the model and observations, but the prevailing wind directions are captured well throughout the year.

"Observed ambient temperatures are slightly higher and more variable on average than the model, although there is good agreement between the model and observations. Rainfall does not appear well represented with higher hourly means and more frequent low intensity events when compared to the observations, which show less frequent high-intensity rainfall associated with the convective activity that dominates rainfall within the tropics. Modelled total monthly rainfall is higher than observed during 2013, which may decrease modelled PM levels through wet deposition and contribute to the often negative bias observed in PM10, see Sect. 3. As discussed in Redington et al. (2016) and Hertwig et al. (2015), the uncertainties from the meteorological data feed into the dispersion simulation"

Second, P4:L1-2 mentions that the metrology runs were different (resolution and settings?) for different years. Based on previous studies, the modeled meteorological values (especially wind speed) are sensitive to the model resolution. Considering that in this manuscript, the authors compared different years with each other, I strongly recommend use of consistent settings and resolutions for the NWP runs. Else the inter-annual difference between the sources of biomass burning can easily be attributed to the difference between meteorological model differences.

Kim et al 2015 use the same meteorology in an attribution study of biomass burning, we believe that it is better to use the highest resolution met data available. In spite of the changes in the resolution of the met data, differences between major contributing source regions (pie charts) for earlier years – 2011 and 2012 – shows results similar to 2014 and 2015 in the sense that there is significant difference between major contributing source regions at the two monitoring stations for 2011 and less so for 2012, hence, the differences in major contributing source regions at the two monitoring stations in 2014 and 2015 is not due to the changes in the NWP data resolution.

In general, there are large discrepancies between modeled and observed PM10 for all years and both stations. The authors assumed a constant 25 ug/m3 background concentration for all year. However, the emissions from various sectors (especially residential) have high seasonal variability (see Sobhani et al. 2018). Considering the same background concentration (meaning the same contribution of other sectors to your PM10) for all seasons may introduce large errors to your analysis.

- The paper the reviewer mentioned studies regions further north with stronger seasonal variation in domestic emissions than is the case for Singapore, so whilst we agree that variation in background PM occurs, there is no strong seasonal signal in Singapore. Review #2: we have elaborated on our reasoning behind the constant background concentration and commented on the difference between attribution and apportionment:

"Subtracting a constant background from the observations does not give the exact contribution of PM10 from biomass burning alone because it does not remove all contributions from all other sources. However, it does give an indication of the periods with increased PM10 concentrations due to biomass burning. This is not an attempt to perform an *apportionment* of the observed PM10 concentrations in Singapore, as the observations, even with the subtracted background concentration, still includes contributions from sources other than biomass burning. However, the

observations minus the constant background compared to the modelled time series provides an indication of the performance of the model, and through that the quality of the input used for the modelling. Using the modelled time series and the related source region information we are able to *attribute* the PM10 contribution in Singapore originating from biomass burning in Southeast Asia to the respective source regions. "

**Specific Comments: Introduction:**

1- This part lacking significant discussions and references. For example, the reference this part of P2, L2 is missing, "it is not caused by activities within Singapore, rather it is a transboundary problem caused by biomass burning across the wider region." Maybe adding sample studies.

- References to Hertwig et al 2015, Reid et al 2013 have been added:

"Though haze occurs in Singapore (Hertwig et al., 2015; Lee et al., 2016b; Nichol, 1997, 1998; Sulong et al., 2017), it is not caused by activities within Singapore, rather it is a transboundary problem caused by biomass burning across the wider region (see Fig. 1 for a map of the region), which occurs during distinct 'burning seasons' (Hertwig et al., 2015; Reid et al., 2013)."

2- P2, L13: I would recommend adding more discussions here. I suggest citing and/or describing some source attribution studies with the focus on other regions or bigger domains using different methods (Eulerian, Lagrangian, Observation analysis). I am not sure why the very few studies in the next lines are cited. Few examples for source appointment in different region of the world (with Eulerian methods) are: (Ikeda et al., 2017; Sobhani et al., 2018; Wang et al., 2011; Yang et al., 2017), With both Eulerian and Lagrangian methods:(Kulkarni et al., 2015) Observation Analysis + Lagrangian: (Winiger et al., 2017).

- Additional details and references have been added as suggested:

"Several previous studies have looked at attributing air pollution for different regions. Source attribution can be performed both through modelling and by looking at observations of air pollution in detail. For example, Heimann et al. (2015) carried out a source attribution study of UK air pollution using observations to distinguish between local and regional emissions, whereas Redington et al. (2016) estimated the sources of annual emissions of particulate matter from the UK and the EU by using the NAME model to look at threshold exceedences and episodes. Attribution studies have been performed using Eulerian models such as GEOS-chem, CMAQ, and WRF-STEM to study both Asia and the Arctic (Ikeda et al., 2017; Kim et al., 2015; Sobhani et al., 2018; Yang et al., 2017; Matsui et al., 2013) sometimes in combination with flight campaigns (Wang et al., 2011) to better constrain the emissions. Lagrangian models have also been used in combination with observations by Winiger et al. (2017). Combinations of Eulerian and Lagrangian models (Kulkarni et al., 2015) and Eulerian models and observations (Lee et al., 2017b) have been used to assess whether low visibility days were caused by fossil fuel combustion, biomass burning or a combination of the two. In Southeast Asia, Reddington et al. (2014) used an Eulerian model to study haze and estimated emissions through a bottom up approach. Source apportionment for studies of biomass burning related degradation of air quality and visibility in Southeast Asia has also been applied by Lee et al. (2017a) who used the WRF model to study the

sensitivity of the results to different met data and emission inventories and Engling et al. (2014), who used observations and a chemical mass balance receptor model to compare the chemical composition of total suspended particulate matter on haze and non-haze days during a haze event in 2006."

**3- P2, L 22: Any reference for this sentence or is it the result of the study? If it is this result of the study please mention so.**

- This can be seen from Figure 4:

In the six-year period, haze occurs almost annually during the season of August, September, and October (ASO), known as the haze season (see Fig. 4).

**4- P2, L34 and P3, L3: Please add a reference for each sentence.**

- Citations and a supporting figure with reference added:

"Generally, the inter-monsoon periods are characterised by light and variable winds, influenced by land and sea breezes with afternoon and early evening thunderstorms (Reid et al., 2012). The later inter-monsoon period is often wetter than the earlier inter-monsoon period (Chang et al., 2005; Reid et al., 2012). Furthermore, the inter-monsoon periods with weaker winds lead to air arriving in Singapore originating from the countries immediately west of and surrounding Singapore, see Fig A1. Previous studies have shown the importance of the ENSO in relation to reduction in convection and precipitation over the Martime Continent (MC) and corresponding increase in haze in Southeast Asia (Ashfold et al., 2017; Inness et al., 2015; Reid et al., 2012)."

5- In general, I suggest restructuring this section a bit for cohesiveness by moving few first sentences of the 5th paragraph (P2, L28) in introduction before the 4th paragraph (P2, L20). It is not clear if some the sentences in the 4th paragraph are result of this study or previous studies.

- the Introduction has been modified and sentences have been restructured for content and readability. Please refer to the updated manuscript.

6- It is not obvious why the focus of study is Singapore. Can you please add why the focus of this study Singapore?

 Singapore is the focus of this study due to the availability of observations with high spatial and temporal resolution and the interest in understanding more about the regions impacting the air quality here. We have amended the text to make this clearer.

"The Met Office (MO) and the Meteorological Service Singapore (MSS) have previously established a haze forecast system to predict haze in Singapore (Hertwig et al., 2015). This study advances the previous work to improve our understanding of haze and the underlying causes by analysing and attributing haze events of the recent past to their sources. The work focuses on Singapore due to the availability of air quality observations with high spatial and temporal resolution for recent years."

"The aim of this study is to investigate spatial variation of haze across Singapore through source attribution, including the variation in concentration and the contributing source regions across Singapore depending on the distance to source regions and the seasonal variation by looking at four recent haze events occurring during different seasons between January 2010 and December 2015. This is done by linking meteorology, biomass burning, and dispersion modelling to study how the origin of haze has varied across Singapore during this whole period"

**Methods:**

1- This section lacks a lot of details. Can you please add some information and a paragraph describing the NAME model? How are they numerically represented in the model? What kind of aerosol processes are accounted for? Are there any know biases? Why have you used this model for this study?

- Description of the model has been added to text:

"The Numerical Atmospheric-dispersion Modelling Environment (NAME) III v6.5 (Jones et al., 2007) is a Lagrangian particle trajectory model, designed to forecast dispersion and deposition of particles and gasses on all ranges. Using the topography from the relevant met input, as NAME does not resolve buildings or terrain on scales smaller than the NWP. Emissions in the model are released as particles that contain information of one or more species, during the simulation these particles are exposed to various chemical and physical processes. NAME includes a comprehensive chemistry scheme which is not used in this study. Plume rise can also be considered, if applicable, in the model, here injection height is inferred from plume height information from the GFAS emissions. The only aerosol processes considered here are dispersion and wet and dry deposition of primary PM10. In NAME the dry deposition is parametrised using the resistance-based deposition velocity and wet deposition is based on the depletion equation. The advection is based on the winds obtained from the meteorology provided and a random component is added to represent the effects of atmospheric turbulence NAME is driven by meteorological data, which can be of various forms, in this case the Met Office's operational weather prediction model."

2- Can you please add some information and more description on the modeling setup for this study instead of just citing Hertwig et al. 2015. How are the wet and dry deposition processes calculated in the model?

- See added text above.

3- Also, can you please describe your meteorological model (UM) and why this model is used to drive NAME?

- Descriptions have been added to text:

"The Unified Model (UM) is the Met Office's operational numerical weather forecast model. The UM is a global model based on the non-hydrostatic fully compressible deep-atmosphere equations of motion solved using at semi-implicit semi-Lagrangian approach on a regular longitude-latitude grid (Walters et al., 2017). Archived meteorology from the global version of the Met Office Unified Model (UM) (Davies et al., 2005) was used to drive the NAME model"

Review #2: This has been elaborated on further in the second review, please see updated manuscript and document highlighting changes between the original submission and the revised manuscript.

4- Significant lack of clarity and explanations regarding observations: It is not clear at all where the locations of the observation sites are (maybe add them to all maps and include lat lon of the measurement sites?)? The authors should add more information on the method of

measurements in those locations. Also, tit is not clear where does these measurements come from (paper?, organization?)? Also, is this data available for public if so please include the link to the data either here or in the code and data availability section of the paper (or both).

- The location of the stations has been added as an insert to Fig 1, we thank the reviewer for pointing out that this information was missing. The observations data are from the Singapore NEA and are not publically available. The manuscript text has been extended to include details:

"Some 20 observation sites are located across Singapore, of these, one eastern and one western station have been chosen for best representation of trans-boundary PM10 concentrations across the main island of Singapore. In this analysis, the western station, Nanyang Technological University (NTU; 1.34505N, 103.6836E), is located relatively close to the industrial western part of Singapore and the eastern station, Temasek Polytechnic (TP; 1.34506N, 103.9304E), is placed next to the polytechnic but is also near open fields and a water reservoir, the location of the two sites in Singapore can be seen from Fig. 1. In Singapore the National Environment Agency measure hourly PM10 at several sites using the beta attenuation monitoring, where air is drawn through a size selective inlet down a vertically mounted heated sample tube to reduce particle bound water and to decrease the relative humidity of the sample stream to prevent condensation on the filter tape. The PM is drawn onto a glass fibre filter tape placed between a detector and a 14C beta source. The beta beam passes upwards through the filter tape and the PM layer. The intensity of the beta beam is attenuated with the increasing mass load on the tape resulting in a reduced beta intensity measured by the detector. From a continuously integrated count rate the mass of the PM on the filter tape is calculated."

**5- Air history map?? Do you mean PM10 or all aerosols (air?) lumped together? What chemical species and aerosols are considered in air history maps? Or is it only PM10 or tracer? This term is very confusing.**

Following our reply to Reviewer 1, we agree that the explanation was confusing.
 To clarify, each 10-day back run is based on 24 hours emissions of PM10 so there is no double counting, the text has been updated to reflect this:

"Air history maps provide an indication of where air at a given location has originated from. Fig. 2, illustrates an air history map for Singapore for the years 2010 to 2015. This helps determine the regions that influence the composition of the air arriving in Singapore. NAME backruns were conducted using the UM global Numerical Weather Prediction (NWP) model with PM10 as a tracer within a domain of 90.0\_E, 140.0\_E, 15.0\_S, 23.0\_N (Fig. 2). Wet and dry deposition are both turned on to simulate actual scenarios during the modelled time periods. Concentration values in the 0-2km layer were integrated at 10 minute time steps up till 10 days previous. The emission rate was set at a unit 1 g/s and emitted over 24 hours. A 10-day backrun was conducted for every single day in the six year time period from 2010 to 2015. The resulting 10-day back air concentrations for each day's run were summed over the entire analysis period and a percentile value calculated to ascertain the likelihood of air originating from a particular grid cell (0.1 x 0.1) vis-à-vis other areas. The backruns shown were conducted from a receptor site in central Singapore, after comparison between a coastal receptor site and this inland site showed insignificant variation, meaning that the central receptor site can be considered representative for the whole island. This also helped inform the decision of domain size for the actual haze simulations."

Review #2: The text describing the air history maps has been revised again, please refer to previous comments and the revised manuscript.

**Results:**

1- The assumption of 25 ug/m3 for both stations is problematic. Could it be because the background value from another source is higher in the western station??Also emissions from other important sectors are not accounted for which might cause the difference between the stations.

- Looking at observations for periods without haze in 2013 and 2015 we found average concentrations between 23 and 29 ug/m3 at both sites (see numbers below), therefore a background concentration of 25 seems entirely reasonable when looking at haze contributions that are of a similar order of magnitude. We do acknowledge that we are not capturing variations in local contributions to these sites and that there does appear to be a slight difference between the two sites.

| 2013:e   | except June |
|----------|-------------|
| P09      | 25.99653    |
| P28      | 23.66771    |
| 2013 A   | SO:         |
| P09      | 26.92174    |
| P28      | 22.79807    |
| 2015 : e | except ASO  |
| P09      | 28.76008    |
| P28      | 23.44001    |
|          |             |

Review #2: In the previous response to reviewers, background concentrations at the two stations were estimated based on averages of the  $PM_{10}$  concentrations during non-haze periods, this follows the method used by Kim et al, 2015:

"We estimate the smoke concentration at each site in the observations by subtracting as baseline the mean concentration for the bracketing non-burning months (June and December)."

**https://doi.org/10.1016/j.atmosenv.2014.09.045**

The text now reads:

"Observations of PM10 in Singapore from 2010 - 2015 show an overall background concentration during months of little or no burning of between 23 - 29  $\mu$ g/m3 at the two monitoring stations. These values fit well with those determined in other studies for Singapore. For example, Hertwig et al. (2015) estimated background concentrations for PM10 to be around 30  $\mu$ g/m3, based on the 2013 haze episode. In general, both background and peak concentrations vary between NTU and TP. Following the approach of Kim et al. (2015) we assume a constant background of 25  $\mu$ g/m3 for the PM10 observations at both sites and subtract this value from the observation time series. "

2- P14, L13: Can you add some figure (maybe to SM) to show the meteorology difference for 2013 and other years. In general, this sentence is vague. What do you mean by 2013 is a unique year in terms of meteorology and burning?

- Figure A1 has been added to show air history for each season for all years (also see point 6 under Figures below), a reference to Oozeer et al 2016 has been added to P4 L10 to explain the meteorology of June 2013:

"In June 2013 a typhoon (Gaveau et al., 2014) coincided with major atmospheric emissions from peat fires in Southeast Asia (Oozeer et al., 2016)."

3- Are peaks concurrent with biomass burning incidents? Several other factors influence the peaks. For example, high residential emissions in winter in South East Asia can be attributed to the peaks.

- We are not entirely clear what the reviewer means by "winter" in this context. We are not convinced that high residential emissions in northern hemisphere winter have an impact as far south as Singapore.

By "winter", we assume that the reviewer is saying it is cold in winter, therefore people burn more for heating. Then that "winter" will be during JJASON when it is wet in Northern SEA. Besides rain being great for wet deposition, the monsoonal flow is mainly southwesterly which is unfavourable for transport from Northern SEA to Singapore (Reid et al., 2013). People in maritime SEA also don't burn more for heating as we are in the tropics. We experience a "wet" and "wetter" season for JJASON and DJFMAM respectively (Reid et al., 2013).

Reid et al., 2013. Observing and understanding the Southeast Asian aerosol system by remote sensing. Atm Res.

We see no peaks/increased background concentrations in the observations during any particular season for any year, nor ASO 2013. See previous general introductory comments on meteorology, comment 1 and 2 in this section, and corresponding replies in this response to reviewers. The increased concentrations and peaks during ASO coincides with the biomass burning season in the region, which supports the idea that the increases are due to haze caused by biomass burning.

**4- There is a large redundancy between results (section 3) and discussions (section 4). I suggest merging section 4 into section 3 and conclusions.**

- this has been done, please refer to the last section of the Results section and the Conclusions in the revised manuscript.

**5- P 15, L30: It seems like the model did not capture the observation contrary to the claim.**

- See reply to comment below and the updated Conclusions:

"For the four haze events focused on here, there is variability in the correlation between the modelled and observed time series, with the best correlations seen for haze events where the emission sources are close to Singapore. As discussed by Hertwig et al. (2015), uncertainty in these results originates from the emissions and the meteorology. For the former, the uncertainties result from the fact that the emissions used here are based on one daily snapshot of FRP and IH, and though some attempts are made to resolve issues with missing fire emissions caused by the lack of transparency of clouds the data will naturally be incomplete. At the same time, hourly emissions are calculated based on this one daily snapshot adding a temporal resolution that the data does not provide, which also means that peak concentrations will not always be captured in the model simulations. The meteorology provides another significant source of uncertainty, as is usually the case in atmospheric modelling. When considering the resolution of the analysis meteorology used here and the size of Singapore it is clear that there will be unresolved features in both topography and in the meteorology and hence in the dispersion modelling. However, the differences we see between the two sites show that we are starting to capture this scale. Uncertainties in the NWP data such as elevated wind speeds and too frequent and too low intensity precipitation will disperse the pollutants further and wash out more than should be, resulting in lower modelled concentrations These uncertainties naturally have a larger impact over longer travel distances, which is reflected in our statistics. It should also be kept in mind that the observations are measuring all PM10 and we are only modelling primary PM10 emissions from biomass burning. Other sources of PM10 include sea salt, dust, secondary organic aerosol, emissions from industry, local and transboundary road traffic, as well as domestic heating, not all of which are constant throughout the year. Some of the varying difference between observed and modelled time series is also likely to be due to these many other sources of PM10 in Singapore. However, in spite of these uncertainties our results show that we are able to model dispersion of particulate matter from biomass burning in Southeast Asia and the resulting haze in Singapore with reasonable confidence."

**6- P16, L10: The model significantly underestimates the peaks (30/125) Please explain why?**

- We have taken out the peak values from the paper due to the uncertainties we are discussing in the new conclusions section. Upon reflection on the reviewers feedback, we feel that using only an hourly peak value in the text misrepresents the ability of the model to represent the broader haze event.

**Figures:**

1- Does Figure 1 show the entire domain? It seems smaller that the domain mentioned in the method section. Please correct the figure include all the domain in this figure.
The figure has been updated to include the full domain

**2-** Please add the location of Singapore to Figure1 and all other spatial figures. It is hard for someone who does not know the regions geography to find Singapore in each figure. Based on the captions description the reader might think Singapore is located in the Riau Islands. – An insert has been added to Figure 1 showing Singapore and the relative locations of the monitoring stations. Singapore has also been added to the subplots of Fig 5 – 8; we had not spotted that it was missing, so thank you for pointing this out.

3- Figure 2: Wrong caption. The second line of caption of this figure is not related to manuscript. Central receptor sites???? Inland and coastal sites? Are these sites discussed in this manuscript??

- The figure caption has been reduced to the relevant information and the description in the text has been extended:

"Figure 2. Air history map for 2010 - 2015, showing where air arriving in Singapore during this period originated from. Each shading shows the relative contribution of

air/PM10 to the central receptor site in Singapore in percent integrated over the atmospheric column from 0 to 2 km."

"The backruns shown were conducted from a receptor site in central Singapore. After comparison between a coastal receptor site and this inland site showed insignificant variation, meaning that the central receptor site can be considered representative for the whole island. This also helped inform the decision of domain size for the actual haze simulations."

*4- Figure 2: Please correct the label title.*See reply above

5- Figure 2: Can you please add more discussions about this figure to the paper? It is confusing what these figures show.

- The text has been modified to provide more information. Please refer to previous section of this reply to reviewers, e.g., the final paragraph of Methods/Section 2 and reply to Comment 5 in Methods above.

Review #2: The text in subsection 2.6 has been updated to better explain the process for creating air history maps.

**6-** Figure 2: It is very nice that you included figure A2 (Figure 2 for all years to the discussion). I suggest also adding similar plots for each season. (each season averaged over the years). The season specific "Air history maps" would make it easier to understand the transport pathway in different seasons as discussed in P6, L10.

- Thank you. We have included these maps in the supplementary material. Also see response to point 2 in Methods above

7- Figure 3: This is a good plot; however, it is difficult to compare different years because of the different scales. Also, the y axix label denote T as the unit for monthly emission which is different from the caption.

- The caption is consistent with the plot – the figure shows the monthly emissions in Tonnes which is also the tonnes emitted per month. We have decided not to change the y-axes, as too much information would be lost from using the same scale, but we have removed the units from the figure caption to avoid confusion.

8- Figure 4: This figure is very hard to read and should be modified before publications. First, it seems like hourly observations are plotted against (daily averages of model??). It is very hard to distinguish any modeled data points. Please make different plots for this figure. One way is showing monthly averages for both model and observation similar to Figure 3. Or time series of the daily observations as points overlaid on top of the modeled output. I recommend area chart for modeled value. Please include sum multiple region as "the other regions" multiple of the regions in this plot. Please only include important regions with visible high impacts. Very few of the 28 regions are visible in this plot. Maybe another scale (e.g. a log scale) is better for the purpose of this plot. Please use the same scale for all years and denote the events discuss on these plots.

The scale for all the years vary significantly. I would suggest having all PM 10 for all years on the same scale 0-700. Quick look at the plots, one might think there are higher pm 10 concentrations in 2010 compared to 2013 or 2015.

- The figure has been modified so that all years are on the same scale and only one line is plotted for the modelled time series, which is the sum of all sources (see also response to Reviewer 1 on this matter). We investigated reducing the data to daily averages as suggested, but this removes too much detail from the results so we have chosen to stick with the hourly data.

9- Figures 5-8: Please add a title with the name of event to the plot. Please add the stations (locations of NTU and TP) and denote Singapore on the plots.

- Title added and Singapore added in c) subfigures. The stations have been added to a subset image of Figure 1 as they would not be visible in these smaller figures.

10- Figure A2: What are the colored squares overlaid on the plots? Please put the figure in order that is mentioned in the paper.

- The coloured squares have been removed, and the figures in the supplementary material have been reordered.

**Minor Comments and Technical Corrections:**

7- P2, L3: Fig 1 is technically not related to this sentence.

- Agreed, however, a map of the region is beneficial to readers unfamiliar with Southeast Asia, an explanation has been added to this sentence:

"(see Fig. 1 for a map of the region)"

8- P2, L3: I would recommend adding reference for the second part of this sentence. (the reference for transboundary problem...)

- References have been added:

"Though haze occurs in Singapore (Hertwig et al., 2015; Lee et al., 2016b; Nichol, 1997, 1998; Sulong et al., 2017), it is not caused by activities within Singapore, rather it is a transboundary problem caused by biomass burning across the wider region (see Fig. 1 for a map of the region), which occurs during distinct 'burning seasons' (Hertwig et al., 2015; Reid et al., 2013)."

9- P2, L25: This sentence is very vague. Two events in each of June 2014-2015 and FMA 2014-2015. Or one event in June (2014 or 2015?) and one in FMA (2015?). Also, why using FMA vs June. I would recommend either using month or season. Can you use the season instead of June for consistency?

– The sentence has been removed following the rewriting of the Introduction, we hope the updated manuscript reads more easily. For clarity: the 2013 event occurred during the month of June only, whereas the 2014 event lasted throughout FMA

10- P2, L 33: Can you please elaborate what you mean by north-east monsoon and southwest monsoon seasons?

– it is outside of the scope of this paper to explain the monsoon in detail, figures have been added to the supplementary material to illustrate the seasons as defined in this study, and the manuscript has been amended to include:

"Meteorologically, the year in Singapore is split into four seasons, two monsoon seasons separated by two inter-monsoon seasons: the north-east monsoon season

is generally from December to early March and dominated by northeasterly winds; the first inter-monsoon period from late March through May; the south-west monsoon from June through September, with air in Singapore generally arriving from a southeasterly direction, and the second inter-monsoon period in October and November (Fing, 2012)."

*11- P3, L 9: FMA acronym were explained last page and redundant here.* – text revised to just FMA

12-P3, L30: What does NAME stands for?

– Numerical Atmospheric-dispersion Modelling Environment, this and a model description have been added to text as described in comments above.

13- P3, L32: I recommend adding a figure to SM with the modeling domain. It seems like figure 1 does not show the complete modeling domain. (not extended 14 S or 23 N)
-Thank you for pointing this out, Figure 1 has been expanded to cover the full domain.

14- P4, L1-2: What do you mean? Is it different meteorological setup for each year??? Is the resolution of NWP runs different from 17 km to 40 km??

As the UM is an operational model the resolution has changed over the study period, this does not seem to impact the results significantly – see reply in section on Meteorology above.
 Review #2: A table has been added to the revised manuscript to provide information on the relevant model changes.

15- P4, L 14: Please add what GFED stands for.
Global Fire Emissions Database, added to text:
"with the Global Fire Emissions Database (GFED) data set"

*16- P4, L 16: Redundant, very similar sentence in the above paragraph L10....* – Sentence removed

**17-P4, L24-25: Can you please point to the pie charts?**

- This is a general comment referring to all figures throughout the paper. In general references are included to figures where appropriate, however reference has been added to the text to clarify:

"Annual and seasonal pie charts showing the percentage contribution from each source region at each monitoring station have been produced, to capture the spatial variation of biomass burning across the island, e.g., Figs 5c-8c."

18- P4- L30-35: Can you please explain why did you use these metrics instead of other metrics like R2 or RMSE and many others? I suggest adding few more metrics to the tables 1 and 2 (R2 and RMSE). Please provide references or descriptions of the metrics used.
The metrics are explained in Methods, P6 L22 – 27, and additional references have been added. These metrics have been chosen as they have been used in other related studies.

"The metrics considered are the Pearson correlation coefficient (R), i.e., the correlation between the model and observations used to get an indication of the match between patterns in the modelled and observed time series; the modified normalised mean bias (MNMB) which assesses the bias of the forecast and can have values between -2 and +2 (Seigneur et al., 2000); the fractional gross error (FGE) which gives the overall error of the model prediction and is limited between 0 and +2 (Ordóñez et al., 2010; Savage et al., 2013); and finally, Factor of 2 (FAC2) which gives an indication of the fraction of the model results that fall within a factor 2 of the observations (Hertwig et al., 2015)."

19- P4, L35: I strongly suggest using daily averages instead of hourly averages.
This has been investigated, but not implemented as it does not add any clarity to the visualisation or data analysis.

**20- P5, L7: Please add what NWP stands for.**

- "Numerical Weather Prediction", added to text

**21- P5-6: Air history vs air conc. percentile? Please clarify?**

– This has been clarified in the figure caption and the text. Figure caption now reads: "Air history map for 2010 - 2015, showing where air arriving in Singapore during this period originated from. Each shading shows the relative contribution of air/PM10 to the central receptor site in Singapore in percent integrated over the atmospheric column from 0 to 2 km."

22- P7, L20: I would highly suggest including emission maps for each year.

- The source regions and relative emissions are clear from Fig 3 and subfigures c) of Figs 5-8 highlight the major contributing source region(s) for each of the haze events. Figure limitations also prevent extra maps being added.

**23- P7, L29: What is the reference for this sentence?**

-the figures in the text are based on analysis of the observations, see Results comment 1 above.

**24- P7, L32: Why did you assume constant 25 ug/m3 for background concentration? Is there any reference for that?**

- This value was also used by Hertwig et al 2015, but we have also calculated that this is appropriate - see comments to Results comment 1 and comment above.

Review #2: This has been elaborated on following the second review, please refer to P11, line19 onwards in the updated manuscript.

**25- P8, L3-5: I suggest including the values for clarity and readability.**

- the values have been included:

"For years like 2013, which was dominated by one extreme haze event, the correlation between the modelled time series and the observations is very high (0.79)

and 0.80 at NTU and TP, respectively, see Table 3). To some extent, this is also the case for the 2014 and 2015 events (0.27, 0.35 and 0.44, 0.43 for 2014 and 2015, respectively)."

26- P9, L4: Please add the name of the western monitoring station here and throughout the manuscript

- this has been included:

"When comparing concentrations between the two stations it can be seen that the concentrations are higher at the western monitoring station (NTU) most of the time. The opposite, concentrations at the eastern monitoring (TP) stations being higher than those at the western station (NTU),"

27- *P14*, *L15-16*: A sentence without a paragraph. remove the unnecessary line break. - the linebreak has been removed

28-P14, L 16-18: This sentence is very confusing. Please rephrase it.

- the sentence has been rewritten to:

"Though 2013 was generally a year with weak winds and average burning, the month of June was very unique, both in terms of meteorology and burning (Fig. 5). The June 2013 haze event was caused by a typhoon coinciding with intense burning in Riau (Fig. 3)."

29- P14: In general, adding the locations of the sites to the maps would make reading the paper much easier.

- The locations of the monitoring sites have been added to Figure 1

30- P14, L24: It is not obvious if the maximum observed and modeled are concurrent or the values indicate maximum observation and maximum modeled value occurring at different times?

- The values are not concurrent so have been removed to avoid confusion

31- P15, L25: Would you discuss FMA 2014 or February 2014 only. Earlier in the text you mentioned June 2013 and February 2014 as the haze events but discuss FMA 2014 as the haze event here. In general, there is a lack of consistency between using months and seasons. – this has been corrected to FMA, June is used on occasion as the 2013 event lasted less than a month which is not the case for the other events considered here

32- P15, L30: Can you explain the reason why concentrations at TP is double of NTU? We have not looked into this in great detail, but the difference highlights the importance of local scale meteorology on results (both observations and model) and the importance of using higher-resolution data and Langrangian (and/or very high resolution Eulerian) models for interpretation at this spatial scale. As noted above the peak values this refers to have been removed from the text (see previous comment). 33- P16, L1-2: Very unclear and vague sentence. Different meteorology for events or between the monitoring stations? The sentence implies that in spite of the clear dominance of one source region, there is a little variation in the source regions across the monitoring stations???

– text updated for clarity:

"Common for these two atypical haze events is little variation in the source regions across the monitoring stations, most likely due to the atypical and different meteorological conditions and the clear dominance of one source region."

34- P16, L3: I suggest adding ASO to the title of this section. The inconsistency between using southwest monsoon haze and ASO makes it confusing.
– title now reads:

"3.2 ASO - southeast monsoon season haze"

35- P16, L4: Please remove the extra line break.

- linebreak removed

36- In general, not clear when the events are. I would highly suggest making a table including all the events discussed (and their corresponding figure) and also denoting each event on the time series plot.

- Thank you for this suggestion. A table has been added to beginning of results section, see Table 1.

Review #2: Following the addition of a table listing changes to the met data, this is now Table 2.

37- P 19, L1-3: This sentence is extremely confusing. For ASO or for the seasons with the most significant haze events including MJJ, FMA, and ASO?

- Sentence moved to results section and reworded to read:

"Of the seasons with the most significant haze events (e.g., MJJ 2013, FMA 2014, ASO 2014, and ASO 2015) in Singapore, the air history maps show that the region of influence for Singapore generally covers the largest area during ASO when air is coming from southeasterly directions"

38- Please check the sentence constructions of discussions and conclusions sections carefully.

- as mentioned above (Results comment 4) these sections have been restructured and reworded. Please see revised manuscript.

Review #2: Please refer to the attached pdf document highlighting changes between the original and the revised manuscripts.

39- P19- 20: The discussions seems like an extended conclusion section and there is a lot of redundancy between results (section3), discussions (section 4) and conclusions (section 5), which decrease the readability of paper.

- as above (Results comment 4) these sections have been restructured and reworded, please refer to the manuscript for updated text.

**40- P20, L30: Please restate this sentence. It is very confusing.**

**- the sentences now read:**

"Looking at emissions during ASO for the four years with the most variation across the island (2011, 2012, 2014, and 2015), the largest emissions were seen from Central Kalimantan, South Sumatra, Jambi, and also West Kalimantan. For events during FMA Cambodia, East Kalimantan, Myanmar, Thailand, and Vietnam showed larger emissions during FMA."

41- Code and data availability: Please include the link to the observations used for this study.This is not possible as the observations are not publicly available.

**Specific Comments**

The third sentence of the introduction is a run-on sentence. Additionally, the second sentence of the second paragraph in the introduction is a run-on sentence. I will not note other grammatical errors, but someone must correct these and others before this article is suitable for publication.

- These sentences read/have been edited to read:

- Clearing forest for plantations by burning is a quick and easy way to open up and fertilise the soil, however, it is also a process that is difficult to control. The emissions from these fires can have massive and detrimental impacts far from where the original fires were lit.
- Scientific studies such as Kim et al. (2015), as well as the popular press, often attribute peatland destruction and related haze in the region to Indonesia (Reid et al., 2013). However, the haze cannot be attributed to only one region or country alone.

Throughout the document sentences have been shortened, please refer to the document detailing differences between the original submission and the revised manuscript.

Acronyms including but not limited to NAME, GFAS, and CAMS are introduced after being used previously. Please ensure that every acronym is introduced upon first usage.

- This has been corrected.

"Sec" is not an appropriate abbreviation for "Section". Please replace all occurrences.

- Following the "Manuscript preparation guidelines for authors" (see excerpt below) "Sec." has been replaced with "Sect.":

Sections: The headings of all sections, including introduction, results, discussions or summary must be numbered. Three levels of sectioning are allowed, e.g. 3, 3.1, and 3.1.1. The abbreviation "Sect." should be used when it appears in running text and should be followed by a number unless it comes at the beginning of a sentence.

"Air history" is an inaccurate term to refer to the convolution of emissions and back trajectory information. Please revise throughout.

- The back trajectories do not consider emissions, please see response to the general comments above.

**Line Comment**

p. 4, I. 16 "validated" should be "evaluated" here and elsewhere (e.g., p.7, I. 4). Please change all occurrences when speaking of a comparison of measurements and models. Both have errors, which indicates that neither is sufficient for validating the other.

- "validated" has been replaced by "evaluated" throughout the manuscript.

p. 7, I. 25-7 This statement conflicts with the last sentence of the abstract, which states that "variation in local meteorology can impact concentrations of particulate matter significantly". If that were true, it would not be sufficient to use a central meteorological site. Please resolve by removing one of the statements. If the abstract statement is not changed, then the entire study needs to be presented for only one Singapore site. If a single receptor site was used for the back trajectories, it is not clear how the modeled concentrations in Figure 4 would be distinct as they appear to be or how these distinctions were investigated as indicated on p. 21, I. 1-3.

- This only applies to the air history maps where the concentrations are averaged over time (seasons or years) and biomass burning emissions are not considered.

We feel that the original (multi-coloured) time series plot made it easier for the reader to distinguish the contributions and relate the time series and pie charts.

**Haze in Singapore - Source Attribution of Biomass Burning $\underline{PM}_{10}$ from Southeast Asia**

Ayoe Buus Hansen1, Claire Suzanne Witham1, Wei Ming Chong2, Emma Kendall1, Boon Ning Chew2, Christopher Gan2, Matthew Craig Hort1, and Shao-Yi Lee2

[revised manuscript text omitted]

---

## Author Response (AR3)

**Response to review of 28th February 2019.**

First of all, we want to thank both the reviewer and the co-editor for taking the time to review our manuscript and for acknowledging the work put in to the paper during the previous revisions.

Following the feedback of this review we have focussed on the two points highlighted by bold in the comments from the reviewer below. As these were minor revisions that we agree with this has been done and the revised manuscript is uploaded. As stated in the previous review we would be more than happy to provide figures directly in full resolution.

**Co-Editor Decision: Publish subject to minor revisions (review by editor) (28 Feb 2019) by Ari Laaksonen**

Comments to the Author:

I believe that your paper has merits although, as noted by the reviewer, correspondence between the model and real world is not perfect. Please prepare the final version of the manuscript and take account of the reviewer's recommendations.

**Review of second revision of Hansen et al. for Atmospheric Chemistry and Physics**

General Comments

In the second revised submission of "Haze in Singapore - Source Attribution of Biomass Burning from Southeast Asia", Hansen et al. responded to the comments made in the first two rounds of reviews. Importantly, they clarified the application of the forward and backward runs of NAME. The backward run of NAME was done to identify the extent of the domain necessary for forward Lagrangian modeling with tagging of particles. The forward run was then conducted with particles that were tagged with the source region. This method is appropriately deemed source attribution. Since some other authors have used the information in the backwards run with observations to ascertain emissions rates, it should be clear that the only utility of the backward run was to show the extent of the domain. The "air history maps" do not contribute at all to the estimate of the influence of biomass burning on the sites; all of this information comes from the forward execution of NAME. Therefore, **I would recommend changing the order of the methods section by moving 2.4 to 2.3 so that the backwards run can be described as a forerunner to the forwards runs.**

The poor agreement between the model and observations is still concerning, but the authors have characterized this error. Since the observations are not used in ascertaining emissions rates, the reader is left to understand that the "source attribution" is only of the modeled concentrations rather than those observed. To this point, the authors seem to indicate in the responses to Reviewer 2 that "attribution" differs from "apportionment", which seems to be the justification for not treating the background concentration or other sources of PM10 more carefully. However, any literature review will demonstrate that "attribution" and "apportionment" are used interchangeably in atmospheric modeling literature. Therefore, the paragraph beginning on page 11, line 25 of the revised manuscript should not depend on the distinction of these two words. **Rather, using "attribute" instead of "perform an apportionment of" would appropriately indicate that there is no distinction between the meaning of these two words. Also, it would keep intact the clarifying argument the authors make that they are not trying to attribute observed PM10 concentrations but modeled concentrations.** It is left to the editor to decide whether apportioning modeled concentrations is a valuable endeavor when they differ as much as these do from observations.

Accordingly, an extensive effort to clarify the nature of the work has helped tremendously in this revision of the manuscript.

[revised manuscript text omitted]